# Frazzled/DCC directs spatial progenitor integration ensuring steady-state intestinal turnover

Lisa Zipper [1], Pol Ramon-Cañellas[2], Filiz Akkas-Gazzoni[3] & Tobias Reiff [1] ✉

Adult epithelial organs undergo continual steady-state turnover that is achieved by tight coupling of stem cell production with replacement of worn-out epithelial cells by local intercellular signalling[1,2]. Like many eukaryotic epithelia, absorptive enterocytes (EC) of the adult *Drosophila* midgut are arranged in a hexagonal, honeycomb-like pattern. On tricellular nexuses of EC, intestinal stem cells (ISC) are scattered in a way so that around two thirds of EC can be renewed directly by adjacent ISC. However, the mechanism for replacement of the remaining third of remotely located EC is unknown.

Here, we show that a conserved axonal guidance cue directs enteroblasts (EB), the immediate ISC daughters, to selectively replace worn-out adjacent and remote EC with identical frequency. Worn-out EC express Netrin-B ligands that attract Frazzled/DCC-receptor dependent EB protrusions and subsequent EB migration towards the Netrin-B expressing EC. Our newly developed 'Hamelin' assay confirms Frazzled-dependent EB migration towards Netrin-B sources and hints to invasive progenitor behaviour as midgut progenitors cross the organ boundary into the hindgut. Together, we establish spatially directed EB migration and integration as essential for intestinal homeostasis and provide first mechanistic support for recent findings resuscitating conserved Netrins and Frazzled/DCC-signalling as therapeutic target in metastasis.

Epithelial cells in adult organs are continuously renewed by resident stem cell divisions. Robust mechanisms ensure constant organ size by tight temporal coupling of stem cell divisions to cell loss during steady-state turnover. For the study of stem cell behaviour, the adult *Drosophila* posterior midgut epithelium (PMG) proves to be a powerful model due to its conserved structure and molecular mechanisms in physiology and pathology[3]. The PMG epithelium is composed of five major cell types and replenished by multipotent ISC. Asymmetric ISC divisions give rise to two transient progenitor cell types: enteroblasts (EB) that differentiate into absorptive enterocytes (EC)[4–6] and enteroendocrine precursor cells that were shown to symmetrically divide into two enteroendocrine cells (EE)[7]. Under conditions of homoeostatic intestinal turnover, nine out of ten asymmetric ISC divisions give rise to EB ensuring a constant pre-existing pool of progenitor cells[8–10].

In the monolayer of the intestinal epithelium, ISC and EB are basally located and can be identified by the marker gene *escargot*[4–6]. Once needed to replenish EC, Delta ligands on ISC rapidly initiate differentiation of EB from the progenitor pool by activating Notch signalling[4–6]. Notch activity in EB leads to the expression of the EB-lineage specifying marker gene *klumpfuss (klu)*, which enables EB fate tracing methods such as *klu^{ReDDM}* for precise monitoring of individual EC renewal[11,12]. Using *escargot* as marker gene, we and others noticed that ISC and EB progenitors occur in duplets that are scattered between absorptive EC. EC are arranged in a hexagonal, honeycomb-like structure sealed by tight- so called septate junctions in *Drosophila*

[1]Heinrich Heine University, Institute of Genetics, Düsseldorf, Germany. [2]Hospital Universitari Institut Pere Mata, Institut d'Investigacions Pere Virgili, Reus, Spain. [3]University of Cologne, Medical Faculty, Institute for Pharmacology, Cologne, Germany. ✉e-mail: reifft@hhu.de

and duplets of ISC and EB primarily locate on tricellular EC nexuses. In homoeostatic midguts undergoing physiological turnover, we noticed that the total frequency of duplets does not suffice to cover the whole EC population with a direct contact to an ISC. Thus, we hypothesized that a population of remotely located EC (rEC) exists, that would need either the ISC or the EB to actively migrate towards a worn-out EC (depicted in Fig.1B).

Supporting this hypothesis, ISC and EB progenitors were shown to occasionally migrate short distances under homoeostatic conditions and confirmed in sophisticated intravital live-imaging setups[9,13,14]. Although in vivo approaches revealed important information of the dynamics of ISC behaviour and cellular homoeostasis, the comparably slow turnover under physiological conditions impedes mechanistical studies of dynamics. As a reference for that slow homoeostatic turn-over: along the whole PMG, just a handful EC replenishments events take place per day. Luckily, this relatively rare renewal of EC can be visualized over long periods with established tracing systems like 'ReDDM'[9–11,15]. As ReDDM depicts single EC renewal events over long tracing periods, we hypothesized that the discrimination between rEC and adjacent (aEC) renewal will allow quantitatively robust retroactive evaluation of the involvement of progenitor migration during intestinal homoeostasis.

Here, we discovered that EB renew aEC and rEC with identical frequency. In a candidate screen approach for signalling receptors putatively involved in migratory EB behaviour, we identify and characterize Frazzled (fra) / DCC (Deleted in Colorectal Cancer) signalling in the control of protrusion formation on EB that initiates spatially

directed movement towards worn-out EC. Worn-out EC secrete Netrin-B (NetB) ligands that induce EB protrusions and attract the migration of EB towards the NetB source through Fra/DCC-receptors. This is further supported by our newly developed 'Hamelin' assay that introduces dual binary expression systems for the detection of progenitor migration. Hamelin confirms long-distance Fra-dependent attraction and migration of midgut progenitors towards Netrin sources. Underlining the functional conservation and the general interest of our observations, human transgenes of Netrin-1 (hNTN1) and Netrin-5 (hNTN5) and DCC receptors phenocopy their respective fly orthologues. The physiological relevance of spatially controlled EB migration on intestinal homoeostasis is highlighted by early fly demise when NetB-Fra/DCC signalling is impaired. Together, our data establishes spatially directed EB migration by NetB-Frazzled/DCC-signalling as essential for intestinal homoeostasis and adds first mechanistical insights for the recent clinical efforts that aim at targeting Netrin-Fra/DCC signalling in tumour EMT and metastasis.

## Results

### About a third of the EC population is replenished by EB migration

EC are the most abundant intestinal cell type and once worn-out, require renewal through ISC mitosis and subsequent differentiation of EB[4–6]. The relatively rare renewal of EC under homoeostatic conditions can be visualized over long periods with established tracing systems like 'ReDDM' (Fig. 1A)[9,11]. Performing all experiments in the R5-region of the PMG[16], we traced physiological turnover for a period of seven days

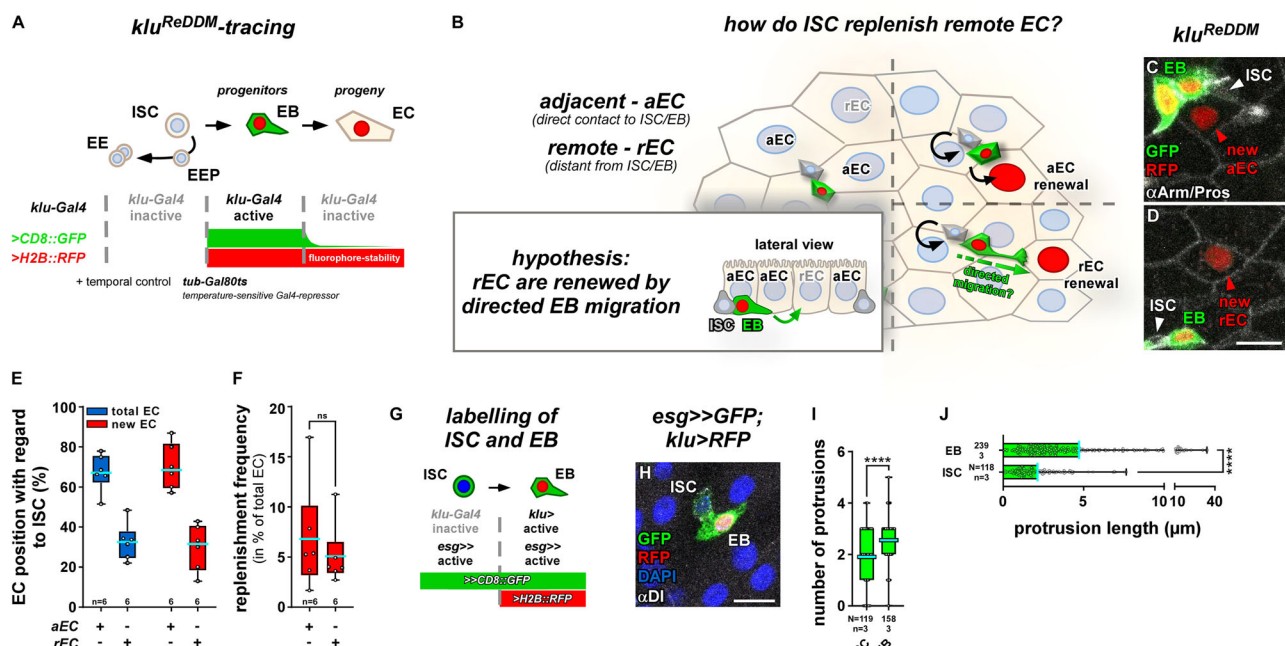

**Fig. 1 | About a third of the EC population is replenished by EB migration.**
**A** Schematic illustration of $klu^{ReDDM}$-tracing. EB specific $klu >$ ('>' abbreviates Gal4/UAS regulation hereafter) drives expression of $>CD8::GFP$ with short half-life and $>H2B::RFP$ with higher stability which persists in newly differentiated EC with inactive $klu >$. Temperature sensitive Gal80 ($tub\text{-}Gal80^{ts}$) ensures temporal expression control. **B** Scheme depicting distribution and renewal of adjacent EC (aEC) and remote EC (rEC). Confocal images of $klu^{ReDDM}$ controls ($w^{1118}$) (R5) with antibody staining targeting Armadillo (Arm) and Prospero (Pros) and showing renewed aEC (red arrowhead) with direct contact to ISC (white arrowhead, **C**) and rEC (red arrowhead) distant from ISC (**D**). Scale bar is 10 μm. Quantification of aEC and rEC as percentage of total EC and new EC (**E**), and replenishment frequency of aEC and rEC compared by two-sided Mann–Whitney test (**F**, $p = 0.4848$).

**G** Schematic illustration for differential labelling of midgut ISC and EB. ISC and EB are both labelled by $esg\text{-}lexA$ driven expression of $Aop\text{-}CD8::GFP$ (lexA/Aop abbreviated as '»' hereafter) whereas only EB are additionally labelled by $klu > H2B::RFP$. **H** Confocal image showing differentially labelled ISC (GFP+ aDl+ RFP-) and EB (GFP+ aDl- RFP+) in the R5 region. Quantification of protrusion numbers (**I**) and lengths (**J**) in ISC compared to EB. **E, F, I** Box plots show 25th and 75th percentiles with indication of ranges by whiskers. **J** Bar plots show means and range indicated by whiskers. **E, F, I, J** All data points are shown by dots and the means are indicated by light blues lines. 'n' are numbers of biological replicas and 'N' values describe numbers of technical replicas. Asterisks denote significances from comparisons by two-sided Mann–Whitney tests (****$p < 0.0001$).

and noticed that around ten percent of total EC undergo renewal. Importantly, we noticed that a third of EC possess no adjacent ISC/EB duplet (Fig.1B, D, E), which poses the question how such remote EC are renewed (rEC, Fig.1B). We discovered that renewal of single rEC, indicated by a red nucleus, occurs frequently and, importantly, also in the absence of adjacent ISC/EB nor any further renewed EC in the surrounding (Fig.1D). Strikingly, by quantifying hundreds of EC renewal events in homoeostatic midguts, we found that rEC and aEC (Fig.1B–D) are replaced with identical frequency (Fig.1F). Such selective replenishment of rEC would imply two central prerequisites: (1) migration of progenitors, which was previously demonstrated[9,17] as well as (2) an underlying molecular attraction cue providing spatial guidance information for progenitors towards dying EC.

Previous data shows that the midgut progenitors, ISC and EB, possess mesenchymal characteristics[18] and extend membrane protrusions with so far unknown function in midgut physiology[9,10]. Thus, we assessed ISC and EB protrusion lengths under homoeostatic conditions and found that protrusions on EB, identified by the lineage marker *klumpfuss (klu*, Fig.1A, G, H)[11,12], occur more frequent (Fig.1I) and are longer (Fig. 1J) than previously described ISC (*esg⁺,klu⁻*) lamellipodia formed upon laser ablated injury[17]. Membrane protrusions are well-known in coordinating cell migration during development, which prompted us to screen for guidance receptors expressed in the PMG.

### Fra/DCC and Unc5 receptors are expressed in EB and locate to membrane protrusions

Consequently, we examined in silico resources for guidance cues known to coordinate directed migration[19] and isolated the conserved transmembrane receptors *frazzled/DCC (frazzled, fra* in *Drosophila* and *Deleted in Colorectal Cancer, DCC* in *humans)* and *unc-5 (un-coordinated* in fly and human)*, and their ligands Netrin-A and -B (NetA, NetB). NetA and NetB signalling through Fra and Unc-5 is key for guiding cell migration and axon outgrowth during *C. elegans, Drosophila* and mammalian nervous system development[20–23] and might thus fulfil a guidance function in the intestine as well.

All four genes are expressed in the midgut (Fig. S1A) and confocal analysis of endogenously GFP-tagged Fra and Unc-5 receptors revealed exclusive expression of both receptors in EB (Gbe+Su(H)-dsRed⁺, Fig. S1B-C,F-G), but not in ISC (Delta⁺, Fig. S1B-C,F-G), EC (Discs-large 1⁺, Fig. S1D-D',H-H') and EE (Pros⁺, Fig. S1D-D',H-H') using established antibodies identifying each cell type. Supporting a putative role for Fra/DCC in EB protrusion formation, GFP-tagged Fra receptors driven from an endogenous promoter and tagged Fra/DCC receptors driven by UAS ('>' abbreviates Gal4/UAS regulation hereafter) localize to EB membrane protrusions (Fig. S1C,I-I'',J-J''', arrowheads) with yet unknown function[9,10]. Underlining a role for Unc-5 in EB, we detected Unc-5 with antibodies (Fig. S1E-E'') and GFP-tagged Unc-5 (Fig. S1F-G) in Gbe+Su(H)-dsRed⁺/Delta⁻-progenitor cells. Interestingly, we also detected internalization of GFP- and even overexpressed HA-tagged Unc-5 receptors into vesicles (arrows in Fig. S1G,I''-I'''), which was previously shown to indicate active fra/DCC signalling[24]. We next tested whether Fra/DCC receptors on EB protrusions might enable integration of spatial information on the whereabouts of an EC in need of replacement.

### Fra/DCC receptors control protrusion formation on EB that preferentially contact old EC

Therefore, we employed *klu^ReDDM*, an established spatiotemporally controlled EB lineage manipulation and tracing method, which allows retrospective analysis of all single EC replacement events during a given tracing period (Fig.1A, B)[11]. Using *klu^ReDDM* (Fig.1A), we investigated *fra* and *unc-5* up- and downregulation and its impact on EC renewal in midgut EB for a tracing period of seven days. Strikingly, overexpression of *fra* induces more numerous (Fig.2B, Fig. S2A) and two-fold longer EB protrusions (Fig.2H) compared to controls (Fig.2A).

In contrast, Fra-depletion using three different RNAi stocks (>fra^RNAi, Fig.2C, Fig. S2B–E,), guideRNA (gRNA)-mediated Cas9 excision (>fra^gRNA, Fig.2F, Fig. S2Q-R) and a dominant-negative truncated Fra receptor lacking the cytoplasmic domain (>fra^Δ, Fig. S2I–J)[25], strongly reduce protrusion number (Fig. S2A,G,L,U) and length (Fig.2H, Fig. S2H,K,T). For these experiments, we extended *klu^ReDDM* with >*Cas9* and detected over 90% excision efficiency with guides targeting eGFP in the ReDDM stock (Fig. S2O-P',S). Developmental studies of Fra and Unc-5 during axon guidance revealed an antagonism between both receptors by showing that heterodimers lead to repulsion, whereas Fra homodimers function in attraction[26]. Underlining this antagonistic function of Fra and Unc-5 receptors in midgut EB, RNAi-knockdown and gRNA mediated excision of *unc-5* induce protrusion length and number (Fig.2E, H, Fig. S2A,F–H,V–Y) whereas forced expression of >*unc-5* phenocopies Fra-depletion (Fig.2D, H, Fig. S2A). Epistatic analysis of simultaneous *fra* and *unc-5* overexpression and knockdown conditions shows that the *fra* phenotype prevails (Fig. S3A–G). This suggests that in EB Fra/DCC receptors dominate in the control of protrusion formation and rEC renewal (Fig. S3A–G), which is further supported by our finding that Unc-5 is internalized into vesicles (Fig. S1G,I''''). Thus, our data supports the idea that Unc-5, either being internalized or depleted genetically, allows signalling for protrusion formation and migration through Fra homodimers similar to axon guidance[26]. Indicative for high functional conservation, human DCC phenocopies protrusion formation and number in *fra* mutant EB (Fig.2G, H, Fig. S2A).

Next, we followed the hypothesis that EB protrusions might serve detection of worn-out EC. Consequently, EB protrusions should preferentially extend towards old EC rather than recently renewed EC. Strikingly, we found several lines of evidence for a role of Fra/DCC-controlled EB protrusion directionality in 'EC health assessment': (i) By hand-selecting and analysing protrusions of EB 'with a choice', located between old and renewed EC (Fig.2I–K), we found that around 80% of protrusions project towards old EC in controls (Fig.2I, K). (ii) Up- and downregulation of Fra and Unc-5 receptor levels randomizes protrusion directionality, suggesting that directionality of protrusions depends on a fine-tuned balance of Fra and Unc-5 receptor abundance at the membrane and additional mechanisms such as internalization (Fig.2K, Fig. S1G,I''') as described for Unc-5 receptors previously[24]. In addition, it was described previously that removal of Unc-5 from the membrane reflects a switch from repulsive to attractive signalling through Fra/DCC homodimers[27]. This notion is further supported by the finding that only homodimers of Fra/DCC are able to connect to the cytoskeleton and thus enable protrusion extension[25,28]. (iii) EB located between renewed EC extend less protrusions (Fig. S3H–J, arrow), (iv) whereas EB surrounded by old EC survey more actively (Fig. S3H-J, arrowhead). (v) Most strikingly, Fra-LOF conditions and >*fra^Δ* diminish the renewal of rEC (Fig.2L, Fig. S2M).

Following the idea that EB protrusions serve EC replenishment, overall tissue renewal is accelerated under protrusion stimulating conditions (Fig.2B, E, M), whereas manipulations abolishing protrusion formation disrupt midgut turnover reflected by reduced numbers of rEC, aEC and total new EC (Fig.2L, M, S2N). It is well established that intestinal integrity is compromised without constant EC renewal, which prompted us to test whether *fra* and *unc-5* manipulations impact overall survival. Underlining the physiological relevance of Fra/DCC-signalling during midgut homoeostasis, lack of Fra-mediated EC replenishment upon depletion of *fra* in EB results in early fly demise (Fig.2N, P). To exclude putative effects of extra-intestinal activity of *klu >*, we additionally employed the bipartite midgut progenitor-specific I-KCKT expression system to assess survival of *fra*-depleted flies and observed identical reduction of fly survival (Fig. S2Z)[29]. Like >*fra-RNAi*, overexpression of *unc-5* in EB reduces not only rEC and total EC formation (Fig.2L, M) but also fly survival (Fig.2O). Although these effects appear relatively mild, one has to keep in mind that flies can

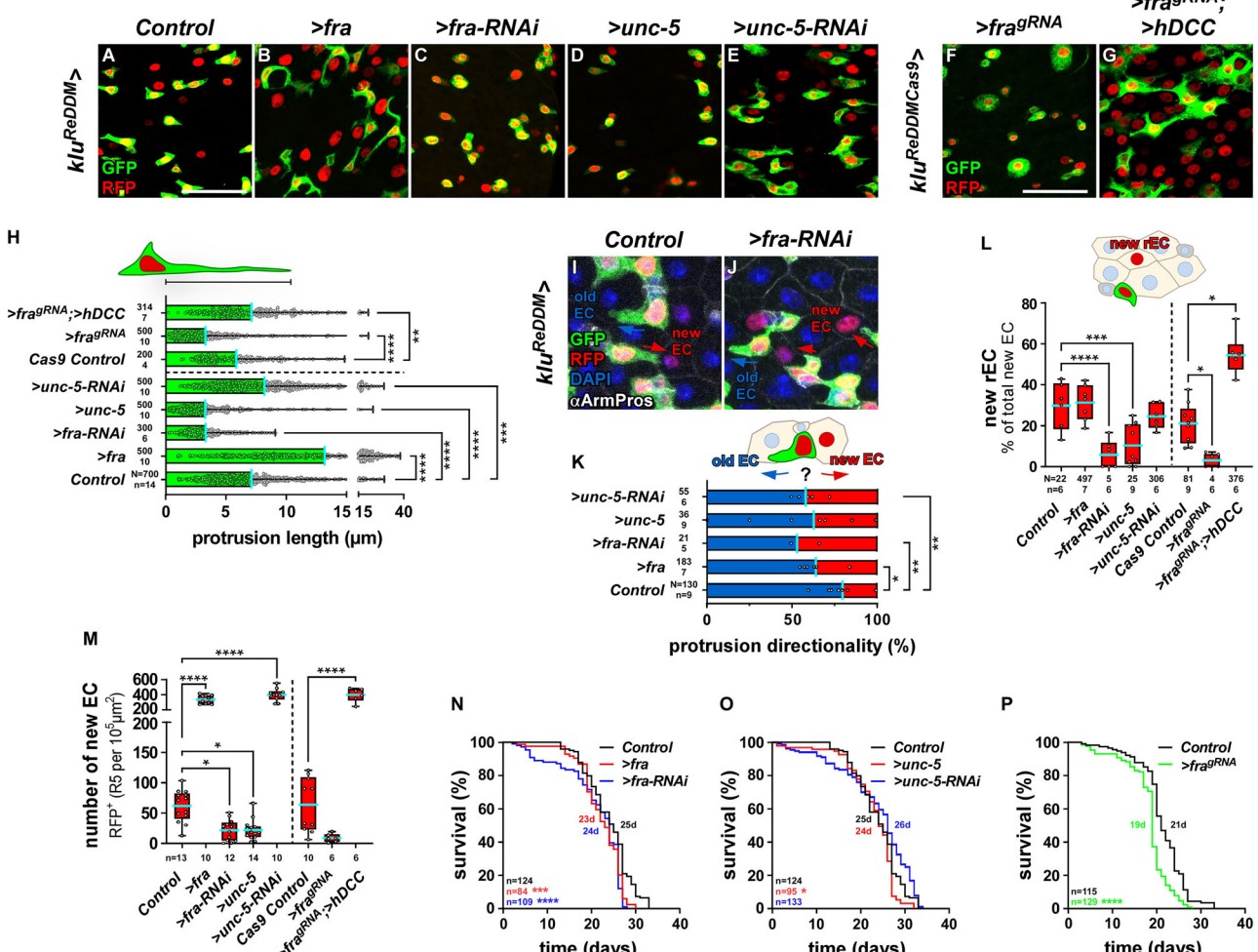

**Fig. 2 | Fra/DCC receptors control protrusion formation on EB that preferentially contact old EC. A–G** Confocal images of midguts (R5) after 7 days of $klu^{ReDDM}$-tracing and EB specific manipulations of Fra/DCC and Unc-5 receptors. Scale bar is 50 μm. **H** Quantification of EB protrusion lengths upon Fra/DCC and Unc-5 manipulations ($p = 0.0003$ for *>unc-5-RNAi; p = 0.0011* for *>fra^{gRNA};>hDCC*). Confocal images of $klu^{ReDDM}$ controls ($w^{1118}$) (R5) (**I**) and knockdown of *fra* (**J**) showing EB with protrusions targeting to old EC (blue arrows) and to new EC (red arrows). **K** Quantification of protrusions targeting to old EC and to new EC upon EB specific manipulations of Fra and Unc-5 ($p = 0.019767; p = 0.001382; p = 0.177377; p = 0.003663$). **L** Quantification of rEC renewal frequency upon manipulations of Fra/DCC and Unc-5($p = 0,7804$ for *>fra; p = 0.0003* for *>unc-5; p = 0.3158* for *>unc-5-*

*RNAi*). **M** Quantification of new EC numbers upon $klu^{ReDDM}$-specific manipulation of Fra/DCC and Unc-5 receptors ($p = 0.0282$ for *>fra-RNAi; p = 0.025* for *>unc-5*). **N, P** Kaplan-Meier estimations of survival in flies with $klu^{ReDDM}$-driven manipulations of Fra and Unc-5 receptors. Mean survival time is shown in days (d). **L, M** For box plots, the boxes show median, 25th and 75th percentiles, and whiskers indicate the range of values. (H) Bar plots show means and range indicated by whiskers. All data points are shown by dots and means are indicated by light blues lines. **L–P** 'n' are numbers of biological replicas and 'N' values describe numbers of technical replicas. Asterisks denote significances from multiple comparisons by Kruskal Wallis tests (**H, L, M**) and Long-rank tests (**N, P**) (*$p < 0.05$; **$p < 0.01$; ***$p < 0.001$; ****$p < 0.0001$).

survive a 30% reduction in intestinal length when EC replenishment is blocked[15]. Interestingly, overexpression of *fra* (Fig.2N, Fig. S2Z) reduces fly survival as well. A possible explanation is that the strong differentiation effects of *>fra* towards EC fate (Fig.2M) result in a loss of cellular intestinal homoeostasis[9,10]. Non-autonomous stimulation of ISC mitosis by EB differentiation has been observed previously[11] and over time exhausts the ISC-pool[30] resulting in the loss of epithelial barrier function and fly demise[31,32]. Additionally, we also approached Fra/DCC LOF on whole organism level using established *fra^3* and *fra^4* constitutive null mutant alleles[33]. Crossing *fra^3* to *fra^4* mutant flies leads to viable but infertile heteroallelic flies (Fig. S3K, M), with reduced survival (Fig. S3N) supporting our findings for Fra-depletion (Fig.2N, P).

In addition, we aimed to validate functionality of the *fra::GFP* transgenic flies (Fig. S1B–D') and found that Fra::GFP is able to rescue both mutant alleles to obtain viable flies (Fig. S3K-M). Suggesting additional ethyl methanesulfonate-induced mutations on *fra^3* and *fra^4* chromosomes[33], only heteroallelic flies present a rescue in thorax

phenotypes (Fig. S3L) and only *fra^3* homozygous mutant flies can be kept as a fertile stock (Fig. S3M). Thus, non-specific mutations on *fra*-mutant chromosomes might be responsible for infertility of *fra^4* homozygous mutants (Fig. S3M) and differing results in our MARCM clonal analysis (Fig. S3O–R)[34]. Like ReDDM-tracing, MARCM labels ISC progeny of *fra^3* and *fra^4* flies. When we quantified GFP⁺-cell number, we found that both alleles significantly reduced clone size (Fig. S3O–R), which supports the reduction of new EC observed with EB-specific Fra-depletion (Fig.2M). Additionally, more small and single cell clones are observed (Fig. S3R), which suggest either mitotically inactive ISC or single EB clones described previously[8]. Unfortunately, non-clonal, fra-heterozygous areas, show strong distortion of the midgut epithelium seen in the tight and irregular distribution of EC shown with nuclear Dapi-staining (Fig. S3P–Q). Thus, we did not further continue experiments with both alleles, as our RNAi, gRNA and dominant-negative Fra experiments in $klu^{ReDDM}$ using multiple stocks showed consistent unanimous phenotypes. Together, our findings implicate Fra/DCC-

dependent EB protrusions in detection of worn-out aEC and rEC and suggest that EB migration to replace rEC is crucial for midgut homoeostasis.

## Fra and Unc-5 induce EB traits and Notch-dependent EC differentiation

One important question arising from these findings is at which time-point Fra and Unc-5 play a role in the transition from ISC over EB to EC in the intestinal lineage. To test whether Fra and Unc-5 can ectopically induce EB traits in ISC, we performed experiments in which we manipulated ISC and ISC-like cells reflecting different stages of ISC maturation towards EB. For ISC-specific manipulation of both receptors, we modified the $esg^{ReDDM}$[9] by adding the EB-specific Su(H)-Gal80 repressor[35], enabling ISC-restricted manipulation as well as labelling of ISC (GFP+RFP+) and their progeny (RFP+ only, Fig.3A). Using ISC-specific ReDDM allowed us to analyse effects of Fra and Unc-5 on protrusion formation and tracing of rEC renewal when only ISC but not EB are manipulated.

Importantly, knockdown of *fra* or *unc-5* using ISC-specific ReDDM had no effect on protrusions or rEC renewal (Fig.3D, F–H), which further underlines our findings that both receptors are exclusively expressed and localized in EB only (Fig. S1). Ectopic ISC-specific expression of >*fra* and >*unc-5* increased protrusion lengths (Fig.3B, C, E, G). Although less pronounced than in EB (Fig.3G), protrusion formation in ISC indicates first traits towards differentiation of EB (Fig.1H)[9,10]. Beyond enhancing protrusions, ISC-specific ectopic >*fra* expression increases ISC and progeny numbers (Fig.3I-J), which further indicates its role in EB differentiation (Fig.2M) by stimulating subsequent non-autonomous ISC proliferation[11]. Conversely, ISC-specific >*unc-5* expression decreased progeny numbers (Fig.3J), which is consistent with opposing functions of the two receptors in lineage production. The earliest event in EB differentiation is activation of Notch signalling, which we approached in a second experiment.

In the fly and mammalian ISC niche, Notch signalling specifies EE and EC fate[4–6,36,37]. In *Drosophila*, mitosis of Notch-mutant ISC generates ISC-like progenitor cells and EE, which results in tumours of ISC-like progenitors (defined in ref. 38) and EE and an intestinal epithelium almost entirely lacking the EB lineage and thus newly produced EC (Fig.3K)[7,38,39]. Here, we used these 'Notch-LOF tumours' as a second experimental paradigm lacking Notch-induced EB differentiation[15] to investigate whether ectopic Frazzled and Unc-5 depletion affects protrusion formation and EB differentiation in ISC-like progenitors.

As previously reported[4–6], Notch receptor LOF in $esg^{ReDDM}$ traced midguts leads to tumours of different sizes (Fig.3M)[15]. Confirming our finding of EB having longer protrusions than ISC (Fig.1J), we detected reduced protrusion length of 'ISC-like' Notch-LOF progenitors (>*CD8::GFP*+) compared to normal $esg^{ReDDM}$ control midguts (Fig.3L–M, P). Arguing for a role of Fra and Unc-5 in EB protrusion formation, >*fra* as well as >*unc-5-RNAi* within ISC-like progenitor cells increase protrusion lengths (Fig.3N–P) and thus EB traits comparable to Fig.3G. A function of >*unc-5-RNAi* in ISC-like progenitors (Fig.3O-P) but not ISC (Fig.3F, G–J) suggests that *fra* and *unc*-5 start to be expressed quite early in the ISC to EB transformation that is occasionally observed in ISC-like progenitors[15]. To replenish worn-out EC, EB differentiation includes a substantial increase in cell size[40]. When we investigated whether >*fra* and >*unc-5-RNAi* induce such size increases in 'ISC-like' cells, we measured that both manipulations increase nuclear size (Fig.3M–O, Q) in accordance with previous reports describing endoreplication-dependent growth as part of EB differentiation[40].

Together, these data show that forced expression of >*fra* and depletion of *unc-5* by RNAi induce EC differentiation in $klu^{ReDDM}$ (Fig.2B, E, M) and seem to stimulate formation of more 'EB-like' cells in Notch-LOF tumours (Fig.3P–Q). However, both manipulations do not significantly increase EB integration as EC in the absence of Notch demonstrated by unchanged ratios of new EC per 'ISC-like' cells (Fig.3R) and in comparison with previously described factors that drive EB and EC formation independently of Notch such as *nubbin*[18] and *Eip75B*[15]. This suggests that Fra and Unc-5 induce specific EB traits without active Notch-signalling. Terminal EB to EC differentiation however remains Notch-dependent and Fra per se is not sufficient to induce intestinal lineage differentiation. Together, these data made us wonder whether the observed induction of protrusions might be a prerequisite of migratory processes.

## EB protrusions and migration depends on binding of Fra/DCC to the actin cytoskeleton

Next, we aimed to confirm whether the molecular framework known from neurodevelopmental studies[41,42] also controls Fra/DCC-induced protrusion formation in intestinal progenitors. Fra and DCC receptors share three highly conserved cytoplasmic P-motifs (P1-P3) in their intracellular domain that upon receptor homodimerization connect Fra to the F-Actin polymerization complex factors[25,28]. Well-described members of the F-Actin complex encompass *Arpc3* (Actin-related protein 2/3 complex, subunit 3B), *ena* (enabled, Ena/VASP), and the RhoGEF *trio*, a general regulator of cell motility and axon outgrowth and guidance[43]. To confirm their role in the intestinal progenitor lineage, we depleted the established Fra/DCC interactors *Arpc3*, *ena*, and *trio* using $esg^{ReDDM}$. Confirming previous findings in neurons, *Arpc3*, *ena* and *trio* depleted progenitors fail to elongate protrusions (Fig. S4A–D,L–M, Fig. S5A–N) and to migrate, reflected by the absence of rEC (Fig. S4N). Importantly, protrusions remain absent even when protrusion formation is genetically stimulated by >*fra* (Fig. S4E–H,L–N) or >*unc-5-RNAi* (Fig. S4I–K,L–N) confirming the essential role of Arpc3, Ena and Trio downstream of Fra.

During fly development, cell motility depends on the P3 domain of Fra[28,44]. In line with this, expression of Fra receptors lacking the P3 domain in *fra*-mutant background (>*fra^{ΔP3}*, Fig.4B, F) abolishes rEC renewal (Fig.4I), whereas EB with Fra receptors lacking P1 or P2 domains migrate normally (>*fra^{ΔP1}*, >*fra^{ΔP2}*, Fig.4D, E, I). However, Fra receptor levels induced by UAS-driven expression of Fra-variants result in increased protrusion numbers (Fig.4G) and lengths (Fig.4H) when compared to controls with endogenous Fra levels (Fig.4A–H). Interestingly, >*fra^{ΔP3}* expression shows an intermediate phenotype rescuing the >*fra^{gRNA}* phenotype partially to control levels but not to the full extent of >*fra*, >*fra^{ΔP1}* and >*fra^{ΔP2}* levels concerning protrusion lengths and rEC renewal (Fig.2H, L, Fig.4H–I). One explanation for this observation is a weakened interaction with the RhoGEF Trio and the Abelson Kinase that are both known to bind to all Fra P-motifs. The improper function of this signalling hub, when the P3 domain is lacking, might disrupt proper binding to the cytoskeleton[41,45], reduce multimerization of Fra[42,44] and thus reduce protrusion length and EB motility as shown previously in the nervous system[28,44]. Molecular details of the differential interaction of P3 vs. P1 and P2 will be an interesting subject for future studies. Supporting Fra receptors as central molecular cue for directed EB migration by extracellular ligands, the cytoplasmic intracellular domain of Fra, containing all three P-motifs but no extracellular domain, is unable to initiate protrusions and rEC generation (>*fra^{ICD}*, Fig.4C, G–I). Given this importance for Fra/DCC and Unc-5-signalling in migration and pathfinding in EB, we addressed the role of Netrin ligands in the midgut epithelium.

## Worn-out EC express Netrin-B to attract EB protrusions and direct EB migration

Netrin ligands guide axon outgrowth and neuron migration in nematodes, vertebrates and flies. Constitutive double mutant fly embryos for both *Drosophila* Netrins entirely lack Fra-dependent commissural projections[21,46]. Homozygous *NetA/NetB*-double deficiency (*NetA^{-/-}*,*NetB^{-/-}*) is lethal[21], but progeny of *NetA^{-/+}*,*NetB^{-/+}* mutants crossed to single *NetA* or *NetB* mutant flies survive to adult imago[47]. By crossing in

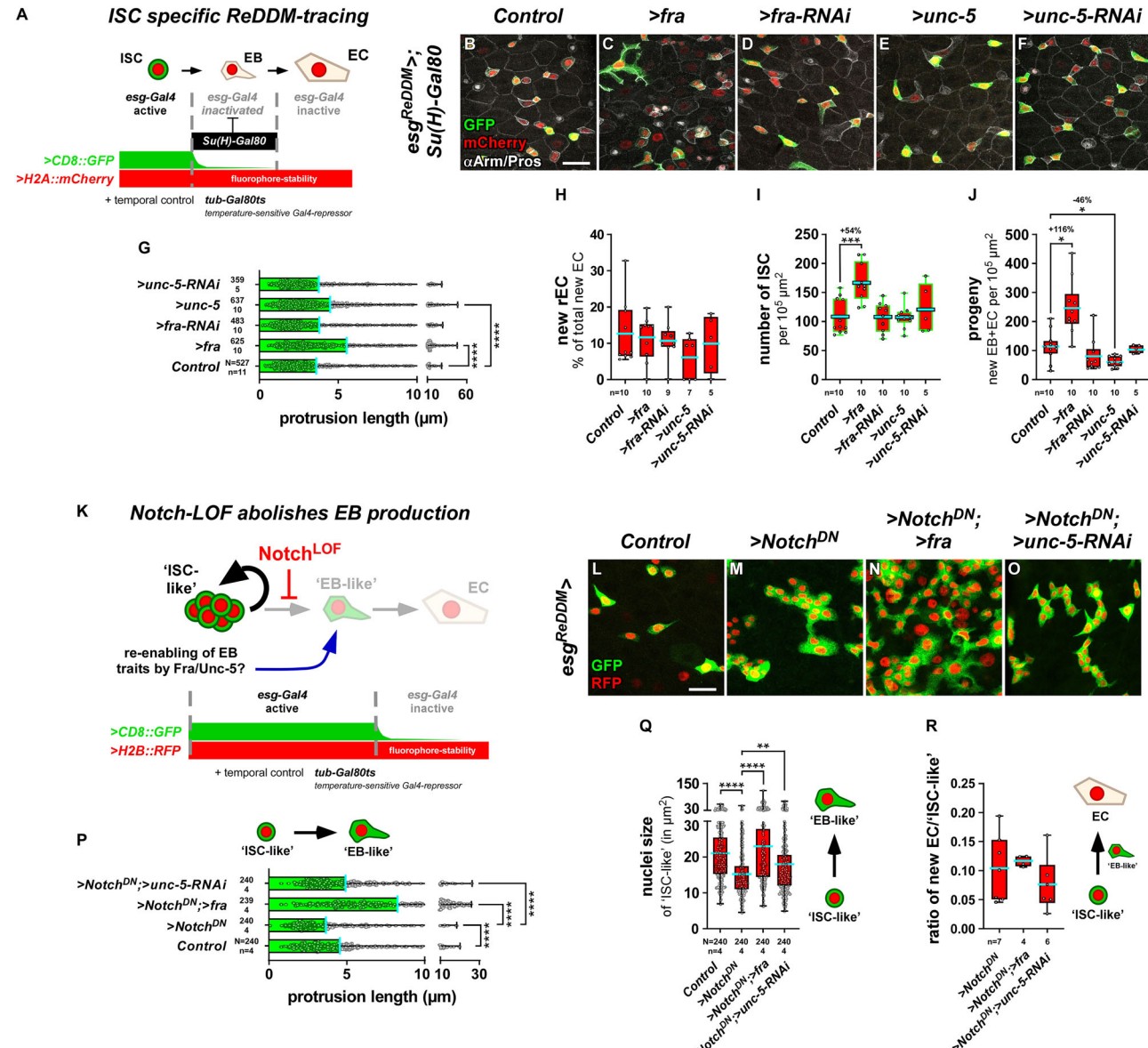

**Fig. 3 | Fra and Unc-5 induce EB traits and Notch-dependent EC differentiation.**
**A** Schematic illustration depicting ISC specific manipulation and ReDDM-tracing. ISC/EB specific *esg>* drives expression of *>CD8::GFP* with short half-life and *>H2B::RFP* with higher stability which persists in newly differentiated EC with inactive *esg >*. Temperature sensitive Gal80 (*tub-Gal80[s]*) ensures temporal expression control and Su(H)-Gal80 represses Gal4 activity in EB independent of temperature. Thus *esg>* is specifically active in ISC, allowing manipulation and ReDDM tracing of ISC. **B–F** Confocal images of midguts (R5) after 7 days of ISC-specific ReDDM-tracing and manipulations of Fra/DCC and Unc-5 receptors. Scale bar is 20 μm. Quantification of protrusion lengths (**G**, *p* = 0.4381 for *>fra-RNAi;p* = 0.3096 for *>unc-5-RNAi*), rEC renewal frequency (**H**, *p* = 0.6268;*p* = 0.9798;*p* = 0.1844;*p* = 0.9209), ISC numbers (**I**, *p* = 0.0007;0.9247;*p* = 0.979;*p* = 0.5384) and progeny encompassing new EB and EC (**J**, *p* = 0.0121;*p* = 0.1253;*p* = 0.0234;*p* = 0.9439) upon ISC specific manipulations of *fra* and *unc-5*. **K** Schematic illustration depicting the formation of 'ISC-like' tumours upon Notch-LOF induced blockade of 'EB-like' cells, and thereby EC

production. Fra/Unc-5 might induce 'ISC-like' to 'EB-like' transition despite Notch-LOF. Confocal images of *esg[ReDDM]*-tracing in controls (*w[1118]*) (**L**), with Notch-LOF tumours induced by *esg[ReDDM]*-driven expression of dominant negative Notch (*>Notch[DN]*, **M**) combined with expression of *>fra* (**N**) and *>unc-5-RNAi* (**O**). Scale bar is 20 μm. **P** Quantification of protrusion lengths (**P**) and nuclei size (**Q**, *p* = 0.0012 for *>N[DN];>unc-5-RNAi*) in 'ISC-like' cells. Longer protrusions (**P**) and larger nuclei (**Q**) upon additional expression of *>fra* and *>unc-5-RNAi* in Notch-LOF tumours suggest a transition to an 'EB-like' state. **R** Quantification of the ratio of new EC per 'ISC-like' cells in Notch-LOF tumours and combined with expression of *>fra* (*p* = 0.6801) and *>unc-5-RNAi* (*p* = 0.3121). An increased ratio would suggest a shift towards EC differentiation. **G, P** Bar plots show means and range indicated by whiskers. **H–J, Q–R** Box plots show 25th and 75th percentiles and whiskers indicate the range of values. **G–J, P–R** All data points are shown by dots and the means are indicated by light blues lines. 'n' are numbers of biological replicas and 'N' values describe numbers of technical replicas. Asterisks denote significances from comparisons by Kruskal Wallis tests (**p* < 0.05;*p* < 0.05;***p* < 0.001;****p* < 0.0001).

*esg > CD8::GFP*, into Netrin mutant background, we investigated protrusion formation in *NetA[-/-],NetB[-/+]* and *NetA[-/+],NetB[-/-]* adult midguts and found that homozygous lack of *NetB*, but not *NetA*, reduces protrusion length (Fig.5A−D).

Thus, we followed the hypothesis whether worn-out EC secrete NetB to attract EB, similar to previously described mitogen release

controlling homoeostatic ISC division[2]. Indeed, NetB is detected on singular not yet renewed EC with antibodies (RFP, Fig.5E) and two NetB enhancer traps reveal occasional *NetB* locus activation in EC of homeostatically kept flies (Fig. S6A-B). This relatively low number of NetB[+]- is coherent with the idea that only worn-out EC express NetB as under homoeostatic conditions only about 70 EC per week (10 EC/day)

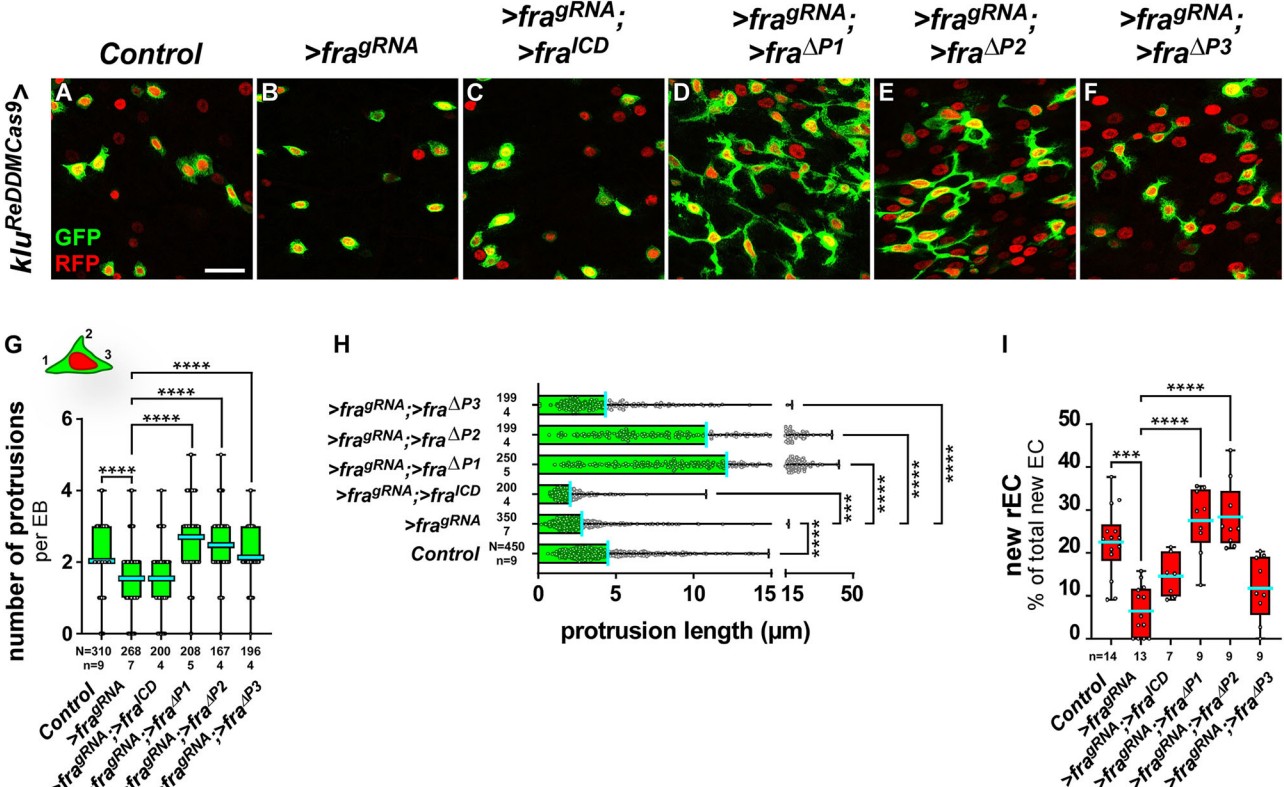

**Fig. 4 | Fra/DCC-induced EB migration depends on the P3 domain.** Confocal images of midguts (R5) after seven days of $klu^{ReDDM}$-tracing combined with expression of >$Cas9$ in controls (**A**), guideRNA (gRNA) mediated excision of $fra$ (>$fra^{gRNA}$, **B**), combined with expression of truncated Fra receptors with intracellular domain only (>$fra^{ICD}$, **C**), lacking the P1 (>$fra^{\Delta P1}$, **D**), P2 (>$fra^{\Delta P2}$, **E**) and P3 (>$fra^{\Delta P3}$, **F**) domain of the intracellular part. Scale bar is 20 µm. Quantification of EB protrusion number (**G**, $p = 0.8794$ for >$fra^{gRNA}$;>$fra^{ICD}$), EB protrusion lengths (**H**, $p = 0.0001$ for >$fra^{gRNA}$;>$fra^{ICD}$) and rEC renewal frequency (**I**, $p = 0.0001$ for >$fra^{gRNA}$;$p = 0.1507$ for

>$fra^{gRNA}$;>$fra^{ICD}$;$p = 0.352$ for >$fra^{gRNA}$;>$fra^{\Delta P3}$) in $klu^{ReDDMCas9}$-mediated excision of $fra$ combined with expression of truncated Fra receptors. **G, I** Box plots show 25th and 75th percentiles and whiskers indicate the range of values. **H** Bar plots show means and range indicated by whiskers. **G–I** All data points are shown by dots and the means are indicated by light blues lines. 'n' are numbers of biological replicas and 'N' values describe numbers of technical replicas. Asterisks denote significances from comparisons by Kruskal Wallis tests (***$p < 0.001$;****$p < 0.0001$).

need replacement (Fig. 2M)[9,10]. In line with this, stimulation of EC turnover with DSS and bacterial infection significantly increases NetB levels (Fig. 5F, G)[48]. In accordance with previous observations that blockade of EC apoptosis with baculovirus p35 reduces mitogen release[2], we detected reduced *NetB* transcription (Fig. 5H) and protrusion formation (Fig. 5J–K, Q–R) when EC death was blocked with >$p35$ using EC-specific Rapport (Fig. 5I)[49]. Briefly, Rapport allows 'ReDDM-like tracing' and manipulation by using dual expression systems. ReDDM-tracing is achieved by employing the lexA-Aop system with identical components to *esgReDDM*[9] and in parallel non-autonomous effects from manipulation of e.g. EC (Fig. 5I, EC-Rapport) is achieved by the Gal4/UAS system. Details on Rapport can be found in ref. 49.

In the contrary, EC death induces release of mitogens upon EC-specific loss of *Drosophila* E-cadherin[2]. Thus, we used EC-Rapport to drive >$E$-cadherin-RNAi to simulate worn-out EC. Supporting our hypothesis, we detected significantly longer and more numerous protrusions (Fig. 5M–N, Q–R) and an increase of NetB-levels in EC upon depletion of E-cadherin (Fig. 5S–U). In line with this, when EC-Rapport is used to deplete NetB from EC, protrusion length is significantly reduced (Fig. 5L, Q), which in turn strongly reduces the number of new rEC (Fig. 5V–X). Reciprocally, >$NetB$-expression using EC-specific Rapport, induces protrusion length and number (Fig. 5O–R) but does not allow conclusions to be drawn for protrusion directionality as NetB is released by the whole EC population. Therefore, we sought to design an experiment that unequivocally allows to detect whether protrusions point toward NetB-expressing EC.

To reveal whether NetB is indeed directing protrusion formation and EB migration, we needed a way to activate NetB in singular EC. For that purpose, we generated sporadic FLP-Out clones[50] in EC that express >$NetB$ with the Gal4/UAS system. Independently, ISC/EB membranes are labelled with *esg»CD8::GFP* (Fig. 6A) to assess whether ISC/EB extend protrusions towards NetB sources. Strikingly, ISC/EB within a 25 µm radius of >$NetB$ expressing EC clones (Fig. 6C) showed increased protrusion lengths (Fig. 6D) and numbers (Fig. 6E) compared with ISC/EB close to control clones (Fig. 6B). A defined NetB-source in single EC allowed us to analyse protrusion directionality and determine the angle in which protrusions point towards NetB releasing EC (Fig. 6F). An analysis of the angle between protrusions and clonal EC confirmed a directional extension of protrusions towards >$NetB$ expressing EC clones (Fig. 6G).

Apart from NetB release by EC, we also considered different sources for Netrins suggested by a previous publication[17]. Upon mechanical injury by laser ablation, Hu and colleagues found that the Ptk7-orthologue Otk (off-track) attracts ISC towards EE. To test whether EE-released NetA and NetB might play a role in progenitor migration during homoeostasis, we created a *Rab3-Gal4*-based 'EE-Rapport' (Fig. S6C-D) and investigated whether Netrins attract progenitors towards EE. Arguing against a role of Netrins released from EE, progenitors are not attracted by EE that express NetA (Fig. S6E,H) and NetB (Fig. S6F,H), whereas *otk*-expression in EE-Rapport stimulated gathering of progenitors around EE (Fig. S6G-H) as demonstrated during laser-induced gut injury[17]. Our findings and the data from Hu

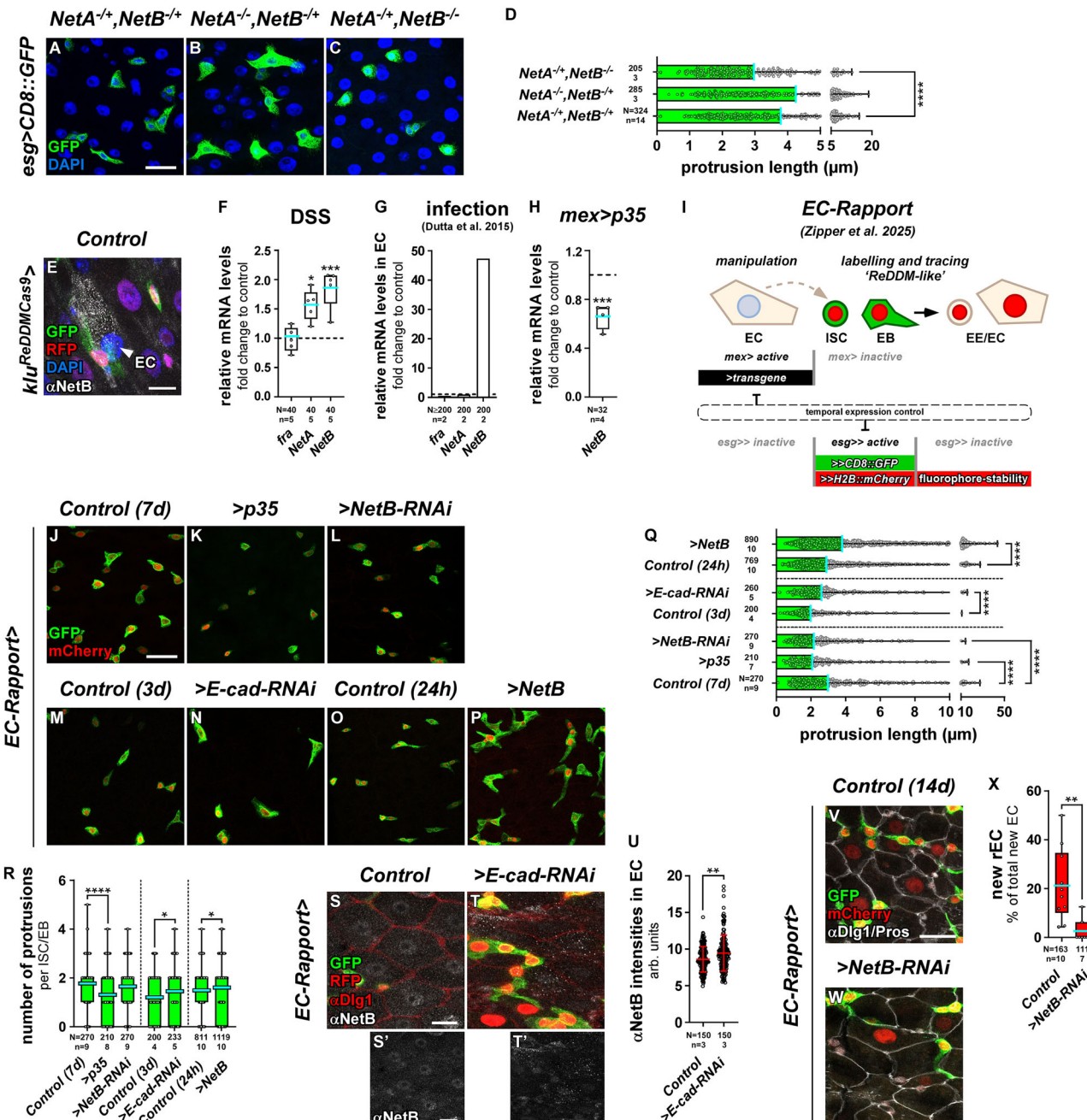

**Fig. 5 | EC-derived Netrin-B controls EB protrusion formation and attracts their migration.** Midguts (R5) with esg>CD8::GFP (ISC/EB) in Net^-/+,NetB^-/+ heterozygous mutant background (*Control*, **A**) and NetA^-/-,NetB^-/+ (**B**) and NetA^-/+,NetB^-/- (**C**) mutant backgrounds. Scale bar is 20 μm. **D** Quantification of ISC/EB protrusion lengths (p = 0.1295 for NetA^-/-,NetB^+/+) (**E**) Klu^ReDDMCas9 control (w^1118) showing an old EC (white arrowhead, RFP) positive for anti-NetB staining. Scale bar is 10 μm. Expression levels of *fra*, *NetA* and *NetB* measured by RT-qPCR in midgut-derived cDNA (**F**, **H**) after feeding of dioctyl sodium sulfosuccinate (DSS) (**F**, p = 0.8294;p = 0.0357;p = 0.0031), upon bacterial infection analyzed by RNA-Seq following FAC-sorting[48] (**G**), and upon expression of >p35 in EC (mex > p35) (**H**, p = 0.0006). **I** Schematic illustration of *EC-Rapport*-tracing combining EC-specific mex> with independent lexA/Aop-driven esg^lexReDDM-tracing. esg» drives expression of »CD8::GFP with short half-life and »H2B::mCherry with longer half-life persisting also in newly differentiated cells. Both expression systems are temporally controlled by Gal80^ts. Midguts (R5) after seven days (**J–L**), three days (**M, N**) and 24 h (**O, P**) of *EC-Rapport*-tracing in controls (w^1118) (**J, M, O**), with expression of >p35

(**K**), >NetB-RNAi (**L**), >E-cad-RNAi (**N**), and >NetB (**P**). Quantification of protrusion lengths (**Q**) and numbers (**R**, p = 0.2202 for >NetB-RNAi;p = 0.0216 for >E-cad-RNAi;p = 0.0273 for >NetB). (**S-T'**) Midguts (R5) after three days of *EC-Rapport*-tracing in controls (w^1118) (**S-S'**) and upon expression of >E-cad-RNAi (**T-T'**) with antibody staining targeting NetB and Dlg1. Scale bars are 20 μm. **U** Quantification of NetB intensities in EC (p = 0.0073). Midguts (R5) after 14 days of *EC-Rapport*-tracing in controls (w^1118) (**V**) and upon knockdown of *NetB* (**W**). Scale bar is 20 μm. **X** Quantification of rEC renewal frequency (p = 0.0021). **F**, **H**, **R**, **X** Box plots show 25th and 75th percentiles and whiskers indicate the range of values. **D**, **Q** Bar plots show means and range indicated by whiskers. **U** Scatter dot plots show all data points with indication of means and standard deviations. All data points are shown by dots and means are indicated by light blues lines. 'n' are numbers of biological replicas and 'N' values describe numbers of technical replicas. Asterisks denote significances from multiple comparisons by Kruskal Wallis tests (**D**, **F**, **Q**, **R**) and two-sided Mann–Whitney tests (**H**, **Q**, **R**, **U**, **X**) (*p < 0.05; **p < 0.01; ***p < 0.001; ****p < 0.0001).

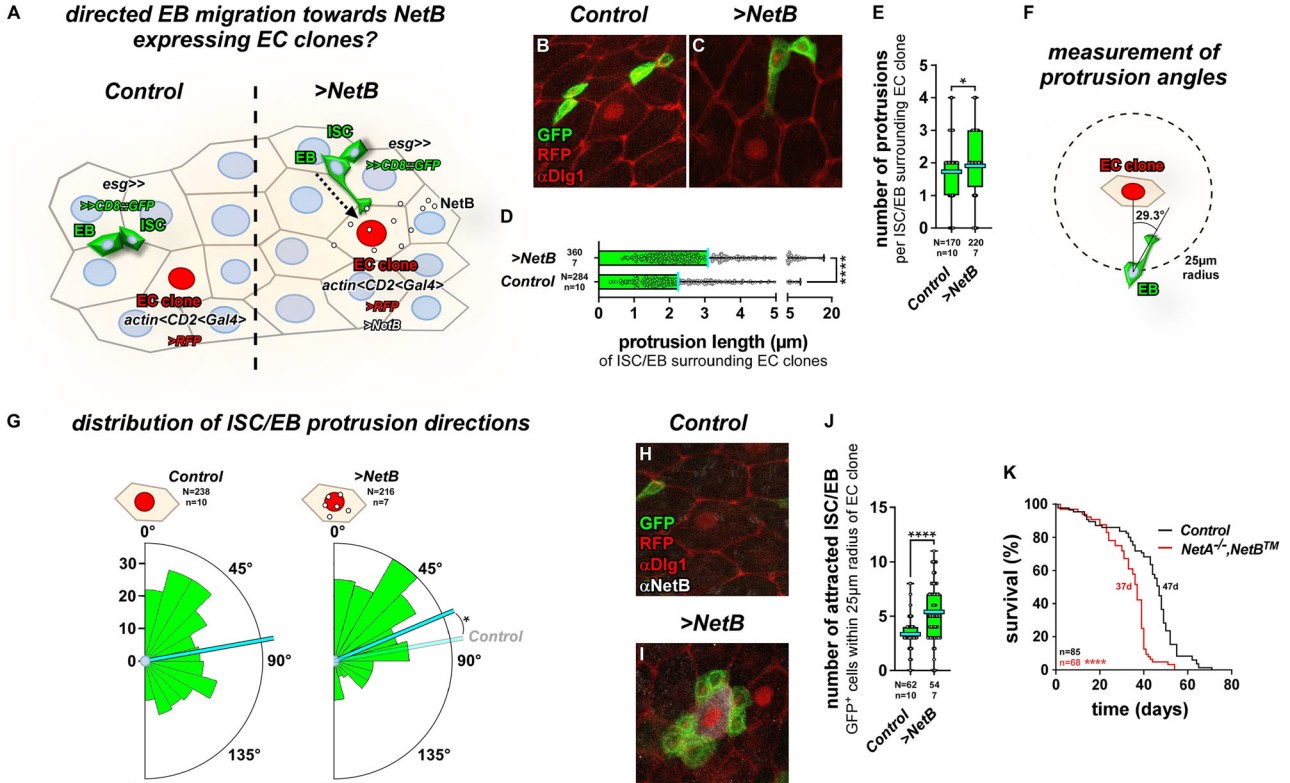

**Fig. 6 | EC-derived Netrin-B induces directed EB migration. A** Schematic illustration depicting RFP-labelled EC clones induced by flippase mediated excision of a CD2 cassette (<CD2<) inserted between an actin-enhancer and Gal4. Heatshock induced removal of the CD2 cassette allows Gal4 to translocate to the nucleus and activate expression of UAS(>)-transgenes. In addition to >RFP, expression of >NetB can be induced in clones. Independent of the FLP-Out clones all ISC/EB are labelled by esg»CD8::GFP. Confocal images of midguts (R5) with heatshock-induced FLP-Out clones in controls ($w^{1118}$) (**B**) and combined with clone specific expression of >NetB (**C**). Antibody staining targeting Dlg1 labels EC. Quantification of protrusion lengths (**D**) and numbers (**E**, $p = 0.046$) of CD8::GFP labelled ISC/EB located within a 25 μm radius around RFP-labelled EC clones with expression of >NetB compared to control ($w^{1118}$) clones. **F** Cartoon depicting angular measurements of EB protrusion directionality towards EC clones. **G** Polar histograms depicting the direction of ISC/EB protrusions measured as angles related to >NetB expressing EC clones compared to control ($w^{1118}$) clones ($p = 0.0109$). Confocal images of midguts (R5) with

heatshock-induced FLP-Out clones in controls ($w^{1118}$) (**H**) and combined with clone specific expression of >NetB (**I**) with antibody staining for NetB and the EC-marker Dlg1. **J** Quantification of ISC/EB numbers within 25 μm radius of >NetB expressing clones compared to control ($w^{1118}$) clones. **K** Kaplan-Meier estimation of survival in $w^{1118}$ controls compared to flies homozygous mutant for NetA and with membrane tethered NetB ($NetA^{-/-},NetB^{TM}$) thus unable to secrete NetB. Mean survival time is shown in days (d). **E, J** Box plots show 25th and 75th percentiles, and whiskers indicate the range of values. **D** Bar plots show means and range indicated by whiskers. **G** Polar histograms show angles with indication of frequency. Means are indicated by light blue lines. All data points are shown by dots and means indicated by light blues lines. 'n' are numbers of biological replicas and 'N' values describe numbers of technical replicas. Asterisks denote significances from multiple comparisons by two-sided Mann–Whitney tests (**D, E, G**) and Long-rank test (**K**) (*$p < 0.05$; ****$p < 0.0001$).

and colleagues point to different control of protrusion formation on EB and ISC and that protrusions are induced differently in homoeostasis and injury respectively. Supporting two different mechanisms for protrusion formation at work in homoeostasis versus injury, mechanical injury (Fig. S7A) stimulates protrusion formation in controls (Fig. S7B-B') as well as >fra-RNAi (Fig. S7C-C'). In summary, this suggests that protrusions formed upon injury form independent of Fra. In contrast and in line with previous observations generally implying Trio in protrusion formation upon injury[17], >trio-RNAi prevents protrusion formation upon mechanical injury as well (Fig. S7D-D')[9,10].

To assess a function of Netrins in adult flies, we investigated the survival of constitutive $NetA^{-/-},NetB^{TM}$ mutants that are null mutant for NetA and cannot secrete NetB[47]. Interestingly, $NetA^{-/-},NetB^{TM}$ mutant flies develop into imago without obvious neurodevelopmental defects. Although not providing a direct involvement of Netrins in adult midgut homoeostasis, or any other degenerative process, $NetA^{-/-},NetB^{TM}$ mutants die significantly earlier than controls (Fig.6K) and reveal a requirement for Netrins in adult flies. Together, our data suggests that spatially directed EB migration towards dying EC acts directly after

previously described one-by-one EC replaced through mitogen-induced ISC proliferation[2]. Having established a connection between dying EC, NetB release and EB protrusion formation (Fig.6J), we aimed to detail on the actual range of NetB signalling to Fra/DCC-receptors on EB migration based on our observation that progenitors accumulated around >NetB expressing EC clones (Fig.6H-J). This data reveals spatial protrusion orientation (Fig.6G), protrusion attraction towards NetB-expressing EC (Fig.6C) and subsequent directed migration (Fig.6I) and raises the question how far Netrins are actually able to signal. To further assess whether and how far Netrins are attracting midgut progenitors, we developed the Hamelin assay.

## 'Hamelin' proves long range attraction of midgut progenitors towards Netrin sources

Finally, we assessed whether Netrins attract midgut progenitors over long distances. For that purpose, we designed the 'Hamelin' assay in remembrance of Grimm's pied piper and his magic pipe with which he lured a rat plague into a nearby river. Hamelin allows the in vivo assessment of genetically directed migration and is technically similar to Rapport[49]: we employed two independent binary expressions

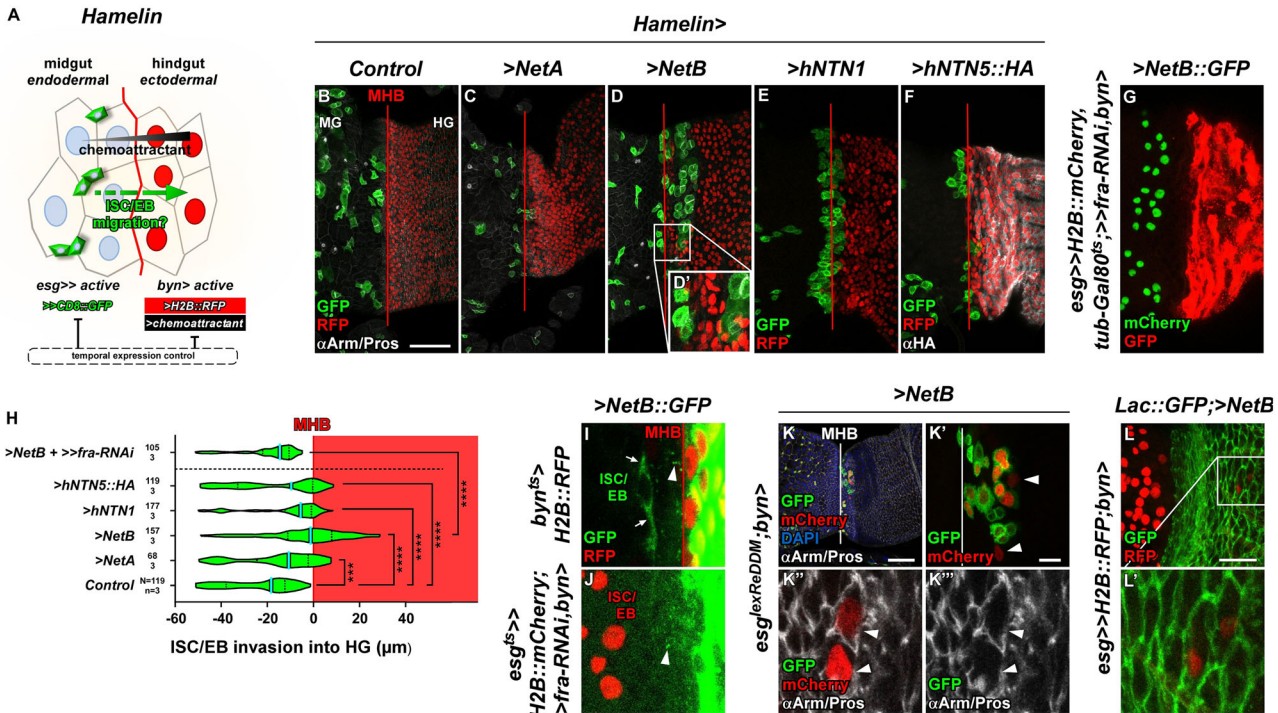

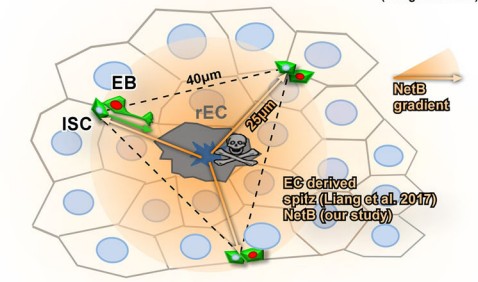

**Fig. 7 | 'Hamelin' proves long range attraction of midgut progenitors towards Netrin sources. A** Hamelin assay with the midgut-hindgut boundary (MHB, red line) separating midgut (endodermal origin) and hindgut (ectodermal origin). Midgut progenitor cells are labelled by *esg»CD8::GFP*, and hindgut specific *byn>* is used for expression of *>H2B::RFP* and chemoattractants to test their potential to lure GFP⁺-midgut progenitors across the MHB. Hamelin assay in controls (*w^1118*) (**B**) and combined with expression of *Drosophila >NetA* (**C**) and *>NetB* (**D**), human Netrin-1 (*>hNTN1*, **E**) and Netrin-5 (*>hNTN5*, **F**). MHB are identified by *byn > H2B::RFP* signal (red lines). Scale bar is 40 μm. **G** Midgut (R5) with *esg»H2B::mCherry* and *»fra-RNAi* labelling and manipulating midgut progenitor cells. Additionally, *byn>* enables expression of *>NetB::GFP* in hindgut cells. **H** Quantification of ISC/EB migration distances upon hindgut specific expression of Netrins and combined with *»fra-RNAi*. Violin blots show 25th and 75th percentiles indicated by dotted lines and means indicated by light blue lines. 'n' are numbers of biological replicas and 'N' values describe numbers of technical replicas. Asterisks denote significances from multiple comparisons by Kruskal Wallis test (***$p < 0.001$;****$p < 0.0001$). **I** Hindgut

cells labelled by *byn > H2B::RFP* and expressing *>NetB::GFP*. The MHB is identified by *byn > H2B::RFP* (red line). **J** Hindgut specific expression of NetB::GFP combined with *esg»fra-RNAi*. **I, J** GFP positive dots anterior to the MHB (white arrowhead) suggest secretion of NetB::GFP which is bound by ISC/EB (**I**, white arrows). **K-K'''** lexA/Aop-driven *ReDDM*-tracing (as in *EC-Rapport*) combined with hindgut-specific *byn>* to allow expression of *>NetB* in hindgut cells and independent tracing of midgut progenitors. **L-L'** *esg»H2B::mCherry* labelling midgut ISC/EB combined with Lac::GFP labelling pleated septate junctions thus serving as a hindgut marker. mCherry tracing suggest integration of midgut derived progenitor cells into hindgut epithelium and show formation of new septate junctions visualized by Lac::GFP. **M** Schematic illustration showing midgut epithelium with a dying rEC located in between three ISC/EB duplets with an equidistant spacing of 40 μm (Fig. S7E–F). The dying rEC secretes NetB which forms a gradient of 25 μm reaching ISC/EB-duplets ensuring protrusion formation and directed EB migration (Fig.6) similar to previously described secretion of EGF-ligands ensuring ISC mitosis within a similar radius[2].

systems and apply them for the investigation of migration and homing behaviour by the capacity of manipulating both, receptor and ligand. Therefore, we combined established components for spatiotemporal control of ectopic Netrin expression with the Gal4-UAS expression system to attract differentially labelled progenitors using a second expression system (lexA-Aop, Fig.7A). As well-defined remote expression site, we chose the established *byn-Gal4* driver that is active in a ring of ectodermal hindgut boundary cells at the junction with the

endodermal PMG[51]. Anterior to the midgut-hindgut boundary (MHB) endodermal midgut progenitors are labelled with *esg-lexA* driven *»CD8::GFP* and temperature-sensitive Gal80^ts adds temporal control for lexA and Gal4-activity (Fig.7A)[52].

Beseechingly, *byn*-driven ectopic expression of both, *>NetA* and *>NetB*, attracts midgut progenitors not only towards, but also clearly across the MHB (Fig.7C, D, H) into ectodermal hindgut territory[51] after seven days. Crossing the strict MHB between organs of two different

**Table 1 | List of primer pairs used for expression analyses of *fra*, *NetA*, *NetB* and *unc-5***

| Primer | Forward (5'-3') | Reverse (5'-3') |
|--------|-----------------|-----------------|
| *Rp49* | TGGTTTCCGGCAAGCTTCAA | TGTTGTCGATACCCTTGGGC |
| *fra* | GCCATCACAACAAACAGAAGC | CTCTTGCCTCCACGTCCAAT |
| *NetA* | GCAGCGATATTGCGAGTACCA | AAGGTTACGTTGTCCCCACTT |
| *NetB* | TATCCTCCAACGATCCCTGCT | AGTTCGCAGTAACGCTCCG |
| *unc-5* | GATGCTATTGACCCACTCACG | GCTGCGGTCTCACATTCTCAT |

germ layers (midgut is endodermal, whereas hindgut is ectodermal) is considered an invasive cell behaviour[53]. Invasion is depicted more clearly in the inset in Fig.7D', which shows *esg*⁺-cells well beyond the MHB on *byn > H2B::RFP*⁺ area. Equally important, invasion of midgut progenitors into hindgut tissue is not observed in controls (Fig.7B, H). Underlining a high degree of conservation, human Netrin-1 (hNTN1) and Netrin-5 (hNTN5) phenocopy *Drosophila* Netrins in the attraction of midgut progenitors (Fig.7E–H), which underlines that Netrin ligands as well as Fra/DCC receptors are functionally interchangeable between homologous fly and human genes (Fig.2G, Fig.7E, F). Strikingly, using the second expression system in Hamelin to deplete Fra in midgut progenitors with *»fra-RNAi*, we provide first evidence that NetB-attraction acts through Fra (Fig.7G, H). In line with previous studies of nervous system development, Hamelin shows that Netrins act as long-range attraction signals in intestinal tissue of different germ layer origin.

Measurements of migratory distances suggested that midgut progenitors are attracted from up to 40 μm distance to the MHB territory by all four Netrins (Fig.7C–F, H). Intriguingly, the average distance of normally scattered ISC/EB duplets in the PMG is around 40 μm (Fig. S7E-F), thus enabling Netrins to potentially attract EB to any given position in the vicinity of a worn-out EC. Netrins are thought to form chemotactile gradients serving directed migration[54]. To visualize such gradient, we expressed >*NetB::GFP*[55] from the hindgut using *byn >*. Interestingly, NetB::GFP puncta are found anterior to the MHB (Fig.7I, arrowhead). Additionally, NetB::GFP puncta are abundant on progenitor membranes across the MHB (Fig.7I, arrows) suggesting their exogenous offspring which is corroborating a weak NetB signal that we also occasionally detected on progenitor membranes (Fig.5S'-T'). This observation is supported by data from neuronal growth cones, where extracellular Netrins were shown to cluster and stabilize DCC receptors[56]. In line with this idea, the membrane accumulation of NetB::GFP is gone upon depletion of Fra (Fig.7J), which suggests that exogenous EC-derived NetB might play a role in protrusion stabilization on intestinal progenitors as well (Fig.5Q, R).

Intriguingly, our initial Hamelin images with Netrin-attracted endodermal midgut progenitors suggested that progenitors might even form new junctions inside the ectodermal hindgut epithelium. Formation of new junctions might suggest that midgut progenitors occasionally perform MET (Mesenchymal to Epithelial Transition) in hindgut tissue (Fig.7K-K'''). During midgut homoeostasis, MET is indicated by the loss of *esg* and subsequent terminal differentiation of EB into EC[9,10,18,57]. To analyse whether midgut progenitors integrate into the hindgut epithelium, we combined the *esg*^*lexReDDM* cassette[49] (Fig.5I) with *byn-Gal4* and investigated whether NetB-attracted progenitors perform MET in hindgut tissue indicated by the loss of *esg*-activity (*esg»CD8::GFP,H2A::mCherry*⁺)[9,10]. Indeed, we found traced cells that lost activation of *esg»* indicated by the loss of *»GFP*-signal (Fig.7K-K'''). These RFP-traced cells additionally formed new adherence junctions indicated by Arm (Fig.7K'', K'''). Importantly, EC of the endodermal PMG are connected with smooth septate junctions (such as Dlg1, Fig.5V), whereas the hindgut employs pleated septate junctions, which can be labelled by Lachesin::GFP (Lac::GFP)[58–60]. When we used

Lac::GFP to investigate whether PMG progenitors attracted by NetB are able to switch to pleated septate junctions, we found indeed traced cells that performed MET and integrated into the Lac::GFP area (Fig.7L-L'). This remarkable plasticity of endodermal ISC will be a fascinating topic for future studies.

Our findings support the idea that midgut progenitors migrate towards Netrin sources guided by Fra-receptors. Interestingly, the range of Netrin B signalling covers the necessary distances between EB and rEC (Fig.7M) ensuring their spatial renewal. Taken together, our data proofs the capacity of Netrins to spatially organize timely EB migration and integration with similar metrics as described for the stimulation of ISC mitosis from worn-out EC[2].

## Discussion

In the two decades since the discovery of ISC in the adult *Drosophila* gut, seminal studies discovered how ISC behaviour ensures intestinal homoeostasis by constantly replenishing worn-out epithelial cells[3]. Only recently, new studies started shedding light on the immediate ISC daughter cells called enteroblasts[9,12,15,61,62]. Here, we identify and characterize a conserved guidance cue that directs EB migration to replace single, worn-out enterocytes that is essential for intestinal homoeostasis. Lack of Fra-mediated EB migration and integration disrupts midgut homoeostasis and results in fly demise. Our discovered mechanism of spatially directed replacement of remote epithelial cells closes a gap of mechanistical knowledge in intestinal homoeostatic turnover and might as well play a role in further epithelia (Fig.7M). Additionally, this mechanism seems highly conserved as Netrin-ligands and Fra and DCC receptors are interchangeable, which is shown in our functional *klu*^*ReDDM* and Hamelin assays.

### Netrin B, Fra/DCC and Unc-5 control directionality of enteroblast protrusion formation and migration

In our EB-specific manipulation of Fra/DCC and Unc-5 receptors, we uncovered antagonistic functions between Fra/DCC and Unc-5 receptors on EB protrusion formation and migration during homoeostatic midgut turnover, which is consistent with previous data describing their opposing roles in axon pathfinding in great detail[26]. Our findings, especially when Fra/DCC and Unc-5 receptors are overexpressed and depleted at the same time (Fig. S3A–G), support a model in which activity of Fra/DCC mediates the attractive response that drives EB protrusion formation and migration. However, it needs to be kept in mind that our analyses are based on static 'snapshots' and thus reflect probabilities of whether a protrusion is pointing to an EC (Fig.2I–K). We speculate that EB protrusions are dynamic entities and are regularly retracted depending on detected EC-specific NetB-levels. This is in line with our observation that more protrusions point towards old EC that bear a higher probability for need of replacement and thus a higher probability to be NetB-positive (Fig.2I–K, Fig.5). This is corroborated in our genetic manipulations of ligand and receptor levels in which changes in Fra/DCC and Unc-5 receptor levels and NetB-expression affect protrusion directionality towards EC (Fig.2K, Fig.6G). Our data is further supported by findings from axonal guidance studies and suggests that the detection of NetB by Fra-homodimers stabilizes EB protrusions and subsequently, EC renewal takes place. Of important note here, our quantitative assessment of rEC numbers over a large timespan using ReDDM reveals the true frequency of coordinated EB migration and EC renewal independent from protrusion dynamics.

During axon guidance, it was shown that interaction of Fra/DCC and Unc-5 receptors can take place as homodimers (Fra/Fra, Unc-5/Unc-5) as well as heterodimers (Fra/Unc-5), which is also conserved for mammalian DCC[24]. Our observation that Unc-5 is internalized into vesicles, even when overexpressed (Fig. S1I'''-I''''), argues for a mechanism that allows Fra/DCC to signal as homodimer. Indeed, it is commonly accepted that internalization of Unc-5 acts as a switch from repulsion to attraction allowing Fra and DCC homodimers[24,27,63].

Similarly, Unc-5-depletion shows EB protrusion elongation and proper migration, which provides additional evidence that the absence of Fra/Unc-5 heterodimers allows EB protrusion formation through Fra homodimers. In line with this, localization of Fra and DCC at the EB membrane (Fig. S1I-I″,J-J‴) even when Unc-5 is overexpressed (Fig. S1I−I‴) provides additional evidence for this mechanistic conservation. However, it cannot be entirely ruled out that Fra/DCC and Unc-5 play a role in ISC as well, as *unc-5* overexpression reduces progeny (Fig.3J). This observation either suggests a role for Unc-5 homodimers in ISC, or the presence of Fra-Unc-5 heterodimers stemming from Fra-receptor expression below the detection limit of our confocal microscopes that then act similarly as observed in EB (Fig.2M). Homodimers of Unc-5 are well known in cancer research and involved in the control of cell survival, migration, and differentiation[64,65]. However, as depletions of Fra and Unc-5, do not affect protrusion length, new rEC, number of ISC and progeny (Fig.3G-J), we concluded no major function of both receptors in ISC. Intriguingly, only homodimers of Fra/DCC were shown to connect to the F-Actin network allowing protrusion formation[25,28], which is in line with our datasets on Arpc3, Ena and Trio (Fig. S4+5).

On the ligand side, binding of NetB induces either Fra/DCC homo- or heterodimers with Unc-5 and in turn allows attractive and repulsive signalling respectively[24]. A long-standing question in the neurodevelopmental field investigated the range over which Netrin-1 can be detected by axons[23]. In the midgut these ranges are rather small and ISC /EB duplets are scattered with a distance of around 40 µm in R5 along the PMG. In EC-clones (Fig.6) and Hamelin (Fig.7), NetB induced migration in a range of 25 µm and 40 µm respectively. From a mathematical point of view, this finding allows the speculation that NetB⁺-EC need to signal around 20 µm, which in turn means that they not only reach one ISC/EB duplet but several (Fig.7M). In line with this model, we find accumulation of progenitors around NetB⁺-EC in our clonal experiments (Fig.6J). A topic that definitely warrants future investigation is the onset of NetB-expression in worn-out EC (Fig.5S−T). One possible mechanism is that E-cadherin depletion releases not only α-catenin[2] but also β-catenin/arm from adherence junctions of worn-out EC into the nucleus[66,67]. β-catenin was shown to activate NF-κB during intestinal inflammation[66,68] and NF-κB in turn was previously shown to activate Netrin expression in colonic crypts[69,70] suggesting a similar activation in fly EC.

Taken together, our mechanism of NetB coordinating spatially directed EB integration adds to mechanistical knowledge in intestinal homoeostatic turnover with similar metrics previously described for mitogens acting on ISC proliferation (Fig.7M)[2]. As observed for the control of ISC mitosis, our findings suggest a one-by-one mechanism of cell renewal upon directed EB migration.

### Implications of Fra/DCC-directed protrusions and migration for mammalian homoeostasis

In addition to the functional and genetic conservation of Fra/DCC signalling, there is also mechanistic consideration for active cellular migration in the mammalian crypts that was only recently suggested by functional experiments. For decades the predominant theory prevailed that cells migrate in a passive manner driven by mitotic pressure from the ISC compartment[71,72]. Although constantly challenged by experimental studies that e.g. blocked ISC proliferation[73,74], only recently active migration was suggested in functional experiments[75,76]. Krndija et al. investigated crypt-specific genetic and pharmacological disruption of the Arp2/3 complex and detected reduced cell migration in villi. In accordance with our data showing that Arpc3B-depletion reduces EB protrusion length and number (Fig. S5A−D, M-N), basal protrusions of migrating cells in villi project towards the intestinal lumen and confer a front-back polarity to migrating cells until extruded at the tip of villi[75]. Following this study, Azkanaz and colleagues describe that Lgr5⁺-small cells are even capable to perform retrograde

movement towards the crypt against the stream of cells following the conveyor belt-like upward movement. These retrogradely moving cells are able to repopulate the crypt upon depletion of Lgr5⁺ cells.

It is tempting to speculate whether Netrins and Fra/DCC- and UNC5-signalling might play a role in directing these retrograde progenitor movements towards the tip of the crypt. Netrin-1 is found abundantly in the tip of the crypts with a declining gradient towards the luminal shedding sites[77], which led to the hypothesis of DCC and UNC5 as dependence receptors for EC survival. Our data on Netrin-DCC/UNC5 coordinating spatial migration of midgut EB and studies of nervous system development additionally imply Netrin-1, UNC-5 and Fra/DCC in migration and axonal pathfinding. In the light of our findings, it is thus tempting to speculate whether another role for Netrin-1, UNC-5 and Fra/DCC is an involvement in migratory properties of intestinal cells (Fig.7E, H). Indeed, mice with a point mutation in DCC (*DCC^{D1290N}*) that abolishes pro-apoptotic signalling, present a 2.5-fold higher adenocarcinoma incidence. Interestingly, the same authors observed a five-fold increase in serosal invasion and cells metastasizing primarily to the liver, but did not further investigate an impact of Fra/DCC on migratory properties[78]. Similar observations have been made in mice overexpressing Netrin-1[77] in which crypts display increased invasion from adenocarcinoma as well, further supporting the idea that apoptosis and induction of migration might be uncoupled.

### Implication of Fra/DCC in EMT and metastasis

Initiation of migratory behaviour during development and from solid tumours requires acquisition of mesenchymal traits through EMT (Epithelial-to-Mesenchymal-Transition) and its reverse process MET when homing[79]. Regulation of EMT and MET is complex and intermediate plastic cell states are governed by dozens of EMT-inducers like snail and non-coding factors such as microRNAs[80], which could explain why specific genetic mutations causative for EMT and metastatic disease remain yet to be discovered[81]. Interestingly, acquisition of EMT-traits and dissemination events are observed in early preneoplastic lesions of various epithelial tumours[82] including CRC[83–85]. Supporting the idea that non-mutational events induce migratory behaviour, recent observations establish active migration as part of mammalian intestinal homoeostasis[75,76]. Currently discussed factors for EMT and MET involve an influence from the tumour and homing microenvironment[86–89] following the 'seed and soil' hypothesis[90].

In a fly nutshell of 'seed and soil', our 'Hamelin' assay reveals attraction and homing of endodermal progenitors by Netrins into the substantially different heterophilic environment of the ectodermal hindgut[91] reflecting an immense plasticity of midgut progenitors. Fly midgut progenitors are inherently mesenchymal and their EMT/MET processes involve conserved frameworks such as microRNA-8/200 regulating snail-like EMT-inducers like *esg* and *zfh1*[9,10,17]. Interestingly, NetB and Frazzled were previously shown to promote MET during *Drosophila* embryogenesis[92]. Thus our data not only establishes a role for NetB-Frazzled/DCC-signalling in the spatial coordination of EB migration towards worn-out rEC, but also hints to a role during intestinal homoeostasis for renewal of aEC and rEC through MET. Even though aEC and rEC renewal is affected by Fra-LOF (Fig.2L, M, S2N), Fra-signalling alone does not suffice for MET and depends on Notch-signalling activity (Fig.2M–O, R) suggesting Notch-dependent regulation of at least parts of NetB and Fra/DCC-signalling in the intestinal lineage as well[93–95].

In line with a role in EMT/MET processes, endometrial cancer patients treated with antibodies blocking Netrin-1 (NP137) show induction of cell death and inhibition of EMT features accompanied with an overall more epithelial phenotype in tumour biopsies[96,97]. Further underlining the clinical potential of studies focussing on axon guidance cues, the axon guidance receptor PlexinB2 promotes initial seeding of CRC metastases into liver parenchyma ahead of growth promoting niche formation[98]. Adding to these intriguing findings, our

study shows that the invasive cells do not necessarily need a complex mutation landscape, but can be rather singular and genetically wild-type midgut progenitors that home in ectodermal tissue and acquire epithelial traits by sole expression of Netrins (Fig.7K'''-L'). The molecular adaptations in addition to EMT and MET processes needed for such plasticity, will be a thrilling topic for future research for the basic and applied understanding of homing processes including the targeting of Netrin-DCC/Unc-5 signalling in tumour EMT and metastasis.

Together, our data establishes spatially directed EB migration to be essential for intestinal homoeostasis. The identified factors, NetB-Frazzled/DCC and Unc-5, are highly disease relevant and conserved, which warrants future investigation in mammalian models.

# Methods

## Genetics and fly husbandry/fly strains
The following transgenic fly stocks were used: *w^1118^* serving as genetic background control throughout the manuscript, *esg^ReDDM9^*, *esg^ReDDMCas99,99^*, *klu^ReDDM11^*, *Gbe + Su(H)-dsRed* (T. Klein), *UAS-fra, UAS-unc-5::HA, UAS-NetA, UAS-NetB* (C. Goodman), *Su(H)-Gal80^35^,UAS-fra::myc, UAS-fra^ΔP1^::myc, UAS-fra^ΔP2^::myc, UAS-fra^ΔP3^::myc, UAS-fra^Δ44^, UAS-fra^ICD100^, NetA^-/+^, NetB^-/+^, NetA^-/-^, NetB^-/-^, NetA^-/-^,NetB^TM^* (C. Goodman), *hsFlp;;actin < CD2<Gal4,UAS-RFP* (A. Wodarz), *byn-Gal4^101^, UAS-NetB::GFP^55^, klu-Gal4^12^, Rab3-Gal4* (L. Jones), *UAS-Otk^102^, UAS-N^DN^* (J. Treisman), *13xLexAop2-H2B::mCherry::HA^103^, Lac::GFP* (C. Klämbt).

From Bloomington Drosophila Stock Centre (BDSC): *UAS-fra-RNAi* (BL31496, BL31664, BL40826), *UAS-unc-5-RNAi* (BL33756), *UAS-p35* (BL5072), *UAS-E-cad-RNAi* (BL32904), *esg-lexA* (BL66632), *13xLexAop2-mCD8::GFP* (BL32205), *UAS-hNTN1* (BL78495), *I-KCKT-Gal4* (BL91419), *UAS-trio-RNAi* (BL43549), *FRTG13,fra^3^* (BL8813), *FRTG13,fra^4^* (BL8743), *NetB-lexA* (BL54725), *U6-EGFP^gRNA^* (BL79393), *UAS-Arpc3B-RNAi* (BL27528), *UAS-ena-RNAi* (BL31582, BL39034), *hs-Flp;FRTG13,UAS-CD8::GFP* (BL5131), *nos-phiC31;;attP86Fb* (BL24749)

From Kyoto Stock Centre (DGGR): *UAS-hDCC* (303875), *UAS-hNTN5::HA* (304368), *fra::GFP* (115148), *NetB-Gal4* (113850).

From Vienna Drosophila Resource Centre (VDRC): *UAS-fra^gRNA^* (v342231, v342369), *UAS-NetB-RNAi* (SH330183), *unc-5::GFP* (v318261), *UAS-Arpc3-RNAi* (GD47207, KK105278), *UAS-ena-RNAi* (GD43056, GD43058, KK106484), *UAS-trio-RNAi* (GD40137, GD40138).

## Food composition and fly keeping
Fly food contained 1424 g corn meal, 900 g malt extract, 800 g sugar beet syrup, 336 g dried yeast, 190 g soy flour, 100 g agarose in 20 l H$_2$O. The ingredients were mixed and cooked for about one hour to reduce bioburden. After cooling down the food 90 ml propionic acid and 30 g NIPAGIN (antimycotic agent) were added, and the food was filled in small plastic vials plugged with foam. Flies were kept at 25 °C. Crosses containing a temperature sensitive Gal80^ts^ repressor were kept at 18 °C to repress Gal4 activity during development and shifted to 29 °C to start transgene expression in adult flies.

## Rapport tracing
For the investigation of paracrine effects on stem cell behaviour shown in Fig.5I-XS, Fig.7K-K''' and Fig. S6C–H, we employed 'Rapport' ('Repressible activity paracrine reporter'), a dual binary expression system that combines spatiotemporally controlled transgene expression with ReDDM tracing of stem cell progeny.

Briefly, with the design of Rapport, we aimed to preserve the advantages of the existing established Gal4/UAS drivers and toolbox. Therefore, we created an entirely new and Gal4-independent lexA/Aop-based tracing system 'esg^lexReDDM^' (*esg»CD8::GFP, »H2B::mCherry::HA, tub-Gal80^ts^*). Rapport combines established elements with new state of the dual binary expression control. Importantly, the *esg^lexReDDM^*-transgenic (*esg»CD8::GFP, »H2B::mCherry::HA, tub-Gal80^ts^*) is a quadruple recombinant on the second chromosome and thus allows

straightforward generation of toolkit-stocks with any Gal4-driver on the third chromosome. Importantly, when combined with *mex-Gal4* (Fig.5I-X, *mex >* ) for EC specific expression[104], with *byn-Gal4* (Fig.7K-K''') for hindgut specific expression and with *Rab3-Gal4* (Fig. S6C-H) for EE specific expression, the lexA-operator driven in *esg^lexReDDM^* as well as Gal4 driven by *mex* and *Rab3* are repressed by temperature-sensitive Gal80^ts^, which allows simultaneous temporally controlled onset of UAS-transgenes as well as *esg^lexReDDM^* tracing and Aop-transgenes[52] established in ref. 49.

## FLP-Out clones for clonal expression of >*NetB*
Flp-Out clones[50] were used to enable the analysis of EB protrusion orientation and migration towards single EC clones labelled by >*RFP* and ectopically expressing >*NetB* (Fig.6). Clones were induced by 10 min heatshock at 37 °C followed by 24 h shift at 29 °C to allow expression of >*NetB* in the presence of *tub-Gal80^ts^*. Dissections and antibody staining were performed as described in the Immunohistochemistry section. Protrusion numbers, lengths and directions as well as ISC/EB numbers were analysed within a 25 μm radius around an EC clone.

## Hamelin
To investigate whether Netrins are secreted and are able to attract intestinal progenitors over long distances (Fig.7) we developed 'Hamelin' (Hindgut attracted migration evaluation – luring intestinal nascent cells), a dual binary expression system that combines spatio-temporally controlled remote expression of ligands with differentially labelled intestinal progenitors. As described in the main text, Hamelin consists of established driver stocks for midgut (*esg*) and hindgut (*byn*)[51,105]. While combining both systems and adding fluorophores, we also added a common repressor *tub-Gal80^ts^* for both expression systems that allows temperature mediated temporal expression control.

## Cloning of *lexAop-fra-RNAi* construct
The *lexAop-fra-RNAi* construct was generated as described previously[106]. Genomic DNA of the VALIUM20 based fra-RNAi line from the Transgenic RNAi Project (Trip-3, BL40826) was isolated and added to a PCR reaction using the Q5 High-Fidelity DNA Polymerase (NEB) and the following primers: shRNA_GA_F primer (5′-GAGAACTCTGAA-TAGATCTGTTCTAGAAAACATCCCATAAAACATCCCATATTCA-3′) and shRNA_GA_R1 primer (5′-CTCTAGTCCTAGGTGCATATGTCCACTCTA GTA-3′)[106].

The pWALEXA20 vector[106] was digested with XbaI and NdeI prior to assembling with the amplified fra-shRNA by Gibson-assembly. After verification by colony PCR and sanger sequencing constructed plasmids were amplified and injected into embryos of *nos-phiC31;;attP86Fb* flies for integration on the third chromosome[107].

## Immunohistochemistry
The guts of adult female flies were dissected in 1xPBS and transferred into glass wells containing 4% PFA immediately after dissection. After 45 min of fixation the guts were washed once by replacing the PFA with 1xPBS. Primary antibodies were diluted in 1xPBS with 0.5% Triton-X and 5% normal goat serum. The incubation with primary antibodies (1:250 anti-Arm [mouse; Developmental studies Hybridoma Bank (DSHB)]; 1:200 anti-Dl [mouse; Developmental studies Hybridoma Bank (DSHB)]; 1:250 anti-Dlg1 [mouse; Developmental studies Hybridoma Bank (DSHB)]; 1:200 anti-GFP [chicken; Abcam ab290]; 1:500 anti-HA [rat; Roche clone 3F10]; 1:500 anti-myc [rabbit; Invitrogen PA1-981]; 1:100 anti-NetB, [rabbit[108];];1:250 anti-Pros [mouse; Developmental studies Hybridoma Bank (DSHB)]) was performed on an orbital shaker at 4 °C over night. The guts were washed with 1xPBS prior to incubation with secondary antibodies (1:500 Goat anti-Chicken Alexa488 [Invitrogen]; 1:500 Goat anti-Rabbit Alexa568 [Invitrogen]; 1:500 Goat

anti-Mouse Alexa647 [Invitrogen]; 1:500 Goat anti-Rat Alexa647 [Invitrogen]) and DAPI (1:1000; 100 µg/ml stock solution in 0.18 M Tris pH 7.4; DAPI No. 18860, Serva, Heidelberg) for at least 1 ½ h at RT. After washing with 1xPBS for a last time, the stained guts were mounted in a drop of Fluoromount-G Mounting Medium (Electron Microscopy Sciences) mixed with a drop of 1xPBS on a microscope slide and covered with a coverslip.

### Image acquisition
After immunostaining PMGs were imaged using a LSM 710 confocal microscope (Carl Zeiss Microscopy GmbH, Germany) with an 40x objective. Image resolution was set to at least 3440 × 3440 pixels. About four to five confocal planes with 1 µm interval were scanned and combined into a Z-stack to image one cell layer with all different cell types.

### Survival analysis
For survival assays of *Netrin* mutants (Fig. S6C) and *fra* mutants (Fig. S3N) flies were collected after hatching and kept at 25 °C. For survival analyses of and *klu$^{ReDDM}$* specific manipulations (Fig.2N–P) and I-KCKT specific manipulations (Fig. S2Y) of *fra* and *unc-5* flies were raised at 18 °C and collected at an age of 0–3 days before shifting them to 29 °C. All flies analysed in survival assays were kept in vials of at least 10 flies with a 2:1 ratio of females:males. Flies were checked daily to record death events and flipped every three days to avoid demise by contaminated food. At least 26 flies were assessed per genotype. Survival curves were plotted using GraphPad Prism 9.0 and analysed by Kaplan-Meier Log-rank survival test.

### RNA isolation and cDNA synthesis
For expression analysis of *fra, NetA, NetB* and *unc-5* in the adult *Drosophila* midgut (Fig. S1A) cDNA of adult mated *w$^{1118}$* females was isolated.

For expression analysis of *fra* and *Netrins* in midguts of flies challenged by feeding dioctyl sodium sulfosuccinate (DSS, Fig.5F) mated *w$^{1118}$* flies were aged for two days at 25 °C and starved for four hours in empty vials prior to feeding with Sucrose and Sucrose with 5% DSS. A piece of Whatman paper was soaked in a solution of 5% Sucrose in dH$_2$O or 5% Sucrose solution with 5%DSS and added to an empty vial. Flies were added to the prepared vial and kept at 29 °C. The Whatman paper was changed after 24 h to prevent dehydration of the flies. After 48 h at 29 °C midguts of female flies were dissected for RNA-isolation and cDNA synthesis.

For expression analysis of *NetB* in midguts with EC specific expression of >*p35* to block EC apoptosis[109–111] (*mex* > *p35*, Fig.5H) *EC-Rapport* flies were crossed to >*p35* at 18 °C and their progeny shifted to 29 °C for seven days prior to RNA-isolation and cDNA synthesis.

For all conditions 8 flies were dissected per condition and cDNA and RNA isolation and cDNA synthesis were performed as described previously[112].

### Expression analyses by conventional and quantitative real-time PCR
Expression of *fra, NetA, NetB* and *unc-5* in whole midguts of adult female flies was tested by conventional PCR by amplification of gene fragments using the same primer pairs that were also used for quantitative real-time PCR (Table 1). After conventional PCR amplified gene fragments were run on a 0.8% agarose gel.

Expression levels of *fra, NetA* and *NetB* were determined upon drug induced injury by feeding dextran sulphate sodium (DSS, Fig.5F) as described previously[113]. Expression levels of NetB were determined upon blockage of EC death by overexpression of >*P35*[114] (Fig.5H). Real-time qPCR was performed as described previously[99]. Relative expression levels were normalized to the house-keeping gene *Rp49* and calculated by ΔΔCt-Method.

### Mechanical injury of posterior midguts
Mechanical injury of PMGs was induced by pinching the abdomen of female flies as performed previously[9]. Flies were pinched after seven days of *esg$^{ReDDM}$* tracing and put back to 29 °C for 30 min to recover prior to dissection.

### Quantification of cell numbers, protrusion numbers and lengths, protrusion directions and migration distances
Fiji (ImageJ 1.51 n, Wayne Rasband, National Institutes of Health, USA) was used to calculate maximum intensity images out of Z-stacks from PMGs. Progenitor cell numbers, numbers of progeny, and number of protrusions from *esg$^{ReDDM}$*[9] *klu$^{ReDDM}$*[11], ISC specific ReDDM tracing, *EC-Rapport* and FLP-Out clones experiments were counted manually using Fiji.

Protrusion lengths were measured using Fiji by drawing free hand lines reaching from the nuclear envelope to the tip of the protrusion. EB specific manipulations by *klu$^{ReDDM}$*, ISC specific manipulations by ISC specific ReDDM tracing and ISC/EB specific manipulations by *esg$^{ReDDM}$* were performed for seven days by shifting the flies to 29 °C for one week. *EC-Rapport* specific expression of >*p35* and >*NetB-RNAi* (Fig.5J–L, Q–R) was also performed for seven days, whereas *EC-Rapport* specific knockdown of *E-cad* (Fig.5M–N, Q–R, S–U) was run for three days due to a severe disruption of midgut homoeostasis upon shifting for seven days and *EC-Rapport* specific manipulation of >*NetB* (Fig.5P–R) were run for 24 h due to early fly death after shifting for three days.

For measuring protrusion directionality upon EB specific manipulations of *fra* and *unc-5* shown in Fig.2I–K EB located in between old and new EC were considered. For each of these EB it was analysed whether the leading protrusion (longest protrusion per EB) points towards old EC or new EC.

Protrusion directions shown in Fig.6F–G were measured using the angle tool from Fiji. Two lines were drawn, one following the protrusion of an ISC/EB and one connecting the ISC/EB with an RFP labelled EC clone. The angle between these two lines was used to describe the direction of a protrusion in relation to an EC clone. Polar histograms were generated using the ggplot 2 package v3.3.3[115] in R Studio.

For measurements of migration distances as readout for the 'Hamelin-assay' (Fig.7) a line was drawn along the midgut-hindgut boundary (MHB) defined by the *byn > H2B::RFP* positive cells. Distances of single GFP$^+$ cells to the MHB were measured by drawing a straight line which is orthogonal to the MHB and reaching from the most posterior point of the cell membrane to the MHB. Measurements of EE to ISC/EB distances upon EE-Rapport manipulations (Fig. S6D–I) were performed similarly. A straight line was drawn between the membrane of an EE and its nearest ISC/EB.

### Quantification of rEC renewal frequency
For quantification of rEC renewal frequency *klu$^{ReDDM}$* manipulations were performed for seven days by shifting adult flies to 29 °C. *EC-Rapport* specific expression of >*NetB-RNAi* (Fig.5V–X) was run for 14 d. After dissection midguts were stained by antibodies targeting Arm[5] and Pros[6] to allow identification of ISC (GFP$^-$RFP$^-$aArm$^+$aPros$^-$) in confocal images. Imaged midguts were analysed using Fiji to manually count the number of new rEC distant from ISC. This number was then divided by the total number of new EC to calculate rEC renewal frequency.

For quantification of rEC renewal frequency upon *EC-Rapport*-induced knockdown of *NetB* (Fig.5T–V) flies were shifted for 14 days due to the lack of renewed EC after shifting for seven days. In *EC-Rapport* tracing experiments ISC can not be distinguished from EB and we defined rEC with no contact to GFP$^+$ ISC/EB as rEC.

### Quantification of aNetB intensities
For intensity measurements of aNetB staining (Fig.5S–U), PMGs were scanned with fixed laser/exposure time settings. Fluorescence

intensities were analysed in Fiji by determining the mean intensity per area of manually selected ROI within EC, which we identified by aDlg1 staining. When selecting the ROI we excluded the nuclei to prevent measurement of false signal. 50 EC per gut (indicated as 'N' in figure panels) in three different guts (indicated as 'n' in figure panels) were measured.

## Statistical analyses

Statistical analyses were run in GraphPad Prism 9.0. Box plots show the first and third quartile with mean indicated by light blue lines and full range of values indicated by whiskers, additionally all individual values are shown by dots. No data were excluded. Bar plots show the mean (light blue lines) and full range of values indicated by whiskers, additionally all individual values are shown by dots. For statistical comparisons we first tested all data for normal distribution using the Shapiro-Wilk test[116] and then used either two-sided students $t$ test[117] for comparison of two groups with normal distribution or One-way ANOVA[118] followed by Dunn's test or the two-stage step-up method of Benjamini, Krieger and Yekutieli[119] for multiple comparisons of normally distributed data[120]. For data that did not show normal distribution we used non-parametric Mann–Whitney U test[121] for comparison of two groups and Kruskal-Wallis test[122] followed by two-stage step-up method of Benjamini, Krieger and Yekutieli[119] for multiple comparisons[120]. Survival curves were analyzed using Kaplan-Meier log-rank tests[123,124]. Significant differences are displayed as * for $p \leq 0.05$, ** for $p \leq 0.01$, *** for $p \leq 0.001$ and **** for $p \leq 0.0001$. Results of all statistical tests can be found in the raw data sheets.

## Reporting summary

Further information on research design is available in the Nature Portfolio Reporting Summary linked to this article.

## Data availability

All datasets generated in this study are provided in the Supplementary Information/Source Data file. Source data are provided with this paper.

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

## Acknowledgements

The authors thank the Vienna Drosophila RNAi Centre (VDRC) and the Bloomington Drosophila Stock Centre (NIHP400OD018537) and Hermann Aberle, Greg Bashaw, Ross Cagan, Maria Dominguez, Thomas Klein, Michael Murray and Achim Paululat for reagents. We also thank the Centre for Advanced Imaging (CAi) at Heinrich Heine University (DFG INST 208/539-1 FUGG) for imaging training and facilities. In addition, we are grateful to Guido Grossmann, Matias Zurbriggen and Reiff laboratory members for comments and feedback on experiments and the manuscript. TR is supported by the Deutsche Forschungsgemeinschaft (DFG-Sachbeihilfe RE 3453/2–1, RE 3453/6-1), the Wilhelm Sander-Stiftung (2018.145.1) and the Deutsche Krebshilfe (70115333).

## Author contributions

L.Z. performed and analysed all experiments except Fig.2H,Fig. S2A,H,S,T,Fig. S3E-F,Fig. S4N-O,Fig. S5M-N conducted by P.R. and F.A.G. The conception and design of experiments was by L.Z., P.R. and T.R. L.Z. and T.R. wrote the manuscript.

## Funding

## Competing interests

The authors declare no competing interests.
