## [Transparent Peer Review file · Nature Communications]

Frazzled/DCC directs spatial progenitor integration ensuring steady-state intestinal turnover

Corresponding Author: Dr Tobias Reiff

Version 1:

Reviewer comments:

Reviewer #1

(Remarks to the Author)

In this manuscript by Zipper et al., the authors identify roles for the Netrin-Frazzled chemotactic signaling axis in directing enteroblasts in the *Drosophila* midgut towards “worn-out” enterocytes for homeostatic replacement. Among the key findings, interfering with Frazzled function in intestinal stem cells impairs enteroblast protrusion formation and enterocyte replacement, Netrin expression in the midgut is increased when enterocyte turnover is stimulated, loss of Netrin-B produces similar defects as Frazzled loss of function, and ectopic Netrin-B expression can attract enteroblasts towards regions of the gut they normally do not populate. The manuscript also reports effects of Unc-5 gain and loss of function on enteroblast polarization and enterocyte replacement, and it demonstrates stimulating effects of Frazzled and Unc5 knockdown on tumor cell dispersion in a fly model of colorectal cancer.

The authors arrive at a model in which enterocytes that are targeted for turnover produce Netrin-B to attract enteroblasts via Frazzled, facilitating their replacement. This provides an explanation for the ability of enteroblasts to replace enterocytes that are not directly in contact with any stem cells, and many of the presented data, which are of generally high quality, provide good evidence supporting this idea. These findings would be of interest to a fairly broad readership of stem cell biologists and researchers that study Netrin signaling in other contexts, including developmental neurobiologists. However, there are a number of conceptual gaps in the manuscript, and data that are not supporting the main hypothesis or cannot be easily reconciled with each other are either not sufficiently discussed or entirely ignored. These inconsistencies have to be addressed, and there is also a need for additional experiments to further bolster the main conclusions. Moreover, the cancer metastasis portion of the manuscript is underdeveloped and does not integrate well with the rest of the study. I recommend removing it. With appropriate revisions, the manuscript could make an important contribution.

Specific comments and questions:

1. Questions remain about the contributions of Frazzled and Unc-5 to Netrin-dependent enteroblast polarization and migration.

a) The conclusion that DCC and Unc-5 fulfill antagonistic functions is not fully supported by the data. For instance, Frazzled LOF, Frazzled GOF, Unc-5 LOF, and Unc-5 GOF all disrupt enteroblast protrusion directionality towards old enterocytes, yet these manipulations have opposite effects on various other parameters, including enterocyte renewal rate. This result is not explained, and it calls into question whether protrusion directionality is causally connected to enterocyte replacement. Furthermore, in the fly colorectal cancer model, Unc-5 and Frazzled knockdown both promote tumor dispersion.

b) Frazzled knockdown impairs enterocyte replacement and leads to reduced fly survival, which is taken as evidence that reduced enterocyte turnover leads to organismal demise. However, Frazzled overexpression, which stimulates enterocyte replacement, also reduces survival, but this is ignored.

c) It is not determined which receptor mediates the attractive effect of Netrin-B in the “Hamelin” assay and leads to the accumulation of Netrin-B puncta on enteroblasts. This seems critical for nailing down the proposed function of Frazzled in enteroblast migration.

d) It should be discussed how Unc-5 does or does not fit into the proposed model for Netrin-Frazzled signaling in enteroblast homing. Unc-5 is observed in intracellular vesicles, yet its knockdown and overexpression cause clear phenotypes. How are these effects explained?

2. The results implicating Frazzled downstream mediators are too preliminary.

a) Knocking down various regulators of cytoskeletal dynamics impairs enteroblast protrusion formation, but whether this

reflects their proposed roles in Frazzled signaling remains unclear. All of these molecules have been implicated downstream of multiple different receptors, and none of the experiments establish an epistatic relationship with Frazzled. The knockdown effects are also more pronounced than for Frazzled itself. Are these molecules recruited to Frazzled and/or change their activity in the relevant context? If these cytoskeletal regulators cannot be clearly linked to Frazzled, maybe the experiments that implicate them should be removed from the manuscript.

b) Various kinases are recruited to the P motifs in Frazzled/DCC and mediate the axon guidance activities of Netrin. Are these mediators required for enteroblast polarization and enterocyte replacement?

c) It is not explained why Frazzled lacking the P3 domain cannot support enterocyte renewal, even though it can restore protrusion number and length under Frazzled gRNA.

3. The relationship between Netrin-B expression, enteroblast protrusion directionality, and enterocyte renewal should be fleshed out better.

a) Figure 5E shows Netrin-B expression by a single enterocyte, and Figures S6A and B provide further evidence for Netrin-B expression by a subset of enterocytes. It remains unclear how many enterocytes express Netrin-B at any given moment and how that relates to enterocyte age and the average turnover rate for enterocytes.

b) Are enteroblast protrusions preferentially oriented towards remote Netrin-B-expressing enterocytes?

c) Does ectopic Netrin-B expression in groups of enterocytes re-orient enteroblast protrusions towards such cells and cell-autonomously increase enterocyte replacement rate? Such experiments would establish enteroblast chemotaxis towards Netrin-B in a slightly more relevant context than the "Hamelin" assay.

4. The colorectal cancer part seems preliminary and does not integrate well into the manuscript.

a) Are the differences in Figure 7F statistically significant?

b) It is unclear why Unc-5 and Frazzled RNAi have similar effects in this model (also see comment 1a).

c) The manuscript title states that "Frazzled/DCC directs intestinal progenitor homing in ... metastasis." This is not supported by the data. How does the Frazzled knockdown phenotype in the colorectal cancer model connect to its effect on enteroblast migration and enterocyte renewal under homeostatic conditions?

d) As mentioned in the discussion section of the manuscript, antibodies that block Netrin-1 function (in the context of Unc-5) show promise in suppressing endometrial cancer metastasis in clinical trials, yet Unc-5 knockdown seems to promote metastatic potential in the fly colorectal cancer model. How can these seemingly contradictory results be integrated into a coherent model?

5. Minor comments:

a) The overall use of abbreviations in the manuscript seems excessive, making it hard to follow the text in certain places. Even a simple word like "Notch", used only a few times in the paper, is abbreviated as "N".

b) Fly survival data from organism-wide Netrin manipulation in Figure 5T do not meaningfully connect to Netrin functions in midgut homeostasis.

c) The model in Figure S7G, which seems to depict Netrin-B-mediated enteroblast attraction to sites of injury, does not seem to be consistent with the result that mechanical injury induces enteroblast protrusion formation even under Frazzled RNAi (Figure S7C; results not quantified).

d) It is unclear why human Netrin-5 was chosen to test functional conservation in the "Hamelin" assay, when Netrin-1 is the canonical attractive DCC ligand in mammals.

Reviewer #2

(Remarks to the Author)

The manuscript entitled "Frazzled/DCC directs intestinal progenitor homing in 1 homeostasis and metastasis" investigates the mechanism underpinning the replacement of "isolated" worn-out ECs (rECs) that do not have immediate ISC/EB neighbours during homeostasis. The authors propose that worn-out rECs secrete the pro-migratory signal, NetB, which signals through Fra on nearby EBs to stimulate their migration and differentiation to replace rECs. The hypothesis that the replacement of isolated ECs involves EB migration is novel and interesting. However, the overall claims need to be substantiated. The authors use the kluREDDM system to monitor EC renewal events and discriminate between the replacement of adjacent and remote ECs (aECs and rECs). They rely on this "end point" readout as a proxy for EB migration. As highlighted below, the contribution of EB migration to rEC replacement events during homeostasis needs to be further strengthened. Furthermore, based on poorly defined Fra and Unc-5 expression patterns (due to low protein expression levels), the authors claim that Fra and Unc-5 are exclusively expressed in EBs. To exclude that Fra and Unc-5 are also expressed in ISCs – which is important for the claim that NetB specifically affects EB migration – effects of ISC-specific Fra and Unc-5 KD should be investigated. Similarly, the connection between worn-out ECs and NetB expression should be further substantiated. Addressing the concerns below would significantly strengthen the main hypothesis of the paper.

Major concerns:

•Based on stainings of guts harboring endogenously GFP-tagged Fra and Unc-5 (Figure 1SB-G'), the authors claim that Fra and Unc-5 are exclusively expressed in EBs. Based on the images presented in Fig. S1, it is not possible to exclude that either of the two might also be expressed in ISCs, and this claim should be further substantiated. 1) If possible, these data should be strengthened by adding the expression patterns of Fra and Unc-5 using existing reporter lines. 2) Does ISC-specific knockdown of Fra or Unc-5 affect ISC protrusions (numbers/length) or the proportion of newly generated rECs? Does Unc-5 KD increase tissue turnover (REDDM)?

•Why are Unc-5 receptors predominantly localized in intracellular compartments? Are they actively signaling during

homeostasis – in the absence of dying ECs?

- The authors make a direct correlation between a reduction in the fraction of newly produced rECs and impaired migration. While EBs are often found in pairs with ISCs, they are also found without ISC neighbors throughout the epithelium. Therefore, rECs could also arise from the differentiation of such isolated EBs. It would be important to quantify the fraction of EBs that are without ISC neighbors. Also, what is the effect of Fra KD or Unc-5 OE on the replacement of aECs? Related to this question, is the reduction in rEC replacement intimately linked with Fra/NetB signaling or would any impairment of EB-to-EC differentiation have relatively bigger effects on the replacement of rECs (compared with aECs)?
- The authors use correlation rather than direct evidence to conclude that EBs migrate to replace worn-out rECs. To fully support this claim, it would be important to monitor EB migration during homeostasis, to replace worn-out rECs, or alternatively, using their Hamlet assay using live cell microscopy (maybe using the approach in Marchetti et al., 2022).
- The effects on lifespan upon fra or unc-5 KD in EBs are quite modest. Did the authors use backcrossed flies for this experiment?
- The authors show that progenitors expressing a dominant negative version of Notch have reduced protrusions, which can be rescued by Fra OE or unc-5 KD. The interpretation of these results is a little hard, as manipulation takes place in both ISCs and EBs. What is the effect of Fra KD or OE or unc-5 KD in ISCs on protrusions in NDN cells?
- As mentioned previously, the assumption that Fra and unc-5 are exclusively present in EBs needs to be substantiated. Does Fra OE/KD or unc-5 KD affect ISC-to-EB differentiation – this could be done using Transtimer.
- It is perplexing that knockout of Net-A, a ligand that is expressed 3- to 4- fold more than Net-B in the midgut, has minimal effects on EB protrusivity, given that Net-A also attracts progenitors across the MHB in the Hamelin assay. Could the authors evaluate the Net-A expression pattern in the midgut?
- The authors present confocal images to show that NetB is detected in singular ECs – it would be great if these images could be accompanied by a quantification of number of NetB positive ECs in the posterior gut to see how that correlate with replacement events.
- The authors suggest that Net-B is expressed by the worn-out rEBs. How did they confirm that these EBs are worn-out? Could they check if markers of cell death co-localize with ECs that express Net-B? Could the authors perform protrusion direction assay as they did for Fra knockdown (Figure 2k) to see if EC-specific Net-B knockdown or OE randomizes the orientation of protrusion?
- Is NetB required for the increase in protrusions triggered by E-Cad KD in figure 5O?
- In Figure 5T, it would be more relevant to examine the effect of EC-specific NetB KD on lifespan
- The authors show that OE of NetB from EECs does not trigger migration/clustering of ISCs/EBs around EECs. Does EC-specific OE of NetB trigger protrusions in ISCs/EBs?

Minor comments

- Several of the figure items are lacking information regarding the drivers used. It would strongly facilitate the “reading” of the figures if driver information was present in all items.
- It would be recommendable to add DAPI channel together with ReDDM confocal micrographs instead of GFP and RFP channels alone. This would help readers to get a better idea of rEC renewal vs aEC renewal.
- In lines 105-106, DI+ instead of D1-, DIg1+ instead of DIg1- and Pros+ instead of Pros-.
- In supplementary figure S4, the figure labels show esgReDDM instead of Klu ReDDM.
- Statistics missing for new rEC quantification for Fra Δ P3 in Figure 4I.
- Authors state that EBs expressing Fra lacking (line 202) P1 or P2 domain migrate normally and behave like controls. But these conditions look more closer to EBs with overexpression of full-length Fra than control EBs. This reviewer would recommend clarifying this.
- This reviewer does not agree with the notion in line 186 that “Fra and Unc-5 are not per se sufficient inducers of differentiation in the intestinal homeostatic lineage”, as their expression in EBs are sufficient to trigger EB-to-EC differentiation (Figure 2M). Their results mainly show that N is required downstream or in parallel to Fra and/or Unc-5 to promote EB-to-EC differentiation, while Fra GOF or unc-5 LOF is sufficient to suppress the reduction in protrusions of ISC-like progenitors triggered by N LOF.

(Remarks to the Author)

Proper proliferation of residential adult stem cells and differentiation of their progeny are required to maintain the homeostasis of adult tissues. Adult intestinal epithelium is a good example of tissue homeostasis maintenance. Intestinal stem cells (ISCs) constantly produce differentiating progeny to replace dead differentiated cells, most of the time, enterocytes (ECs). However, how dead ECs are replaced by differentiating progeny (enteroblasts, EBs) is not fully understood. Here, Lisa Zipper and colleagues propose a hypothesis, in which so called adjacent ECs (aECs) and remote ECs (rECs) in intestinal epithelium, that dying ECs could secrete attracting Netrin-B (NetB), in addition to cytokines and EGFs, to bind to its receptor Frazzled (Fra)/DCC on differentiating EBs to promote their migration to the site of dying ECs and replace them in the end. Furthermore, they claim that Fra/DCC also functions in CRC metastasis.

However, there are critical flaws in their hypothesis and proposal: 1) ISCs are almost evenly scattered in intestinal epithelium, thus, most, if not all, ECs are in the one-EC-diameter cycle of ISCs. So intestinal epithelium is composed of (arranged by) thousands of such cycles with ISCs in the center and only a small portion of ECs are out of the boundary of neighboring cycles. 2) Even such so called rECs exist, their distances to neighboring ISCs are different (not equally away from surrounding ISCs). If such ECs are dying and secrete NetB. Here comes to the term of morphogen gradient. The secreted NetB from the dying ECs form a gradient, with the highest concentration of NetB from the producing dying ECs and decreasing its concentration as the distance increases. So why the more distanced ISC/EB bothers to sense the signal and migrates toward the dying ECs (so called rEC by the authors), but not the much more nearby ISC/EB (so called aECs)? So there is paradigm of their hypothesis and is fatal to their hypo. 3) In the experiments carried out, all EBs are labelled and traced, how the authors differentiate and precisely sure that the so called newly produced so called rEC is produced from which EB? As they can not ascertain the true origin of the newly produced so called rECs, how can they claim the newly produced so called rEC is produced and migrated from an ISC far away from the dying EC? 4) Further, even if their proposal is true under normal conditions, in which they claim that NetB from dying rECs attracts EB from far away (here coined remote EB, rEB, following the authors) and the rEB cells migrate to replace the dying rEC to maintain intestinal homeostasis. If Fra, the receptor, is compromised in EBs, no migration is occurred, thus no repair of intestinal epithelium, leading to damaged epithelium. However, on the contrary to their proposal, in the metastasis model the authors established, compromising Fra/DCC pathway increased metastasis of such tumor cells instead. What a surprise! This observation is totally in contrast to their proposal. So one can conclude that their proposal is wrong or with fatal foundation. Altogether, these fundamental weakness points of this manuscript prevent me to recommend it for publication in Nature Communications.

Other major or minor concerns (from the beginning of the MS to the end):

1. the title of this MS is "Frazzled/DCC directs intestinal progenitor homing in homeostasis and metastasis", the authors used the term homing, which caused a big confusion. As all of us know, home normally refers to where a person or something is originally born or should often resides (like homing of transplanted blood stem cells). In the case of differentiating EBs (they called EBs as progenitor which is also misleading), EBs are produced by ISCs and will migrate a little bit of distance and differentiate into ECs. So where is the home of EB in the eyes of the authors? The place of EB's birth: nearby or in close contact with ISC or the location where it is fully differentiated?
2. the writing of the MS is problematic, there are many grammar and usage errors. For example, mis-usage of singular and plural forms is frequently observed throughout the MS: lines 15-16: worn-out EC express.....; line 20: our data establishes.....; line 21: provides.....; line 28: ..(PMG) prove... And so on and on. Terms such as Notch-signaling, Delta-ligands are not properly written. Punctuations are often missing, such as line 115.
3. The authors should provide clear evidence to show that the so called rECs do exist in midgut epithelium, using membrane labelling and cell identity markers.
4. Fig. 1G, the authors should indicate the way to differentially label ISC+EB and EB (lexA and Gal4).
5. lines 80-87, how the authors claim that the so called aECs and rECs are replaced at identical frequency? This is in contrast to common sense. How they differentiate the origin of the so called rECs? They should perform MARCM clones to examine this.
6. line 103-6, the labelling of cell identity is misleading, not in ISC (DI-), EC (DIg-)....
7. line 139, conclusion from Fig. 2J-K, authors' previous data showed that depletion of fra abolished protrusion formation, while here the authors concluded that randomized protrusion formation upon fra depletion. Without protrusion formation in the absence of fra, how to randomize?
8. lines 146-8, depleting fra caused early death of flies, and they claimed that this is caused by the inability to repair. However, under infection conditions in which ECs are rapidly being renewed, the infected flies are also short lived. The authors should not draw conclusions favoring their claims. The cause of fra depletion may not be related to EC repair at all!
9. the authors should use MARCM clone technique to show the migration of EBs toward the so called rECs and how fra mutation affects this. As the authors concern the background mutations on fra mutant chromosome, they should generate new fra mutant with CRISPR/Cas9 tech and examine their MARCM clones to unambiguously show whether EBs migrate (or the clones grow) toward the so called dying rECs (judged by NetB expression) in control and blockage of this process by fra mutation.
10. lines 172-180, the authors should not express NDN to mimic Notch LOF, as NDN is not complete N LOF. Furthermore, the cells on the outside boundary of the esg clusters are still differentiating. They should use mutants of Notch pathway component to generate MARCM clones, such as FRT42D-mam and FRT42D-mam, fra (new mutant generated on mam background) to draw a more convincible conclusion.
11. lines 190-206, it's already known that these protrusions are mediated by actin cytoskeleton, so why bother again?
12. line 213 and related lines, the labelling of NetAB-/-NetA- and NetAB-/-NetB- is incorrect and misleading, it should be NetA-/-NetB-/+ (only the former is showed).

13. lines 228-239, the authors should clarify what types of injury, infection-induced? Drug-induced or mechanical damage?
14. line 260, Gal80ts can not control *lexA* activity.
15. lines 277-281, the authors should use markers to mark the midgut/hindgut boundary, otherwise, it is misleading to draw conclusions.
16. lines 324-6, how the authors explain the discrepancy in the CRC model and under physiological conditions? These results are contradictory to their former proposal.
17. how can *w1118* be used as control? The authors should specify in what experiments, it was used as control, while what is the control in the other experiments. The authors should not just simply use the term control, they should specify it in every control panel as different genetic backgrounds affect the phenotype, thereby affecting the conclusions drawn.
18. Fig. 1B, in addition to the model, the authors should present representative images of midgut epithelium.
19. The authors should specify which region of midgut is examined, in text and Figs.
20. Fig. 2B. how ectopic expression of *fra* promote more and longer protrusions and more ECs are produced? Does its ectopic expression also promote ISC division? It seems to be so. How?
21. As EBs are accumulated upon *fra* knockdown, does *Fra* knockdown inhibit EC differentiation or promote ISC proliferation as well?
22. Fig. 2I etc, the direction of protrusion showed in the image is not as claimed by the authors, in control, the newly formed ECs are not always face a protrusion. And upon *fra* knockdown, almost all ECs are surrounded by progenitors (incl. ISC), so does in control.
23. Fig. 2N. how you explain o/E *fra* also reduced life span? And LOF of *fra*. *Esg/klu* in not only express in gut, it also expresses in other tissues. So the cause of life span shortening is complex. The authors should be very cautious on this.
24. Fig. 3, the authors should use null mutants (RNAi knockdown) of key components of the Notch pathway to re-perform the experiments.
25. Fig 5A-C, why more ECs are accumulated in 5C compared to 5A and B?
26. Fig. 6. The boundary of MG and HG should be labelled by a marker, such as *Wg-lacZ* etc, otherwise, when artificially draw the boundary is mis-leading. Fig. 6E, why no HA signal is seen in the MG? a gradient of HA should be seen if it is secreted from HG. Again, if *NetB-Fra* could promote protrusion in EBs and their migration toward the source of *NetB* producing cells, then why no such long protrusions toward HG are seen in Fig. 6D, E, G, and H? According to the authors' model, they should producing HG facing protrusions. However, in most of these so called migrating cells, the protrusions are barely observed, neither in the presence of *NetB* overexpression (Fig. 6D-H). Without these protrusions, how progenitors migrate and cross the boundary? The aim of Fig.6 is to strengthen their proposal that *NetB-Fra* attraction, however, this observation is in contradictory to their proposal.
27. Fig. 7G. Besides to the contradiction to their proposal, the authors should be very cautious about the cause of early death in these manipulated flies. to draw a conclusion that the shortening of life span is due to CRC, they should inhibit the activity of *klu* in other tissues, only letting it active in MG.
28. Fig. S1D shows that *Fra* is also expressed in EE, not as claimed by the authors in EBs. Meanwhile, as *fra* and *unc5* are both expressed in EBs and the authors propose that they function in the opposite manner, how EBs are coordinated and *Fra* is dominant over *unc5*?
29. Fig. S3B. no so called protrusions seen, but with more newly produced ECs. How come?
30. Fig. S3H, where is the protrusion in cells in the regenerated region?
31. Fig. S3Q, this is likely an EB clone, not ISC MARCM clone.
32. Fig. S6F. Is *NetB* expressed in EE properly secreted? If yes, why there are no protrusions produced (not even mentioning toward *NetB*-producing EE)? If not, why?
33. Fig. S7B-D. in the control panel, very few protrusions are facing the injury site. Only one cell in the presented panel, while protrusions can still be formed in S7C and at the similar rate.

Reviewer #4

(Remarks to the Author)

Version 2:

Reviewer comments:

Reviewer #1

(Remarks to the Author)

In this revised manuscript by Zipper et al., the authors have addressed most of my prior critiques. In particular, the addition of *fra* LOF data in the Hamelin assay and the evidence supporting EB protrusion chemotaxis towards *NetB*-expressing cell clones considerably strengthened the manuscript. I have two remaining major comments that should be addressed through textual revisions. First, one sticking point for me remains the description and discussion of data where effects of experimental manipulations on EB protrusion number/length/directionality do not go hand in hand with effects on EC replacement. This includes *unc-5* RNAi (disrupts EB protrusion orientation towards old ECs but does not affect EC renewal) and rescue of *fra* LOF with *fra* lacking P3 (restores protrusion number and length to control levels but does not rescue EC replacement). It remains unclear to me how these results can be reconciled with a model where EC replacement depends

on protrusion dynamics, and the authors need to provide a satisfying mechanistic explanation that accommodates these results in the discussion of their model. Second, I commend the authors for removing the cancer metastasis portion of the manuscript, but I believe that the section of the Discussion that speculates about a possible role for Net/Fra in metastasis should be dramatically shortened or removed. If it is included, this part should strictly focus on direct implications of the presented results for tumor EMT and metastasis.

Minor comment:

“Notch”, used only a few times in the paper, is still abbreviated as “N” in the relevant figure. I encourage the authors to further reduce the number of abbreviations and also stay consistent with their use of abbreviations.

Reviewer #2

(Remarks to the Author)

The authors have done a great job addressing several of the major concerns raised by the reviewers. They convincingly demonstrate that Fra function is required in EBs, but not in ISCs, for the replacement of rECs. In addition, they show that progenitor cells extend directional protrusions toward individual ECs with ectopic NetB expression.

That said, several points would benefit from further clarification or additional analysis:

1. In Figure 5E, NetB appears to be upregulated in an aEC located adjacent to an EB rather than in an rEC. Would NetB expression in aECs not obscure or mask a gradient emanating from rECs? Could NetB represent a more general signal associated with worn-out aECs and rECs, rather than being rEC-specific?
2. Related to this, the fraction of newly generated rECs is calculated relative to the total number of new ECs, leading the authors to conclude that the rEC fraction decreases while the aEC fraction increases. However, the absolute number of newly generated aECs is not addressed. Is the number of aECs reduced upon NetB or Fra knockdown? Is NetB specifically required in rECs? The strong reduction in the total number of newly generated ECs in Figure 2M is unlikely to be explained solely by a loss of rECs. A plot showing the absolute numbers of newly generated rECs and aECs would greatly strengthen this conclusion.
3. In Figures 5S–T, NetB is also detected in stem and progenitor cells, which argues against a simple role as a directional guidance cue. Moreover, it is not clear from Figure 5T that NetB is upregulated specifically in rECs. I am not sure that I would include this item in the final manuscript, as the requirement for NetB downstream of E-cad to replace rECs is not corroborated by functional data.
4. Based on Figure S1B, it is not entirely clear that Fra is absent from ISCs, and this conclusion may be overstated. While the functional data showing that Fra is not required in ISCs for rEC renewal are convincing, the observation that Unc-5 overexpression in ISCs reduces progeny numbers (Figure 3J) suggests that endogenous Fra activity may exist in ISCs. Although the authors propose that ectopic Unc-5 may form homodimers with stronger effects, it is equally plausible that Fra is present in ISCs at levels below the detection threshold of the endogenously tagged Fra::GFP.
5. Finally, although knockdown of Fra in ISCs/EBs reduces longevity, this experiment does not allow a clear assessment of the contribution of EC-derived NetB to this phenotype.

Reviewer #3

(Remarks to the Author)

The Authors addressed most of concerns raised in previous manuscript. I would recommend for publishing the revised manuscript.

Reviewer #4

(Remarks to the Author)

Version 3:

Reviewer comments:

Reviewer #1

(Remarks to the Author)

The authors have addressed my most pressing concerns, and I support publication of the manuscript in its current form.

Reviewer #2

(Remarks to the Author)

I am satisfied with the textual changes the authors have made to address my comments. I think the manuscript is ready for publication

General response to all reviewers:

Dear Reviewers,

First of all, we want to thank all Reviewers for the valuable suggestions that helped to improve our revised manuscript. Following the Reviewer's comments, we updated the manuscript with additional suggested experiments and reworked the text in major parts. Especially, the newly included data based on suggested experiments supports our main finding that Netrin-Frazzled/DCC signalling directs enteroblast migration during *Drosophila* midgut homeostasis.

One of the two key questions of all Reviewers was whether NetB is also an attractant in the midgut epithelium adding to our findings from the Hamelin assay that might reflect invasion but not physiological homeostatic turnover in the midgut lineage. This issue can now be answered positively: Developing a new clonal approach to express NetB in single EC reveals that indeed enteroblast protrusions and migration are directed towards NetB-expressing EC. Please find data in our new Fig.6 and experimental details discussed in each reviewer's comments.

The second central question that came up was whether NetB indeed signals through Fra-receptors. Using Hamelin, we now provide evidence that this is indeed the case and that NetB-stimulated migration depends on Fra-receptors included in the new Fig.7.

A central suggestion from the reviews was to whether to include or exclude our datasets addressing the role of Netrin-Frazzled/DCC/Unc-5 signalling in our colorectal cancer (CRC) model (Fig.7 of the original manuscript). Various publications describe a central role for DCC and UNC-5 and their loss-of-function in CRC (Bernet et al 2007, Fearon et al 1990, Kinzler & Vogelstein 1996a, Shibata et al 1996b, Shin et al 2007). Interestingly, our data on migratory behaviour and fly survival in our CRC model (Fig.7 of the original manuscript) supports the findings from human CRC pathology. Even though in accordance with above literature, we decided to remove the original Fig.7 for stringency of the revised manuscript and clarity to fit the new Figures 6 and 7.

However, we keep investigating the CRC model and the role of protrusions and migration as well as Unc-5 and Fra/DCC in a currently running project in which we direct metastatic behaviour of circulating CRC cells to more distant organs. Just as a side note here, we discover essential roles for both genes for survival and identity that only show up in the complexity of multiple tumoral mutations and are not focus of this current manuscript describing the physiological role in migration direction in a non-neuronal tissue. We agree with the reviewers that these effects require more thorough investigations and their exhaustive presentation in a new study and manuscript. Thus, we now removed Fig.7 and focus with new data on the midgut role of Netrin-Frazzled/DCC/Unc-5 signalling added to the new main figures (Figs.3, 6, 7) and the concise pathological findings from the Hamelin assay without the complexity of tumoral mutations clouding their physiological role.

Together, our new experiments presented in Fig.6+7 now connect the previous findings on Fra/DCC-Unc-5 receptors in EB to the release of NetB from EC directly and thus provide the new finding that NetB indeed signals through Fra-receptors (Fig.7G,H). Our findings show that wild-type intestinal cells can be attracted to Netrin sources, which is especially interesting for pathological contexts and therapy development as it shows that no additional mutations are necessary for invasive behaviour.

In the following, we address the major and minor concerns of the Reviewers point-by-point.

To maintain the original numbering of the Reviewers' comments, we added prefixes R1-3 corresponding to Reviewers 1-3 (for example, 'comment R3#4' refers to the fourth comment from Reviewer 3). Our references to figures in this response letter refer to the new figures, if not indicated else.

REVIEWER COMMENTS

Reviewer #1 (Remarks to the Author):

In this manuscript by Zipper et al., the authors identify roles for the Netrin-Frazzled chemotactic signaling axis in directing enteroblasts in the *Drosophila* midgut towards “worn-out” enterocytes for homeostatic replacement. Among the key findings, interfering with Frazzled function in intestinal stem cells impairs enteroblast protrusion formation and enterocyte replacement, Netrin expression in the midgut is increased when enterocyte turnover is stimulated, loss of Netrin-B produces similar defects as Frazzled loss of function, and ectopic Netrin-B expression can attract enteroblasts towards regions of the gut they normally do not populate. The manuscript also reports effects of Unc-5 gain and loss of function on enteroblast polarization and enterocyte replacement, and it demonstrates stimulating effects of Frazzled and Unc5 knockdown on tumor cell dispersion in a fly model of colorectal cancer. The authors arrive at a model in which enterocytes that are targeted for turnover produce Netrin-B to attract enteroblasts via Frazzled, facilitating their replacement. This provides an explanation for the ability of enteroblasts to replace enterocytes that are not directly in contact with any stem cells, and many of the presented data, which are of generally high quality, provide good evidence supporting this idea. These findings would be of interest to a fairly broad readership of stem cell biologists and researchers that study Netrin signaling in other contexts, including developmental neurobiologists. However, there are a number of conceptual gaps in the manuscript, and data that are not supporting the main hypothesis or cannot be easily reconciled with each other are either not sufficiently discussed or entirely ignored. These inconsistencies have to be addressed, and there is also a need for additional experiments to further bolster the main conclusions. Moreover, the cancer metastasis portion of the manuscript is underdeveloped and does not integrate well with the rest of the study. I recommend removing it. With appropriate revisions, the manuscript could make an important contribution.

We thank Reviewer #1 for this very positive and motivating evaluation of our manuscript. In this revision, we add new experiments that address the inconsistencies and bolster our main conclusions. Please find our comments in the following.

Specific comments and questions:

R1#1. Questions remain about the contributions of Frazzled and Unc-5 to Netrin-dependent enteroblast polarization and migration.

a) The conclusion that DCC and Unc-5 fulfill antagonistic functions is not fully supported by the data. For instance, Frazzled LOF, Frazzled GOF, Unc-5 LOF, and Unc-5 GOF all disrupt enteroblast protrusion directionality towards old enterocytes, yet these manipulations have opposite effects on various other parameters, including enterocyte renewal rate. This result is not explained, and it calls into question whether protrusion directionality is causally connected to enterocyte replacement. Furthermore, in the fly colorectal cancer model, Unc-5 and Frazzled knockdown both promote tumor dispersion.

The reviewer is right by pointing out similar effects on protrusion directionality upon fra GOF and LOF as well as unc-5 LOF (Fig. 2K) and several more issues that we address in the upcoming questions. First concerning the protrusion directionality, to our understanding from the current knowledge in the field, protruding lamellipodia are highly dynamic and undergo cycles of protrusion formation and retraction.

Additionally, it is known that lamellipodia are stabilized at the leading edge by the formation of adhesions. This stabilisation of lamellipodia and their adhesion at the leading edge are crucial to initiate actin polymerisation and pushes the plasma membrane forward towards the direction of the leading edge (reviewed in (Petrie & Yamada 2012)). In our analysis shown in Fig.2K, we considered the most prominent and longest protrusion, the ‘leading protrusion’, of the EB located in between a mosaic of old and new EC. We hypothesize that this ‘leading protrusion’ is the lamellipodia which is stabilized and will form adhesions mediated by the interplay of Fra and Unc-5.

We speculate that the stabilization of a leading protrusion strongly depends on physiological levels of Fra and Unc-5 receptors to properly detect NetB levels from EC. In our GOF and LOF experiments we disturb the fine-tuned receptor abundance and protrusions might become over- or desensitized to unchanged NetB abundance. Additional details can be found in R1#1c.

Strikingly, in our newly included experiments suggested by this Reviewer (R1#3c), we now provide strong evidence showing that leading protrusions point towards NetB overexpressing EC (Fig.6), which shows that, with physiological levels of Fra and Unc-5, protrusions indeed point towards NetB sources. We updated the manuscript accordingly.

b) Frazzled knockdown impairs enterocyte replacement and leads to reduced fly survival, which is taken as evidence that reduced enterocyte turnover leads to organismal demise. However, Frazzled overexpression, which stimulates enterocyte replacement, also reduces survival, but this is ignored.

We agree with the reviewer that reduced survival upon overexpression of fra needs mentioning and discussion in the main text. Previous data showed that both cellular phenotypes, impaired as well as stimulated EC replacement, can result in reduced fly survival by exhausting the stem cell pool and subsequent loss of epithelial barrier function (Antonello et al 2015a, Jin et al 2017, Rera et al 2011). We now added this to the main text of the revised manuscript (lines 165ff).

c) It is not determined which receptor mediates the attractive effect of Netrin-B in the “Hamelin” assay and leads to the accumulation of Netrin-B puncta on enteroblasts. This seems critical for nailing down the proposed function of Frazzled in enteroblast migration.

We agree with the reviewer that showing a direct effect Fra on EB migration is key to reveal whether Netrin-B signals through Fra in our Hamelin assay and we now provide this key data in Fig.7. As described in the main text, Hamelin is based on dual expression systems which basically allows an AOP-driven knockdown of fra (>>fra) in midgut progenitors while we express attracting NetB in the hindgut with byn>NetB. Hamelin already consists of a total of six transgenes and two more, >NetB and >>aop-RNAi, needed to be added by recombination of chromosomes. During the revision process, we (finally) managed to perform this experiment by using the same esg>> driver for expression of a fluorophore (>>H2B::mCherry), tub-Gal80^{ES}, and >>fra-RNAi specifically in ISC/EB and combined it with the same byn> as in Hamelin to drive expression of >NetB::GFP (Okada et al 2023) to attract ISC/EB and simultaneously show byn> activity by GFP signal in the HG. After expression of transgenes for seven days like in Hamelin, we could not detect any fra-depleted midgut ISC/EB migrating across the MHB, which suggest that Fra mediates the attractive effect of NetB in our Hamelin assay. We added this important data to Fig.7G-H of the revised manuscript.

In addition, we provide further evidence that support Frazzled, but not Unc-5, in the coordination of EB migration:

- i) Our experiments using klu^{reDDM} to manipulate Fra and Unc-5 receptor levels within EB suggest that Fra mediates the attractive effect in EB migration as OE of fra induces protrusion formation (Fig.2B,H).*
- ii) KD/KO of fra disrupts protrusion formation and EB migration as shown by decreased rEC renewal (Fig.2C,F,H,L).*
- iii) Unc-5 OE however abolishes rEC replenishment (Fig.2L) most probably by antagonistically acting on Fra-receptors (Keleman & Dickson, 2001).*
- iv) unc-5-RNAi neither reduces rEC (Fig.2L) nor...*
- v) ...fly survival (Fig.2O) suggesting that Fra is still functioning ensuring homeostasis and midgut integrity under these conditions.*
- vi) In addition, unc-5-RNAi does not elicit protrusions in ISC (Fig.3F,G).*
- vii) A subordinated role of Unc-5 is further supported by its internalization (Fig.S1E-G,I'''), which is generally assumed to indicate active fra/DCC signalling (Boyer & Gupton 2018).*
- viii) In line with this, Fra receptors remain on the membrane when Unc-5 is overexpressed and internalized (Fig.S1I-I'').*
- ix) Double OE and KD of both receptors phenocopies the sole Fra manipulations (Fig.S3A-G).*
- x) When Fra is depleted with >>fra-RNAi, NetB::GFP puncta on membranes of progenitors are not detected and accumulated anymore (Fig.7J) compared to wild-type progenitors (Fig.7I).*

Taken together, we think this is compelling evidence for Fra mediating the attractive effect in EB protrusion formation and migration. We updated the manuscript accordingly with Figures 6 and 7 as well as in the discussion (lines 492-523).

d) It should be discussed how Unc-5 does or does not fit into the proposed model for Netrin-Frazzled signaling in enteroblast homing. Unc-5 is observed in intracellular vesicles, yet its knockdown and overexpression cause clear phenotypes. How are these effects explained?

As pointed out in the previous comment (R1#1c), Unc-5 is generally accepted to antagonize Fra/DCC receptors in the literature by acting repulsive in heterodimers with Fra/DCC, whereas homodimers of Fra or DCC on the membrane act attractive (O'Donnell et al 2009). Thus, either Unc-5 depletion or internalization allows Fra/DCC to signal as homodimer and direct migration. Our data on the role of Unc-5 in EB migration supports previous findings from neurodevelopmental studies and point to a subordinate role below Fra/DCC:

- i) EB specific manipulations of unc-5 show opposing effects to Fra on protrusion formation (Fig.2H,L, Fig.S2A).*
- ii) In addition, rEC renewal (Fig.2L) works normal when Unc-5 is depleted from EB suggesting Fra homodimers direct EB movement.*
- iii) Unc-5 OE in EB however reduces rEC formation (Fig.2L) and fly survival (Fig.2O) by preventing Fra signalling through its abundance.*
- iv) As mentioned in our comment R1#1c, internalization of Unc-5 into vesicles is described during axon guidance and indicates attractive signalling via Fra/DCC (Boyer & Gupton 2018). In midgut EB, internalisation of Unc-5 into vesicles with GFP tagged Unc-5 (unc-*

- 5::GFP, Fig.S1E-H') and new antibody stainings targeting Unc-5 (new Fig.S1E) is observed as well, which in turn suggests signalling through Fra/DCC homodimers.
- v) Importantly, only homodimers of Fra/DCC were shown to connect to the F-Actin network allowing protrusion formation (Garbe et al 2007, Golenkina et al 2018).
 - vi) In vitro experiments on rat neurons showed removal of UNC5H1 from the cell surface of neurons and growth cones but leaves DCC at the cell surface. These changes of receptor levels at the cell surface could lead to a switch from UNC5H1 mediated repulsion to DCC induced attraction towards a Netrin-1 gradient (Williams et al 2003).
 - vii) As mentioned previously, membrane-bound Netrins to DCC indicate active signalling (Lai Wing Sun et al 2011), which we observe in Hamelin (Fig.7I) and discover in our new experiments that these puncta depend on the presence of Fra-receptors (Fig.7J).

Together, our data suggests that the model from neurodevelopmental studies also holds true for midgut EB. Presence of Unc-5 (by overexpression) on the membrane leads to repulsive heterodimers (Fra/Unc-5) and depletion as well as internalization allows Fra homodimers to direct migration. We made these findings clearer in the text and added this topic to the discussion now.

R1#2. The results implicating Frazzled downstream mediators are too preliminary.

a) Knocking down various regulators of cytoskeletal dynamics impairs enteroblast protrusion formation, but whether this reflects their proposed roles in Frazzled signaling remains unclear. All of these molecules have been implicated downstream of multiple different receptors, and none of the experiments establish an epistatic relationship with Frazzled. The knockdown effects are also more pronounced than for Frazzled itself. Are these molecules recruited to Frazzled and/or change their activity in the relevant context? If these cytoskeletal regulators cannot be clearly linked to Frazzled, maybe the experiments that implicate them should be removed from the manuscript.

We agree with the reviewer that it is important to draw a clear link of the investigated cytoskeleton regulators and Fra. In lines 191 – 199 of the original manuscript, we describe this link that was previously observed during axon outgrowth and guidance and in the revised manuscript (lines 239ff), we state now more clearly that we confirm established findings from neurodevelopmental studies.

Genetic as well as physical interactions of Fra with Abl and Trio were shown as well as that Ena is a substrate of Abl and that it regulates axon pathfinding (Forsthoefel et al 2005, O'Donnell & Bashaw 2013). Additionally, it was already shown that Fra is able to promote cellular, F-actin enriched protrusions in epithelial cells, which depend on the Arp2/3 F-Actin polymerization complex component Arpc3 (Golenkina et al 2018). Taken together, the links between Fra and the cytoskeleton regulators Arpc3, Ena and Trio are solid (Forsthoefel et al 2005, Golenkina et al 2018, O'Donnell & Bashaw 2013), which prompted us to confirm similar roles of these regulators in Fra/DCC induced EB protrusions and migration shown in Fig.S4 and S5.

As the reviewer correctly commented, the investigated cytoskeleton regulators are involved in multiple receptors and signalling pathways. The additional finding, we want to point out with our data is that Arpc3, Ena and Trio act downstream of stimulated protrusion elongation using >fra and >unc-5-RNAi (Fig.S4) and show that in this case protrusion formation and migration depend on presence of Arpc3, Ena and Trio (Fig.S4L,M,N).

Concerning the KD of Arpc3, Ena and Trio being more pronounced than Frazzled: We interpret these findings in two ways, one is that RNAi acts directly on transcripts of these gene that are directly involved in F-Actin dependent protrusion formation and just prevent protrusion induction more efficiently. Second, Henri Jaspers lab already described lamellipodia on ISC that form controlled by Otk upon injury (Hu et al 2021). In Fig.S7A-D', we discover that the same mechanisms (Otk signalling in ISC) might be at work in injury conditions that, importantly, are Frazzled-independent but also still depend on Trio. Thus, Trio, being involved in protrusion formation in general, might logically show the more pronounced KD phenotypes shown for the cytoskeleton regulators in Fig.S4 compared to fra KD (Fig.2C).

In addition to lines 235 – 239 of the original manuscript, we now discuss this in the revised manuscript in lines 304-314 and adjusted the text accordingly. We now point out more clearly that there is an established link between Fra and the shown cytoskeleton regulators that might also function in further migration regulating pathways in EB.

b) Various kinases are recruited to the P motifs in Frazzled/DCC and mediate the axon guidance activities of Netrin. Are these mediators required for enteroblast polarization and enterocyte replacement?

During Drosophila development, it was shown that the Abelson tyrosine kinase (Abl) is recruited to the P motifs in Fra and mediates axon guidance (Dorsten et al 2007, Forsthoefel et al 2005, O'Donnell & Bashaw 2013). We share the curiosity of the reviewer and already investigate the Abl kinase and its role in EB migration as part of an ongoing project in the lab.

c) It is not explained why Frazzled lacking the P3 domain cannot support enterocyte renewal, even though it can restore protrusion number and length under Frazzled gRNA.

This is an interesting point and we also wondered when we first obtained this data: Neurodevelopmental studies of the Fra P-motifs in Drosophila axon guidance made similar observations and already revealed that expression of fra^{ΔP1}, fra^{ΔP2} or the combination fra^{ΔP1ΔP2} all rescued the axon guidance defects in a fra mutant background. However, expression of fra^{ΔP3} does not rescue axon guidance defects. In addition, the same authors observed a significant difference compared to fra mutants upon expression of fra^{ΔP3} suggesting a mild rescue of attractive function, but at much lower levels compared to fra^{ΔP1} and fra^{ΔP2} (Garbe et al 2007).

In our experiments shown in Fig.4 the fold changes in protrusion numbers/length of >fra^{gRNA},>fra^{ΔP3} compared to >fra^{gRNA},>fra^{ΔP1} and >fra^{gRNA},>fra^{ΔP2} are 0.79/0.38 and 0.86/0.40 respectively, which also suggest a mild, but weaker rescue ability of fra^{ΔP3} compared to fra^{ΔP1} and fra^{ΔP2}. We made this clearer in the revised manuscript (lines 254-259).

R1#3. The relationship between Netrin-B expression, enteroblast protrusion directionality, and enterocyte renewal should be fleshed out better.

a) Figure 5E shows Netrin-B expression by a single enterocyte, and Figures S6A and B provide further evidence for Netrin-B expression by a subset of enterocytes. It remains unclear how many enterocytes express Netrin-B at any given moment and how that relates to enterocyte age and the average turnover rate for enterocytes.

The reviewer raises a fundamental point here; it would be important to further strengthen the connection between NetB expressing EC and EC turnover. We previously showed that EC turnover rates in the posterior midguts of adult flies are constant and do not change with age in homeostatic conditions (Antonello et al 2015a).

Methods like ReDDM show that the average number of renewed EC per R5 region is around 73 EC in 7 days. This indicates a renewal of around 10 EC per day and thus an average EC turnover rate of 0.43 EC per hour. An EC turnover rate of 0.43 indicates a low probability to find a dying EC at a fixed given time point and is in line with single NetB expressing EC we find and show in Fig.5E, S6A,B.

Dissected midguts are always fixed prior to antibody staining and subsequent imaging always shows a ‘snapshot’ of the actual state of the midgut epithelium. The average EC turnover rate of 0.43 EC per hour fits our observation of 0-4 EC per R5 positive for aNetB staining or NetB> and NetB>> activity. Using aNetB antibodies, we investigated NetB levels in control midguts (Fig.5S-S’) and compared them to aNetB levels upon EC specific expression of >E-cad-RNAi (Fig.5T-T’) stimulating homeostatic turnover as published by the O’Brien lab (Liang et al 2017). Suggesting a connection between EC death and secretion of NetB, aNetB levels are higher upon >E-cad-RNAi expression affecting the whole EC population (Fig.5U). We added these important datasets to Fig.5 of the revised manuscript and discuss this in the text (lines 269-290).

b) Are enteroblast protrusions preferentially oriented towards remote Netrin-B-expressing enterocytes?

Yes, we kindly refer the reviewer to our answer to the next and former comment (R1#3c & R1#1a) respectively, which describe new results of additional experiments supporting our previous findings (new Fig.6). Briefly, these data indicate that ISC/EB located within a 25 µm radius of single >NetB-expressing EC clones exhibit more numerous and elongated protrusions oriented toward the clone. More details are found in the next comment.

c) Does ectopic Netrin-B expression in groups of enterocytes re-orient enteroblast protrusions towards such cells and cell-autonomously increase enterocyte replacement rate? Such experiments would establish enteroblast chemotaxis towards Netrin-B in a slightly more relevant context than the “Hamelin” assay.

We agree with the reviewer that the analysis of EB protrusion orientation and migration towards EC ectopically expressing >NetB would be more relevant directly in the midgut epithelium. More importantly, we thank the reviewer for suggesting this idea, which lead to a major new experiment. In the course of the revision we took up the technical challenge that finally turned into a very valuable validation to our model that NetB directs EB migration.

*From the technical side: We performed Flp-out clones induced by heat-shock driven flippase (Pignoni & Zipursky 1997), which are positively labelled by Gal4 driving >RFP and additionally are able to express >NetB. In combination with lexA-Aop-based *esg>>CD8::GFP,tub-Gal80^{ts}* this enabled us to label ISC/EB and protrusions as well. Together, >NetB is ectopically expressed in clonal EC that might occur in groups or as single >NetB expressing EC and simultaneously, ISC/EB are labelled by *esg>>CD8::GFP* independently of Gal4/UAS (Fig.6A-C).*

Analysing these clones and ISC/EB behaviour, we were able to show that ISC/EB protrusions project towards RFP⁺ >NetB expressing clones despite their sometime remote location from the

ISC/EB (Fig.6C). We quantified protrusion numbers and lengths in ISC/EB located close to >NetB expressing clones and found them to have more numerous (Fig.6E) and longer protrusions (Fig.6D). Furthermore, this method allowed us to analyse the angle of protrusions regarding the direction of a >NetB expressing clone compared to control clones. An angle of 0° means direct targeting of a protrusion towards a clone (Fig.6F,G). By analysing hundreds of protrusions, we could show that the angles for protrusions of ISC/EB located close to a >NetB expressing clone are significantly smaller compared to the ones of ISC/EB located close to control clones (Fig.6F-G), further underlining that protrusions are directed towards NetB sources. Furthermore, we found accumulation of GFP⁺ ISC/EB along the membranes of RFP⁺ >NetB expressing EC clones (Fig.6H-J).

These observations indicate that ISC/EB are attracted by >NetB expressing EC clones and start forming protrusions pointing towards the NetB source and migrate towards the clone. We added these important data to Fig.6 and lines 291-299 and 318-325 of the revised manuscript.

R1#4. The colorectal cancer part seems preliminary and does not integrate well into the manuscript.

Following Reviewers 1 and 3, we agree and removed Fig.7 from the revised manuscript and focussed on the physiological role of Netrin-Fra/DCC and Unc-5. We are currently elaborating an exhaustive story on protrusions, their effects on cellular identity and migratory behaviour in CRC models, which we will now submit separately including Fra/Unc-5 datasets.

However, we include comments to the Reviewers comments as they also serve understanding for the physiology. Please find more details in the following comments.

a) Are the differences in Figure 7F statistically significant?

The Graph in Fig.7F (of the original manuscript) shows the proportion of midguts with indicated BM phenotype, e.g. in 'CRC avatars' 13.33% of midguts showed non-invasive tumours. These proportions cannot be faithfully compared statistically as we only get one proportion for each class of BM phenotype and genotype.

b) It is unclear why Unc-5 and Frazzled RNAi have similar effects in this model (also see comment 1a).

The reviewer is right and we could speculate on many different possibilities here. Unlike the physiology described in Fig.1-6, our CRC model analysed in Fig.7F (of the original manuscript) mimics a pathological condition specifically for gastrointestinal cancer that is already far from physiology even without further Fra/Unc-5 manipulations. Our CRC-model used to obtain the data shown in Fig.7F (of the original manuscript) includes expression of oncogenic >Ras^{G12V} (Fig.7B of the original manuscript). The expression of this activated Ras in turn activates several downstream signalling pathways encompassing PI3K/Akt/mTOR pathway which was already shown to stimulate cellular migration in neutrophils (Lin et al 2024, Pal et al 2023). It is expected that >Ras^{G12V} also stimulates tumour cell migration independent of Fra/DCC and Unc-5 in our *Drosophila* CRC model.

Interestingly our finding that Unc-5 and Frazzled RNAi have similar effects in this model, is underlined by the mutational situation in CRC patients. In flies, our results of fra- and unc-5-RNAi both stimulating cell dispersion in a CRC model. In lines 291 – 298 of the original manuscript we describe the observations of DCC LOH and UNC-5C LOF occurring more often in

later stages of CRC and correlate with CRC spreading to multiple metastatic sites (Fig.7A of the original manuscript).

In addition after Virchow, tumours are wounds that never heal and migration as well as protrusions in our CRC model might thus also stem from stress-induced regeneration that does not depend on Netrin-Fra/DCC signalling (Fig.S7A-C') but is induced by other signalling pathways encompassing the non-canonical Wnt signalling activated via Otk (Hu et al 2021).

Together, as our model reflects what happens in late stage CRC patients with very pleiotropic effects. Thus, we can only speculate that when the model is combined with >fra-RNAi and >unc-5-RNAi these migrating tumour cells increasingly lose the information (tissue identity, 'context') of where they belong or should migrate to and eventually disperse from the primary tumour in the midgut. In addition to the discussed involvement of Fra/DCC and Unc-5 in MET, there is also data from rodent models that involves both genes as 'dependence receptors' controlling apoptosis (Castets et al 2011, Grady 2013). Although, we did observe indication for apoptosis upon Fra/DCC and Unc-5 manipulations in physiology, this might very well play a part in pathology and apoptosis also facilitating metastatic behaviour. Revealing the underlying mechanisms is a current topic in the lab and an equally interesting and challenging topic of our ongoing research.

c) The manuscript title states that "Frazzled/DCC directs intestinal progenitor homing in ... metastasis." This is not supported by the data. How does the Frazzled knockdown phenotype in the colorectal cancer model connect to its effect on enteroblast migration and enterocyte renewal under homeostatic conditions?

The reviewer is right and after removing the CRC model, we use a new title for the revised manuscript:

'Frazzled/DCC directs spatial progenitor integration ensuring steady-state intestinal turnover'

We do employ the term progenitor here as we think it is more suitable than enteroblast for the broad readership of Nature Communications.

d) As mentioned in the discussion section of the manuscript, antibodies that block Netrin-1 function (in the context of Unc-5) show promise in suppressing endometrial cancer metastasis in clinical trials, yet Unc-5 knockdown seems to promote metastatic potential in the fly colorectal cancer model. How can these seemingly contradictory results be integrated into a coherent model?

The reviewer raises an intriguing point here, which we plan to address in more detail in a more exhaustive analysis separating physiology (this paper) and pathology (new submission).

As mentioned in our general response to all reviewers, we are developing new tumour models to dissect and mimic different cancer stages to trace (and direct) metastasizing cells. This will hopefully eliminate confounding effects (R1#4b) and allow us to further unravel the role of Fra/DCC and Unc-5 within cancer cells and the tumour microenvironment during different steps of cancer progression.

R1#5. Minor comments:

a) The overall use of abbreviations in the manuscript seems excessive, making it hard to follow the text in certain places. Even a simple word like “Notch”, used only a few times in the paper, is abbreviated as “N”.

We agree with the reviewer that an immoderate use of abbreviations makes it unnecessarily difficult for the reader to follow the text, especially in journals with a broad readership such as Nature Communications. This has been corrected and we reduced the use of abbreviations by removing the ones which are not used frequently in the text.

b) Fly survival data from organism-wide Netrin manipulation in Figure 5T do not meaningfully connect to Netrin functions in midgut homeostasis.

It is correct that a constitutive whole animal mutant does not necessarily correlate with Netrin functions in midgut homeostasis. However, the majority of investigations that focussed on Netrin function investigated neuronal development. As flies we investigated develop into living imagos (Brankatschk & Dickson 2006) and phenotypically reproduce our cell-type specific manipulations (Fig.5D, Q), we would like to include the survival data of these well-characterized Netrin mutant flies (Brankatschk & Dickson 2006) (Fig.6K) as well as fra-mutant flies (Kolodziej et al 1996) (Fig.S3N) as they complement our data with established alleles.

c) The model in Figure S7G, which seems to depict Netrin-B-mediated enteroblast attraction to sites of injury, does not seem to be consistent with the result that mechanical injury induces enteroblast protrusion formation even under Frazzled RNAi (Figure S7C; results not quantified).

We thank the reviewer for pointing this out and agree that the data on homeostasis and injury was displayed in a confusing way in the original Fig.S7, which we now removed in favour of a cartoon focussing on homeostasis (Fig.7M).

We updated the manuscript accordingly and also discuss the behaviour of protrusions during homeostasis and injury in more detail (lines 300-314).

d) It is unclear why human Netrin-5 was chosen to test functional conservation in the “Hamelin” assay, when Netrin-1 is the canonical attractive DCC ligand in mammals.

When we started working with the Hamelin assay only hNTN5 transgenics were available. During the revision process, human Netrin-1 (hNTN1) transgenics became commercially available and functional analysis in Hamelin revealed similar results. The new data is now added to Fig.7 (E,H) in the revised manuscript.

Reviewer #2 (Remarks to the Author):

The manuscript entitled “Frazzled/DCC directs intestinal progenitor homing in 1 homeostasis and metastasis” investigates the mechanism underpinning the replacement of “isolated” worn-out ECs (rECs) that do not have immediate ISC/EB neighbours during homeostasis. The authors propose that worn-out rECs secrete the pro-migratory signal, NetB, which signals through Fra on nearby EBs to stimulate their migration and differentiation to replace rECs. The hypothesis that the replacement of isolated ECs involves EB migration is novel and interesting. However, the overall claims need to be substantiated. The authors use the kluREDDM system to monitor EC renewal events and discriminate between the replacement of adjacent and remote ECs (aECs and rECs). They rely on this “end point” readout as a proxy for EB migration. As highlighted below, the contribution of EB migration to rEC replacement events during homeostasis needs to be further strengthened. Furthermore, based on poorly defined Fra and Unc-5 expression patterns (due to low protein expression levels), the authors claim that Fra and Unc-5 are exclusively expressed in EBs. To exclude that Fra and Unc-5 are also expressed in ISCs – which is important for the claim that NetB specifically affects EB migration – effects of ISC-specific Fra and Unc-5 KD should be investigated. Similarly, the connection between worn-out ECs and NetB expression should be further substantiated. Addressing the concerns below would significantly strengthen the main hypothesis of the paper.

We thank the reviewer for carefully evaluating our work and providing valuable suggestions for experiments that substantially improved our revised of our manuscript. In the following, we address the raised concerns point-by-point.

Major concerns:

R2#1 •Based on stainings of guts harboring endogenously GFP-tagged Fra and Unc-5 (Figure 1SB-G’), the authors claim that Fra and Unc-5 are exclusively expressed in EBs. Based the images presented in Fig. S1, it is not possible to exclude that either of the two might also be expressed in ISCs, and this claim should be further substantiated. 1) If possible, these data should be strengthened by adding the expression patterns of Fra and Unc-5 using existing reporter lines. 2) Does ISC-specific knockdown of Fra or Unc-5 affect ISC protrusions (numbers/length) or the proportion of newly generated rECs? Does Unc-5 KD increase tissue turnover (REDDM)?

1) We agree with the reviewer, that the images shown in Fig.S1B-H’ are difficult due to the low expression of both receptors, which is why we performed additional experiments to sustain EB expression of both receptors. We also checked more endogenous GFP-tagged and established stocks from BL and Vienna Stock Centers, the Bashaw, Paululat and Aberle labs but did not find better signal/noise ratio. Furthermore, we tried to strengthen our data by performing antibody stainings targeting Fra and Unc-5 in midguts with differentially labelled ISC (GFP⁺/RFP⁺) and EB (GFP⁺/RFP⁺) as performed in Fig.1G,H.

Unfortunately, with stainings for aFra, we were unable to detect any reproducible signal likely due to the age of the available antibodies. For aUnc-5, we detected signal within vesicular structures inside of EB (white arrow, Fig.S1E-E’), which is absent in ISC (white arrowhead, Fig.S1E-E’). We added this important data to Fig.S1E-E’’ and the revised manuscript (lines 108-111).

2) To further exclude a role of *Fra* and *Unc-5* in ISC, we needed to develop an ISC-specific ReDDM version to perform ISC manipulations and tracing of *Fra* and *Unc-5* as suggested by the reviewer.

In order to enable visualization of protrusions and ReDDM tracing to identify rEC stemming from manipulated ISC, we combined the ISC/EB specific *esg^{ReDDM}* system (Antonello et al 2015a) with an established EB specific *Su(H)-Gal80* (Wang et al 2014) to repress *Gal4* activity in EB. Consequently, *Gal4* activity is restricted to ISC allowing specific manipulation and labelling by GFP and RFP. As in *esg^{ReDDM}* and *klu^{ReDDM}*, the long-lived H2A::mCherry persists in ISC progeny and labels also newly produced EB, EC and EE (Fig.3A). Antibody staining targeting the ISC/EB marker Armadillo (Arm) (Micchelli & Perrimon 2006) and the EE marker Prospero (Pros) (Ohlstein & Spradling 2006) allows to distinguish EB, EC and EE.

ISC specific, and putatively ectopic, expression of *>fra* increases ISC-protrusion lengths (Fig.3B-C, G-H, Fig.R1) similar to the results of EB specific OE of *fra* using *klu^{ReDDM}* (Fig.2B, H, L, Fig.S2A, Fig.R1). Most importantly, ISC specific knockdown of *fra* is not affecting ISC protrusion lengths nor the proportion of newly differentiated rEC (Fig.3D, G-H). Lack of the effect of *>fra*-RNAi also underlines EB-only expression of *Fra* (Fig.S1B-D') and our *klu^{ReDDM}* datasets (Fig.2L). This points to a very low to no expression level in ISC and that *>fra* induced protrusions are due to ectopic expression. Additionally, protrusion induction by *>fra* is much weaker when directly compared (Fig.R1). For facilitation, we include a direct comparison for the reviewer as Fig.R1 here.

Fig.R1: Depletion of *Fra* and *Unc-5* affect EB protrusions, but not ISC protrusions

Quantification of protrusion lengths upon manipulations of *fra* and *unc-5* in EB using *klu^{ReDDM}* and in ISC using ISC specific ReDDM. For box plots, the boxes show median, 25th and 75th percentiles, and whiskers indicate the range of values. All data points are shown by dots and means are indicated by light blue lines. 'n' are numbers of biological replicas and 'N' values describe numbers of technical replicas. Asterisks denote significances from Mann Whitney tests (***p<0.0001).

In line with our expression analysis of *Unc-5*, its depletion in ISC does not affect protrusion length nor migration directionality reflected by rEC formation (Fig.3F, G, H). In addition, ISC (Fig.3I) and progeny numbers (Fig.3J) are unaltered, further supporting no *Unc-5* expression in ISC. Ectopic *Unc-5* expression in ISC has a small but significant effect on protrusion length and

significantly reduces progeny numbers (Fig.3E,G,I,J) but not rEC (Fig.3H). Absence of Fra in ISC suggests that ectopic Unc-5 then might form homodimers with function even stronger than normal repulsion and apoptosis (Keleman & Dickson 2001).

The last question concerned whether Unc-5 KD stimulates tissue turnover, which is shown not to be the case (Fig.3F,J) for ISC-ReDDM, almost certainly due to the lack of unc-5 expression. In contrast, in *klu^{ReDDM}* unc-5 depletion strongly stimulates generation of new EC (Fig.2E, M) suggesting stimulation of Fra activation due to the release of antagonism from Unc-5.

Taken together our analysis of i) GFP::tagged Fra and Unc-5, ii) the antibody staining of Unc-5 and iii) the absence of effects of their knockdowns in ISC, underline that the observed functions of both, Fra and Unc-5, in protrusion formation and migration are EB exclusive. We added the new data derived from ISC specific manipulations of Fra and Unc-5 in Fig.3A-J of the revised manuscript.

R2#2 •Why are Unc-5 receptors predominantly localized in intracellular compartments? Are they actively signaling during homeostasis – in the absence of dying ECs?

This is an interesting and central question that was also raised by the other Reviewers.

As pointed out now in the revised manuscript, Unc-5 is generally accepted to antagonize Fra/DCC receptors in the literature by acting repulsive in heterodimers with Fra/DCC, whereas homodimers of Fra or DCC on the membrane act attractive (O'Donnell et al 2009). Thus, either Unc-5 depletion or internalization allows Fra/DCC to signal as homodimer and direct migration. Our data on the role of Unc-5 in EB migration supports previous findings from neurodevelopmental studies and point to a subordinate role below Fra/DCC:

- i) EB specific manipulations of unc-5 show opposing effects to Fra on protrusion formation (Fig.2H,L, Fig.S2A).*
- ii) In addition, rEC renewal (Fig.2L) works normal when Unc-5 is depleted from EB suggesting Fra homodimers direct EB movement.*
- iii) Unc-5 OE in EB however reduces rEC formation (Fig.2L) and fly survival (Fig.2O) by preventing Fra signalling through its abundance.*
- iv) Internalization of Unc-5 into vesicles is described during axon guidance and indicates attractive signalling via Fra/DCC (Boyer & Guppton 2018). In midgut EB, internalisation of Unc-5 into vesicles with GFP tagged Unc-5 (*unc-5::GFP*, Fig.S1E-H') and new antibody stainings targeting Unc-5 (new Fig.S1E) is observed as well, which in turn suggests signalling through Fra/DCC homodimers.*
- v) Importantly, only homodimers of Fra/DCC were shown to connect to the F-Actin network allowing protrusion formation (Garbe et al 2007, Golenkina et al 2018).*
- vi) In vitro experiments on rat neurons showed removal of UNC5H1 from the cell surface of neurons and growth cones but leaves DCC at the cell surface. These changes of receptor levels at the cell surface could lead to a switch from UNC5H1 mediated repulsion to DCC induced attraction towards a Netrin-1 gradient (Williams et al 2003).*
- vii) As mentioned previously, membrane-bound Netrins to DCC indicate active signalling (Lai Wing Sun et al 2011), which we observe in Hamelin (Fig.7I) and discover in our new experiments that these puncta depend on the presence of Fra-receptors (Fig.7J).*

Together, our data suggests that the model from neurodevelopmental studies also holds true for midgut EB. Presence of Unc-5 (by overexpression) on the membrane leads to repulsive heterodimers (Fra/Unc-5) and depletion as well as internalization allows Fra homodimers to

direct migration. We made these findings clearer in the text and added this topic to the discussion now.

In addition, the Reviewer was asking for *Unc-5* internalization before EC need to be replaced. We experimentally addressed this question by dissection midguts of young flies. In these young midguts, internalization into vesicles occurs is also observed (24h), before homeostatic turnover requires first EC replacements. (Fig.R2A-A''). Thus, internalization of *Unc-5* seems to occur and allow *Fra* to signal even before EC replacement is initiated.

unc-5::GFP

Fig.R2: *Unc-5* is internalized in EB of young midguts without the need of EC replacement

(A-A'') Confocal images of GFP-tagged *Unc-5* in midguts of young flies (24h after hatching). *Unc-5::GFP* is internalized in ISC/EB (GFP punctae in small *Arm*⁺ cells).

A second effect, we checked for is that depletion mimicking internalization might elicit a pro-apoptotic role when *Unc-5* functions as dependence receptor (Grady 2013, Mehlen & Tauszig-Delamasure 2014). In the midgut epithelium, we are familiar with cell death (Reiff et al 2019), but did not detect any dying EB upon *fra* and *unc-5* manipulations.

R2#3 •The authors make a direct correlation between a reduction in the fraction of newly produced rECs and impaired migration. While EBs are often found in pairs with ISCs, they are also found without ISCs neighbors throughout the epithelium. Therefore, rECs could also arise from the differentiation of such isolated EBs. It would be important to quantify the fraction of EBs that are without ISC neighbors. Also, what is the effect of *Fra* KD or *Unc-5* OE on the replacement of aECs? Related to this question, is the reduction in rEC replacement intimately linked with *Fra*/*NetB* signaling or would any impairment of EB-to-EC differentiation have relatively bigger effects on the replacement of rECs (compared with aECs)?

The reviewer raises an interesting point here: the majority of EB appear in duplets with an ISC, but EB can also be found isolated without being in contact with an ISC (de Navascues et al 2012). In any case, even single EB have derived from an ISC mitosis and subsequent ISC death, to make them single, is a very rare event (Zipper et al 2022). In the light of our data, we would like to suggest that the existence of single EB can also be taken as prove that these single EB migrate towards rEC in need of replacement. In addition to our data from this manuscript, the findings of integrating migratory EB is also corroborated by previous data from our Lab that in brainbow-clones intermingle and can be found outside their clonal areas (in Fig.2 of (Antonello et al 2015a)).

*To detail on this issue, we also quantified the fraction of singular EB, which lack contact to ISC using our *klu*^{ReDDMCas9} system combined with antibody staining targeting the adherens junction marker *Arm* to unequivocally identify ISC (*GFP*⁺*RFP*⁺*Arm*⁺) and EB (*GFP*⁺*RFP*⁺*Arm*⁻). In line with*

the replacement rate of rEC in homeostasis (Fig.2L) and the number of NetB⁺-EC (please also see later comment R2#9), the fraction of singular EB per total EB is low with an average of 5,1% (Fig.R3). Importantly, EB specific knockout of *fra* significantly reduces the fraction of singular EB to an average of 1.6% (Fig.R3) (but not total EB numbers). This data shows that there is already a reduction visible on single EB numbers when *fra* is depleted (Fig.R3) which logically reduces the number of new rEC (Fig.2L).

Together, this supports our main finding in the paper and we thank the reviewer to point to this issue.

Fig.R3: The fraction of singular EB is reduced upon knockout of *fra*

Quantification of singular EB lacking contact to ISC upon seven days of *klu*^{ReDDMCas9} tracing in controls (*w*¹¹¹⁸) and upon knockout of *fra* (>*fra*^{gRNA}). EB are labelled by *klu*^{ReDDM} tracing (GFP⁺RFP⁺), whereas ISC have been identified by antibody staining targeting the adherence junction marker Arm (GFP⁺RFP⁺Arm⁺). For box plots, the boxes show median, 25th and 75th percentiles, and whiskers indicate the range of values. All data points are shown by dots and means are indicated by light blue lines. 'n' are numbers of biological replicas. Asterisks denote significances from comparison by Mann Whitney test (***) *p*<0.001).

Concerning the last question: The impairment of rEC renewal is always shown in frequencies related to total renewed EC numbers. A lower frequency of rEC renewal upon *fra* KD and *unc-5* OE thus automatically implicates a higher frequency of aEC renewal as well, independent whether the total number of renewed cells is reduced upon *fra* KD and *unc-5* OE (Fig.2L,M). Thus, a general reduction of EC or their renewal does not result in a shifted frequency of aEC and rEC. Together, this supports the hypothesis that a reduction of rEC is linked with NetB/*Fra* signalling.

R2#4 •The authors use correlation rather than direct evidence to conclude that EBs migrate to replace worn-out rECs. To fully support this claim, it would be important to monitor EB migration EBs migrate during homeostasis, to replace worn-out rECs, or alternatively, using their Hamlet assay using live cell microscopy (maybe using the approach in Marchetti et al., 2022).

The reviewer addresses an important point here and we would love to have performed live imaging also following Lucy O'Briens protocols (Koyama et al 2020, Martin et al 2018), that we also mention in the introduction (lines 47-51). The mentioned approach of Marchetti et al. allows live-imaging of explanted midguts for 48 – 72h (Marchetti et al 2022). As stated by Marchetti et al. progenitor cells of explanted midguts are quiescent and also make the observation of a replacement event very unlikely. However, we include new data that further strengthen the connection between NetB expressing EC and EC turnover requiring migration. We previously showed that EC turnover rates in the posterior midguts of adult flies are constant and do not change with age in homeostatic conditions (Antonello et al 2015a).

This is supported by taking a close look at the numbers behind intestinal turnover. Methods like ReDDM show that the average number of renewed EC per R5 region is around 73 EC in 7 days. This indicates a renewal of around 10 EC per day and thus an average EC turnover rate of 0.43 EC per hour. An EC turnover rate of 0.43 indicates a low probability to find a dying EC in need of replacement at a fixed given time point. This also further supports the low number of single NetB expressing EC we detected and show in Fig.5E, S6A,B.

In detail, dissected midguts are always fixed prior to antibody staining and subsequent imaging always shows a 'snapshot' of the actual state of the midgut epithelium. The average EC turnover rate of 0.43 EC per hour fits our observation of 0-4 EC per R5 positive for aNetB staining or NetB> and NetB>> activity. Using aNetB antibodies, we investigated NetB levels in control midguts (Fig.5S-S') and compared them to aNetB levels upon EC specific expression of >E-cad-RNAi (Fig.5T-T') stimulating homeostatic turnover as published by the O'Brien lab (Liang et al 2017). Suggesting a connection between EC death and secretion of NetB, aNetB levels are higher upon >E-cad-RNAi expression affecting the whole EC population (Fig.5U). We added these important datasets to Fig.5 of the revised manuscript and discuss this in the text (lines 269-290).

Apart from this, one could also induce damage but that would cause new issues: i) quiescent progenitor cells are probably even more unlikely to migrate nor replace worn out cells during homeostatic conditions; ii) inducing damage to explanted midguts will stimulate further signalling pathways such as non-canonical Wnt signalling that is induced via Otk (Hu et al 2021). iii) when we shifted the flies for our Hamelin assay, we saw that 72h are not sufficient to induce EB migration across the MHB boundary which argues for a comparably tight controlled and slow process. Together, live-imaging approaches would be a direct and good proof, but it would be very unlikely to catch such a very rare event.

Most importantly, following an experimental suggestion from Reviewer #1, we now provide compelling data that EC indeed attract EB with the release of NetB through Fra receptors (new Fig.6 detailed in R2#10) In addition, the new Fig. 7 shows in Hamelin that NetB-induced migration is absent when Fra is depleted (Fig.7G,H).

R2#5 •The effects on lifespan upon fra or unc-5 KD in EBs are quite modest. Did the authors use backcrossed flies for this experiment?

Yes, flies are always backcrossed to the according background (and displayed experiments are without any balancer chromosomes).

A modest effect of genetic conditions that block rEC renewal can also be explained to a very high tolerance of the midgut to cell loss, which we described previously in ((Zipper et al 2020) in Fig.3 Supplement 1G). As one can see from the attached screenshot from Zipper 2020 on the left, the midgut can suffer a 30% reduction in length and flies are still viable when all EC-production is blocked by N-RNAi (for schematic see Fig.3K). Thus, a substantial loss of rEC over time is necessary to ultimately lead to fly death. In addition, we performed new survival experiments presented in Fig.S2Y now.

We updated the manuscript accordingly (line 165ff).

R2#6 •The authors show that progenitors expressing a dominant negative version of Notch have reduced protrusions, which can be rescued by Fra OE or unc-5 KD. The interpretation of these results is a little hard, as manipulation takes place in both ISCs and EBs. What is the effect of Fra KD or OE or unc-5 KD in ISCs on protrusions in NDN cells?

We thank the reviewer for pointing this out. Upon recommendation of this Reviewer, we now performed new experiments with ISC specific manipulations of fra and unc-5 independent of N-LOF as described in more detail in our answer to comment R2#1(1,2). Investigating the function of both receptors in the differentiation of ISC to EB, we observed an induction of protrusion lengths upon ectopic expression of >fra and included and discuss the new data in Fig.3A-J. Most importantly, both RNAi stocks against fra and unc-5 do not show significant effects using ISC-ReDDM, which further underlines that there is no endogenous role in ISC.

In addition, the Notch LOF model usually results in ISC-like cells (Micchelli & Perrimon 2006, Ohlstein & Spradling 2006, Ohlstein & Spradling 2007, Patel et al 2015, Perdigoto et al 2011) and is often used in the midgut field to show whether factors have EB differentiation properties (Zipper et al 2025, Zipper et al 2020). Having this new dataset (Fig.3A-J), we now included both experiments and more detailed description in the revised manuscript (lines 200-209).

R2#7 •As mentioned previously, the assumption that Fra and unc-5 are exclusively present in EBs needs to be substantiated. Does Fra OE/KD or unc-5 KD affect ISC-to-EB differentiation – this could be done using Transtimer.

We thank the reviewer for this comment and kindly refer to the lines of old and new evidence for EB-specific expression of both receptors presented in R2#1(2) now. Additionally, we now include two experiments in Figure 3 that address this issue. In the initial experiments combining esg> driven >N^{DN} with >fra and >unc-5-RNAi we block the production of EB by >N^{DN} (Micchelli & Perrimon 2006, Ohlstein & Spradling 2007, Patel et al 2015, Zipper et al 2020) and analyse 'ISC-like progenitors' and their transition to 'EB-like' cells reflected by protrusion elongation and size. Quantifications show a reduction in EB-like cells (reduced protrusion lengths and nuclei sizes) upon >N^{DN} compared to esg^{ReDDM} controls (Fig.3P-Q). By additional expression of >fra and >unc-5-RNAi, we induce protrusion lengths and nuclei sizes suggesting a rescue of 'EB-like' cell production (Fig.3P-Q).

In our answer to comment R2#1 and Fig.3A-J of the revised manuscript, we now present additional evidence for EB expression of both receptors and new results of ISC specific depletions of fra and unc-5 as they have been suggested by Reviewer 2. In summary, both depletions show no effect in ISC, which further underlines exclusive expression of fra and unc-5 in EB. Additionally, allowing the speculation that (ectopic) Fra plays a role in ISC to EB differentiation, Fra overexpression results in significantly more numerous new EC in Fig.3J.

Having the three defined time-points for our investigations of both receptors (ISC ReDDM, Notch LOF in ISC-like progenitors and the kluReDDM experiments), we think performing experiments with Transtimer would not provide additional information over ReDDM tracing. We included the new experiments and analysis in Figure 3 and updated the manuscript accordingly (lines 204-259).

R2#8 • It is perplexing that knockout of Net-A, a ligand that is expressed 3- to 4- fold more than Net-B in the midgut, has minimal effects on EB protrusivity, given that Net-A also attracts progenitors across the MHB in the Hamelin assay. Could the authors evaluate the Net-A expression pattern in the midgut?

It is correct that NetA functions equally in Hamelin. Concerning the reviewer's statement: We do not exactly know, where it is published that NetA is expressed 3- to 4- fold higher compared to NetB. What we did to characterize NetA in the midgut was to look at available reporter stocks. First, we analysed an endogenous GFP-tagged NetA to investigate its localization within the adult midgut and found it to be present in visceral muscle (VM) cells (Fig.R4A, (Aghajanian et al 2016)), which we also found using NetA-Gal4 flies. These results explain high expression in midgut samples and led to the hypothesis that NetA is secreted by visceral muscle cells thus having another source than EC-derived NetB (Fig.R4B).

Fig.R4: NetA is expressed in the visceral muscles

(A) Confocal image of GFP-tagged NetA crossed to *how>* specifically driving expression of *>CD8::RFP* within the visceral muscles (VM, *how-Gal4>CD8::RFP* (Aghajanian et al 2016)). (B) Schematic showing NetA ligands within the VM basal of the midgut epithelium encompassing ISC, EB and EC. Midgut epithelium and VM are separated by the basal membrane (BM).

As additional evidence, we performed EC-specific knockdown of NetA using our EC-Rapport system (Fig.R5A-C) similar to NetB in Fig.5L. In contrast to reduced ISC/EB protrusion lengths upon EC specific expression of *>NetB-RNAi* (Fig.5L,Q), we did not observe a reduction in protrusion numbers (Fig.R5D) nor lengths with *>NetA-RNAi* (Fig.R5E). This suggests that in EC, NetA has no role in the non-autonomous induction of EB protrusions.

Continued on next page..

Fig.R5: Depletion of NetA in EC does not affect EB protrusions related to Fig.5

(A-C) Confocal images of midguts (R5) after seven days of EC-Rapport tracing in controls (w^{1118}) (A) and upon knockdown of NetA (B-C). (D-E) Quantification of protrusion numbers (D) and protrusion lengths (E) of EC-Rapport driven depletions of NetA. (D-E) For box plots, the boxes show median, 25th and 75th percentiles, and whiskers indicate the range of values. All data points are shown by dots and means are indicated by light blue lines. 'n' are numbers of biological replicas and 'N' values describe numbers of technical replicas. (D-E) Data have been analyzed using Kruskal Wallis tests, but do not show significant differences.

Although these data suggest that NetA does not confer directionality in the MG, its functional analysis in the VM will be a fascinating topic for future investigations in the lab, where we follow the hypothesis that NetA is involved in maintaining attachment of midgut cells to the basal membrane.

R2#9 •The authors present confocal images to show that NetB is detected in singular ECs – it would be great if these images could be accompanied by a quantification of number of NetB positive ECs in the posterior gut to see how that correlate with replacement events.

The reviewer is right, a correlation of NetB⁺ EC and replacement events would substantially strengthen our model of NetB directed EC replacements. We kindly want to refer to the Reviewer to his remark R2#4 for details.

Briefly, NetB⁺ EC as well as EC replacements are rare events. When correlated, the numbers of NetB⁺-EC and 'to be replaced EC' fit quite well, because we expect less than one EC that is replenished in an hour (0.43EC/h), which is more or less what we found we evaluate NetB⁺ EC per R5 region. We make this clearer in the text now (lines 268-273).

R2#10 •The authors suggest that Net-B is expressed by the worn-out rEBs. How did they confirm that these EBs are worn-out? Could they check if markers of cell death co-localize with ECs that express Net-B? Could the authors perform protrusion direction assay as they did for Fra knockdown (Figure 2k) to see if EC-specific Net-B knockdown or OE randomizes the orientation of protrusion?

We kindly refer the reviewer to our answer to comments R2#3&4 and the new data presented in Fig.5S-U of the revised manuscript, in which we applied a previously published model to stimulate EC turnover (Liang et al 2017). In summary, our new data employing this experimental paradigm now directly shows that E-cad-depleted EC have significantly higher NetB levels (Fig.5S-U).

In detail, dissected midguts are always fixed prior to antibody staining and subsequent imaging always shows a 'snapshot' of the actual state of the midgut epithelium. Using aNetB antibodies, we investigated NetB levels in control midguts (Fig.5S-S') and compared them to aNetB levels upon EC specific expression of >E-cad-RNAi (Fig.5T-T') stimulating homeostatic turnover as published by the O'Brien lab (Liang et al 2017). Suggesting a connection between EC death and secretion of NetB, aNetB levels are higher upon >E-cad-RNAi expression affecting the whole EC population (Fig.5U). We added these important datasets to Fig.5 of the revised manuscript and discuss this in the text (lines 269-290).

We additionally tried to overexpress NetB in EC using our EC-Rapport system but it seems to be cytotoxic, which limits the shift duration and thereby NetB expression to a short period of time. Specifically, we needed to dissect the flies with EC-Rapport driven >NetB after 24h before they die. Unfortunately, after 24h of Rapport tracing, we do not find any replaced EC and cannot analyse the orientation of EB protrusions upon EC specific OE of NetB. However, for chemotactic cells it was already shown that they still polarize and initiate motility when exposed to a uniform concentration of chemoattractants (Wadsworth 2015) like NetB being released from all EC in our Rapport experiments (Fig.5I-X).

Most important to the last issue the reviewer raised here about whether NetB is signalling through Fra: We now provide evidence in Fig.6 that with the clonal expression of >NetB within single EC, we are now able to present data of protrusion directionality upon NetB OE in single EC.

Functionally our new performed Hamelin assays with progenitor specific depletion of Fra with simultaneous NetB expression in the HG show that NetB-induced migration indeed depends on Fra receptors (Fig.7G,H). In detail, we now show a direct effect Fra on EB migration and that Netrin-B signals through Fra in Fig.7. As described in the main text, Hamelin is based on dual expression systems which basically allows an AOP-driven knockdown of fra (>>fra) in midgut progenitors while we express attracting NetB in the hindgut with byn>NetB. Hamelin already consists of a total of six transgenes and two more, >NetB and >>aop-RNAi, needed to be added by recombination of chromosomes. During the revision process, we (finally) managed to perform this experiment by using the same esg>> driver for expression of a fluorophore (>>H2B::mCherry), tub-Gal80^{ts}, and >>fra-RNAi specifically in ISC/EB and combined it with the same byn> as in Hamelin to drive expression of >NetB::GFP (Okada et al 2023) to attract ISC/EB and simultaneously show byn> activity by GFP signal in the HG. After expression of transgenes for seven days like in Hamelin, we could not detect any fra-depleted midgut ISC/EB migrating across the MHB, which suggest that Fra mediates the attractive effect of NetB in our Hamelin assay. We added this important data to Fig.7G-H of the revised manuscript.

We updated Figs.5, 6 and 7 and the sections of the revised manuscript accordingly.

R2#11 • Is NetB required for the increase in protrusions triggered by E-Cad KD in figure 5O?

Yes, during the revision process we got hands on NetB antibodies and measured aNetB intensities in Controls (Fig.5S) and upon EC specific knockdown of E-cad (Fig.5T). Suggesting a connection between EC turnover induced by >E-cad-RNAi and the release of NetB, we found an increase in aNetB intensities in E-cad depleted EC (Fig.5U). This link indicates that NetB is at least involved in the increase in protrusions length triggered by EC specific expression of >E-cad-RNAi.

R2#12 • In Figure 5T, it would be more relevant to examine the effect of EC-specific NetB KD on lifespan

It is correct that a constitutive whole animal mutant does not necessarily correlate with Netrin functions in midgut homeostasis. However, the majority of investigations that focussed on Netrin function investigated neuronal development. As the investigated Net mutant flies develop into living imagos (Brankatschk & Dickson 2006) and phenotypically reproduce our cell-type specific manipulations (Fig.5D, Q), we would like to include the survival data of these well-characterized Netrin mutant flies (Brankatschk & Dickson 2006) (Fig.6K) as well as fra-mutant flies (Kolodziej et al 1996) (Fig.S3N) as they complement our data with established alleles.

Additionally to additionally confirm our kluReDDM survival assays, we now address the possibility of extra-intestinal activity of the drivers and present data showing with a bipartite expression I-KCKT system developed by the Sokol lab that fly survival is also affected when only midgut progenitors are depleted of fra (Buddika et al 2021) (Fig.S2Y). The midgut specific split-Gal4 system (I-KCKT-Gal4) works by restricting Gal4 activity to midgut progenitor cells (ISC/EB) in which a KDR recombinase expressed under control of the midgut specific CG10116 enhancer removes an intervening stop cassette in a progenitor specific, mira-driven Gal4 (Buddika et al 2021). We updated the manuscript and the figures accordingly.

R2#13 • The authors show that OE of NetB from EECs does not trigger migration/clustering of ISCs/EBs around EECs. Does EC-specific OE of NetB trigger protrusions in ISCs/EBs?

Yes, in our newly provided experiment in Figures 5 and 6, we now provide evidence that NetB from single EC attracts protrusions and migration. Please find more details in the revised manuscript and in R2#10&11.

Additionally, we have a dataset that we also added to Fig.5O-R of the revised manuscript supporting this statement. As mentioned in R2#10, expression of >NetB in all EC using mex> is cytotoxic, which is why the flies are dying already after two days. We analysed ISC/EB protrusion numbers and lengths upon EC-specific expression of >NetB after 24h and found an increase in ISC/EB protrusion lengths and numbers (Fig.5O-R, Fig.R6A,C,E). Fig.R6 also shows ectopic expression of NetA (details on the expression of NetA can be found in R2#8), we measured an increase in protrusion lengths upon EC specific ectopic expression of UAS-NetA, whereas protrusion numbers are decreased (Fig.R6A-B,D-E).

But as already pointed out in R2#10, the most compelling data can be found in Fig.6 that shows that NetB triggers protrusions and migration towards EC. We updated the manuscript and figures accordingly.

Fig.R6: Overexpression of Netrins in all EC non-autonomously induces ISC/EB protrusions related to Fig.5

(A-C) Confocal images of midguts (R5) after 24 hours of EC-Rapport tracing in controls (w^{1118}) (A) and upon overexpression of NetA (B) and NetB (C). (D-E) Quantification of protrusion numbers (D) and protrusion lengths (E) of EC-Rapport driven overexpression of Netrins. (D-E) For box plots, the boxes show median, 25th and 75th percentiles, and whiskers indicate the range of values. All data points are shown by dots and means are indicated by light blue lines. 'n' are numbers of biological replicas and 'N' values describe numbers of technical replicas. Asterisks denote significances from multiple comparisons by Kruskal Wallis tests (* $p < 0.05$; ** $p < 0.01$; **** $p < 0.0001$).

Minor comments

R2#14 •Several of the figure items are lacking information regarding the drivers used. It would strongly facilitate the “reading” of the figures if driver information was present in all items.

We thank the reviewer for pointing out this difficulty in extracting all important information from our figures and now include the drivers within the figure panels to simplify understanding of the genotypes the figures.

R2#15 •It would be recommendable to add DAPI channel together with ReDDM confocal micrographs instead of GFP and RFP channels alone. This would help readers to get a better idea of rEC renewal vs aEC renewal.

We tested our figures in both ways and evaluated that the addition of the DAPI channel make the images too complex rather than being helpful.

R2#16 • In lines 105-106, DI+ instead of D1-, Dlg1+ instead of Dlg1- and Pros+ instead of Pros-.

We agree that the used abbreviations can be misleading and changed them in the text.

R2#17 • In supplementary figure S4, the figure labels show esgReDDM instead of Klu ReDDM.

We corrected this issue.

R2#18 • Statistics missing for new rEC quantification for Fra Δ P3 in Figure 4I.

The statistics for the comparison of >fra^{gRNA}; >fra ^{Δ P3} to >fra^{gRNA} are already part of the graph, but did not show a significant result.

R2#19 • Authors state that EBs expressing Fra lacking (line 202) P1 or P2 domain migrate normally and behave like controls. But these conditions look more closer to EBs with overexpression of full-length Fra than control EBs. This reviewer would recommend clarifying this.

The reviewer is right by pointing out an induction of protrusion formation upon expression of >fra ^{Δ P1} and >fra ^{Δ P2}. Fra receptors lacking the P1 or P2 domain are still able to induce protrusion formation and migration, when endogenous Fra is knocked out with >fra^{gRNA}. Compared to >fra^{gRNA} and indicated by statistics in the graph Fig.4I, P1 and P2 domains rescue the formation of new rEC suggesting control-like normal distant rEC replacement.

Neurodevelopmental studies of the Fra P-motifs in Drosophila axon guidance made similar observations and already revealed that expression of fra ^{Δ P1}, fra ^{Δ P2} or the combination fra ^{Δ P1 Δ P2} all rescued the axon guidance defects in a fra mutant background. However, expression of fra ^{Δ P3} does not rescue axon guidance defects. In addition, the same authors observed a significant difference compared to fra mutants upon expression of fra ^{Δ P3} suggesting a mild rescue of attractive function, but at much lower levels compared to fra ^{Δ P1} and fra ^{Δ P2} (Garbe et al 2007).

In our experiments shown in Fig.4 the fold changes in protrusion numbers/length of >fra^{gRNA}; >fra ^{Δ P3} compared to >fra^{gRNA}; >fra ^{Δ P1} and >fra^{gRNA}; >fra ^{Δ P2} are 0.79/0.38 and 0.86/0.40 respectively, which also suggest a mild, but weaker rescue ability of fra ^{Δ P3} compared to fra ^{Δ P1} and fra ^{Δ P2}. We made this clearer in the revised manuscript (lines 251-259).

R2#20 • This reviewer does not agree with the notion in line 186 that “Fra and Unc-5 are not per se sufficient inducers of differentiation in the intestinal homeostatic lineage”, as their expression in EBs are sufficient to trigger EB-to-EC differentiation (Figure 2M). Their results mainly show that N is required downstream or in parallel to Fra and/or Unc-5 to promote EB-to-EC differentiation, while Fra GOF or unc-5 LOF is sufficient to suppress the reduction in protrusions of ISC-like progenitors triggered by N LOF.

The reviewer draws a legitimate concern here: It is true that when expressed in EB overexpression of fra and knockdown of unc-5 both induce EB-to-EC differentiation (Fig.2M). During the revision process and following the suggestions of the Reviewer 2 (R2#1.2), we performed manipulations of Fra and Unc-5 in ISC and observed that ectopic >fra is sufficient to induce production of EB and new EC (progeny, Fig.3J). This induction of EC differentiation however is not observed in Notch depleted ISC/EB (Fig.3N,R) suggesting that EC differentiation induced upon >fra depends on Notch signalling.

When compared to other inducers of differentiation like Eip75B-A/C (Zipper et al 2020) the differentiation stimulus of >fra and >unc-5-RNAi are comparably weak, which led us formulate that Fra and Unc-5 are not sufficient inducers of differentiation per se as stated in the initial manuscript. We agree that this statement can be misleading and changed it in the revised manuscript. Additionally, we adjusted the manuscript and include the new data of ISC specific manipulations of Fra and Unc-5 in a new Fig. 3.

Reviewer #3 (Remarks to the Author):

Proper proliferation of residential adult stem cells and differentiation of their progeny are required to maintain the homeostasis of adult tissues. Adult intestinal epithelium is a good example of tissue homeostasis maintenance. Intestinal stem cells (ISCs) constantly produce differentiating progeny to replace dead differentiated cells, most of the time, enterocytes (ECs). However, how dead ECs are replaced by differentiating progeny (enteroblasts, EBs) is not fully understood. Here, Lisa Zipper and colleagues propose a hypothesis, in which so called adjacent ECs (aECs) and remote ECs (rECs) in intestinal epithelium, that dying ECs could secrete attracting Netrin-B (NetB), in addition to cytokines and EGFs, to bind to its receptor Frazzled (Fra)/DCC on differentiating EBs to promote their migration to the site of dying ECs and replace them in the end. Furthermore, they claim that Fra/DCC also functions in CRC metastasis.

However, there are critical flaws in their hypothesis and proposal: 1) ISCs are almost evenly scattered in intestinal epithelium, thus, most, if not all, ECs are in the one-EC-diameter cycle of ISCs. So intestinal epithelium is composed of (arranged by) thousands of such cycles with ISCs in the center and only a small portion of ECs are out of the boundary of neighboring cycles. 2) Even such so called rECs exist, their distances to neighboring ISCs are different (not equally away from surrounding ISCs). If such ECs are dying and secrete NetB. Here comes to the term of morphogen gradient. The secreted NetB from the dying ECs form a gradient, with the highest concentration of NetB from the producing dying ECs and decreasing its concentration as the distance increases. So why the more distanced ISC/EB bothers to sense the signal and migrates toward the dying ECs (so called rEC by the authors), but not the much more nearby ISC/EB (so called aECs)? So there is paradigm of their hypothesis and is fatal to their hypo. 3) In the experiments carried out, all EBs are labelled and traced, how the authors differentiate and precisely sure that the so called newly produced so called rEC is produced from which EB? As they can not ascertain the true origin of the newly produced so called rECs, how can they claim the newly produced so called rEC is produced and migrated from an ISC far away from the dying EC? 4) Further, even if their proposal is true under normal conditions, in which they claim that NetB from dying rECs attracts EB from far away (here coined remote EB, rEB, following the authors) and the rEB cells migrate to replace the dying rEC to maintain intestinal homeostasis. If Fra, the receptor, is compromised in EBs, no migration is occurred, thus no repair of intestinal epithelium, leading to damaged epithelium. However, on the contrary to their proposal, in the metastasis model the authors established, compromising Fra/DCC pathway increased metastasis of such tumor cells instead. What a surprise! This observation is totally in contrast to their proposal. So one can conclude that their proposal is wrong or with fatal foundation. Altogether, these fundamental weakness points of this manuscript prevent me to recommend it for publication in Nature Communications.

First of all, we want to thank the Reviewer for evaluating our work and his critical comments that improved our revised manuscript with new experiments and discussions. In the beginning of our point-by-point response, we want to briefly address the four issues raised here and refer to our comments to Reviewer's #3 specific concerns where indicated.

1) *Yes, the Reviewer correctly describes that there is a high number of ISC/EB duplets scattered along the midgut epithelium (Micchelli & Perrimon 2006, Ohlstein & Spradling 2006, Ohlstein & Spradling 2007). Here, we address however for the first time the ratio of these cells and present our analysis of rEC vs aEC. Counting manually, roughly 70% are aEC, so according to our hypothesis, no migration for their replacement is necessary. We aimed to illustrate this better in our new Fig.7M now. Fig.1B and Fig.7M shows that there is very small distances to be travelled for EB to reach the remaining roughly 30% rEC, which connects to the next concern about the signal itself:*

- 2) *The Reviewer is asking about how far the NetB-signal travels from dying EC towards EB to attract EB protrusions and subsequently initiate EB migration. In line with our hypothesis, we now include measurements and logical considerations in the new discussion (lines 420-431) based on two new experiments (Fig.6&7) that allow statements to be made on the range of the Netrin gradient. Our measurements in Hamelin (Fig.7H) show that attraction works efficiently for up to 40µm, which is less than necessary to reach the average rEC, which is about 25µm away from EB (Fig.7M). This efficiency is also shown in our new Fig.6 that also reveals that many EB are attracted to one clonal NetB-overexpressing EC (Fig.6I) further supporting our hypothesis. In addition, these distances perfectly align with observations concerning mitogens stimulating ISC division by the O'Brien lab (Liang et al 2017). In the revised manuscript, we include a whole new Fig.6 and Fig.7 with this additional evidence. In addition, we now show additional data for NetB-release from worn-out EC (Fig.5S-U). We kindly ask the Reviewer to find details in comments R3#5,9,22,26 and 31.*
- 3) *It is true that we cannot tell apart which exact EB from a duplet finally replaces a rEC, but this is also not part of our hypothesis. Our data in Fig.1B-F shows that there is roughly 30% rEC that can only be replaced by migration of an EB. Equally important is the fact that the renewal of aEC or rEC occurs with identical frequency (Fig.2E,F). What we observed in our manipulations of NetB-levels in EC is actually supporting that NetB non-selectively attracts many surrounding EB to one clonal NetB-releasing EC (Fig.6I,J). On the level of the individuum and fly survival, we can speculate that this probably reflects a safety mechanism ensuring that the epithelial integrity of the midgut is ensured even at the cost of overproduction of EB (Reiff et al 2019) and their superfluous migration.*
- 4) *The Reviewer is asking about why we do find different effects in our CRC model. Actually, we do not think that there is an actual discrepancy here, as we are looking at a very complex model which includes already alterations of pathways that are known to affect migration. In addition, in the CRC data we observe and measure dispersion into the hemolymph which is not horizontal migration as described in our discovered NetB-dependent EB migration inside the epithelium. Furthermore, LOF of Fra/DCC and Unc-5 behave identically in human patients and worsen the outcome. Please find details concerning this and why we removed this dataset for stringency of the revised manuscript in our comment R3#16.*

Other major or minor concerns (from the beginning of the MS to the end):

R3#1. the title of this MS is "Frazzled/DCC directs intestinal progenitor homing in homeostasis and metastasis", the authors used the term homing, which caused a big confusion. As all of us know, home normally refers to where a person or something is originally born or should often resides (like homing of transplanted blood stem cells). In the case of differentiating EBs (they called EBs as progenitor which is also misleading), EBs are produced by ISCs and will migrate a little bit of distance and differentiate into ECs. So where is the home of EB in the eyes of the authors? The place of EB's birth: nearby or in close contact with ISC or the location where it is fully differentiated?

We agree with the reviewer and changed to a new title of our manuscript:

'Frazzled/DCC directs spatial progenitor integration ensuring steady-state intestinal turnover'

We do employ the term progenitor here as we think it is more suitable than enteroblast for the broad readership of Nature Communications.

R3#2. the writing of the MS is problematic, there are many grammar and usage errors. For example, mis-usage of singular and plural forms is frequently observed throughout the MS: lines 15-16: worn-out EC express.....; line 20: our data establishes.....; line 21: provides.....; line 28: ..(PMG) prove... And so on and on. Terms such as Notch-signaling, Delta-ligands are not properly written. Punctuations are often missing, such as line 115.

We double-checked and corrected all examples for wrong singular and plural forms throughout the manuscript.

The abbreviation 'EC' is both, singular and plural of enterocyte(s), the same is true for the word 'data', which we also used in our previous Nature Communications publication (Zipper et al 2025).

Line 28 'proves' was also corrected in the revised manuscript. Additionally, we changed the terms 'Notch-signaling' and 'Delta-ligands' mentioned by the reviewer and corrected all punctuation errors.

R3#3. The authors should provide clear evidence to show that the so called rECs do exist in midgut epithelium, using membrane labelling and cell identity markers.

We agree with the reviewer that it is important to identify rEC and aEC when quantifying. As rEC and aEC are definend by us by their contact or not to ISC/EB, there is not distinct molecular marker. In Fig.1D and Fig.5V, we show exemplary images in which we included the established EC marker discs large 1 (Dlg1) channel (Chen et al 2018, Woods & Bryant 1991), which was basis of all of our quantifications for aEC, rEC and total EC numbers.

R3#4. Fig. 1G, the authors should indicate the way to differentially label ISC+EB and EB (lexA and Gal4).

In Fig.1G, we show a cartoon explaining how we differentially label ISC&EB and EB. esg-lexA is active in both, ISC&EB, and marks both cell types with Aop-CD8::GFP (green membranes) (Zipper et al 2025). On the other hand, klu-Gal4 is specifically active in EB and only marks EB with UAS-H2B::RFP (red nuclei) (Korzelius et al 2019, Reiff et al 2019). ISC&EB are both labelled with GFP and only EB additionally show RFP signal within nuclei as indicated in the cartoon (Fig.1G). This allows their unequivocal identification based on published markers (lines 88-91).

R3#5. lines 80-87, how the authors claim that the so called aECs and rECs are replaced at identical frequency? This is in contrast to common sense. How they differentiate the origin of the so called rECs? They should perform MARCM clones to examine this.

The frequency of aEC and rEC replenishment is quantified and shown in Fig.1F. For a quantification of rEC numbers, it is not necessary to know exactly from which surrounding ISC/EB duplet the replacing EB come. We describe them to be located distant from ISC/EB and thus called them 'remote EC' as shown in Fig.1B,D.

MARCM clones suggested by the reviewer would not add further evidence in this scenario and clone induction by heat leads to heightened renewal Fig.S1 in (Antonello et al 2015a), which is why methods like ReDDM and esgFlpout by the Edgar lab are advisable. As we experienced in

a previous publication, clones reveal ISC progeny intermingling in Brainbow clones (Antonello et al 2015a), but are not suitable for exhaustive quantitative analysis for migration.

R3#6. line 103-6, the labelling of cell identity is misleading, not in ISC (DI-), EC (Dlg-)....

We thank the reviewer for pointing out the misleading description. We corrected the description within the revised manuscript.

R3#7. line 139, conclusion from Fig. 2J-K, authors' previous data showed that depletion of fra abolished protrusion formation, while here the authors concluded that randomized protrusion formation upon fra depletion. Without protrusion formation in the absence of fra, how to randomize?

We agree with the reviewer that it might be confusing to show a reduction in protrusion formation, but also data for directionality of protrusions for the same conditions. As also shown in Fig.S2A and Fig2H >fra-RNAi does not completely abolish protrusions. We analysed the remaining protrusions in terms of directionality. The randomization was concerning whether they point to an old or new EC.

R3#8. lines 146-8, depleting fra caused early death of flies, and they claimed that this is caused by the inability to repair. However, under infection conditions in which ECs are rapidly being renewed, the infected flies are also short lived. The authors should not draw conclusions favoring their claims. The cause of fra depletion may not be related to EC repair at all!

This is true, there might be multiple causes for short survival including infection. In this manuscript, we address the role of NetB-Fra/DCC signalling in homeostatic turnover, which is very different from infection. During the revision, we now additionally address the possibility of extra-intestinal activity of our used drivers (kluReDDM), which is why we now present data showing with a bipartite expression I-KCKT system developed by the Sokol lab that fly survival is affected the same way when only midgut progenitors are depleted of fra (Buddika et al 2021) (Fig.S2Y). Briefly, the midgut specific split-Gal4 system (I-KCKT-Gal4) works by restricting Gal4 activity to midgut progenitor cells (ISC/EB) in which a KDR recombinase expressed under control of the midgut specific CG10116 enhancer removes an intervening stop cassette in a progenitor specific, mira-driven Gal4 (Buddika et al 2021).

G

activity to midgut progenitor cells (ISC/EB) in which a KDR recombinase expressed under control of the midgut specific CG10116 enhancer removes an intervening stop cassette in a progenitor specific, mira-driven Gal4 (Buddika et al 2021).

Reviewer 3 might also be intrigued by the relatively modest effects of Fra-depletion on survival: We have previously shown a very high tolerance of the midgut to cell loss, which we described previously in ((Zipper et al 2020) in Fig.3 Supplement 1G). As one can see from the attached screenshot from Zipper 2020 on the left, the midgut can suffer a 30% reduction in length and flies are still viable when all EC-production is

blocked by N-RNAi (for a cartoon see Fig.3K of the manuscript). Thus, a substantial loss of rEC over time is necessary to ultimately lead to fly death. Together, our data shows now in two paradigms, that under conditions of lack of rEC replacement, survival is strongly reduced. We updated the manuscript accordingly (line 165ff).

R3#9. the authors should use MARCM clone technique to show the migration of EBs toward the so called rECs and how fra mutation affects this. As the authors concern the background mutations on fra mutant chromosome, they should generate new fra mutant with CRISPR/Cas9 tech and examine their MARCM clones to unambiguously show whether EBs migrate (or the clones grow) toward the so called dying rECs (judged by NetB expression) in control and blockage of this process by fra mutation.

Concerning the migration of EB, in our revised manuscript, we now present clone data that answers this concern directly (in Fig.6). Briefly, Flp-out mediated expression of >NetB in single EC clones shows increased protrusion formation and directionality of EB migration towards >NetB expressing clones (Fig.6), which is abolished when Fra is depleted (Fig.7G,H).

*In detail from the technical side: We performed Flp-out clones induced by heat-shock driven flippase (Pignoni & Zipursky 1997), which are positively labelled by Gal4 driving >RFP and additionally are able to express >NetB. In combination with *lexA-Aop*-based *esg>>CD8::GFP,tub-Gal80^{ts}* this enabled us to label ISC/EB and protrusions as well. Together, >NetB is ectopically expressed in clonal EC that might occur in groups or as single >NetB expressing EC and simultaneously, ISC/EB are labelled by *esg>>CD8::GFP* independently of Gal4/UAS (Fig.6A-C).*

Analysing these clones and ISC/EB behaviour, we were able to show that ISC/EB protrusions project towards RFP⁺ >NetB expressing clones despite their sometime remote location from the ISC/EB (Fig.6C). We quantified protrusion numbers and lengths in ISC/EB located close to >NetB expressing clones and found them to have more numerous (Fig.6E) and longer protrusions (Fig.6D). Furthermore, this method allowed us to analyse the angle of protrusions regarding the direction of a >NetB expressing clone compared to control clones. An angle of 0° means direct targeting of a protrusion towards a clone (Fig.6F,G). By analysing hundreds of protrusions, we could show that the angles for protrusions of ISC/EB located close to a >NetB expressing clone are significantly smaller compared to the ones of ISC/EB located close to control clones (Fig.6F-G), further underlining that protrusions are directed towards NetB sources. Furthermore, we found accumulation of GFP⁺ ISC/EB along the membranes of RFP⁺ >NetB expressing EC clones (Fig.6H-J).

These observations indicate that ISC/EB are attracted by >NetB expressing EC clones and start forming protrusions pointing towards the NetB source and migrate towards the clone. We added these important data to Fig.6 and lines 291-299 and 318-325 of the revised manuscript.

*In addition, we now show that progenitor specific depletion of Fra with simultaneous NetB expression in the HG abolishes NetB-induced migration of progenitors (Fig.7G,H). In detail, we show a direct effect Fra on EB migration and that Netrin-B signals through Fra in Fig.7. As described in the main text, Hamelin is based on dual expression systems which basically allows an AOP-driven knockdown of fra (>>fra) in midgut progenitors while we express attracting NetB in the hindgut with *byn>NetB*. Hamelin already consists of a total of six transgenes and two more, >NetB and >>aop-RNAi, needed to be added by recombination of chromosomes. During the revision process, we (finally) managed to perform this experiment by using the same *esg>>* driver for expression of a fluorophore (>>H2B::mCherry), *tub-Gal80^{ts}*, and >>fra-RNAi specifically in ISC/EB and combined it with the same *byn>* as in Hamelin to drive expression of >NetB::GFP (Okada et al 2023) to attract ISC/EB and simultaneously show *byn>* activity by GFP signal in the HG. After expression of transgenes for seven days like in Hamelin, we could not detect any fra-depleted midgut ISC/EB migrating across the MHB, which suggest that Fra*

mediates the attractive effect of NetB in our Hamelin assay. We added this important data to Fig.7G-H of the revised manuscript.

Concerning the MARCM clones, we kindly refer the reviewer to comment R3#5.

Concerning fra Cas9 mutants, these experiments are part of the manuscript in Figures Fig.2F,H, Fig.4B-I and S2P,Q,S,T and phenocopy our findings with RNAi lines (Fig.2B,H,J,L) concerning new rEC formation (lines 122ff).

Together, evidence concerning the last issue is now included in Fig.6 and Fig.7, where we show that MG progenitors migrate towards clonal NetB expressing EC and that >>fra-RNAi abolishes migration in Hamelin (Fig.7G,H). The revised manuscript includes new descriptions for both figures as well as a discussion of this datasets (lines 493-523).

R3#10. lines 172-180, the authors should not express NDN to mimic Notch LOF, as NDN is not complete N LOF. Furthermore, the cells on the outside boundary of the esg clusters are still differentiating. They should use mutants of Notch pathway component to generate MARCM clones, such as FRT42D-mam and FRT42D-mam, fra (new mutant generated on mam background) to draw a more convincing conclusion.

In our previous work, we carefully evaluated the use of N^{DN} and Notch alleles (N55e11) (Zipper et al 2020). In our hands and combined with esg^{RedDM} tracing that allows quantitative analysis, expression of $>N^{DN}$ leads to more penetrant phenotypes compared to $>N$ -RNAi and N-mutant MARCM clones (Zipper et al 2020), which both allows few newly differentiated EC to be formed (Patel et al 2015, Zipper et al 2020).

R3#11. lines 190-206, it's already known that these protrusions are mediated by actin cytoskeleton, so why bother again?

The reviewer is right with pointing to the known role of actin cytoskeleton in formation of the cellular protrusions in axons. Our experiments in the adult midgut confirm findings from these neurodevelopmental studies. We consider them of interest as e.g. genes like trio are involved in physiology, injury and CRC development, which is why we decided to show the data. What is new concerning the intestinal homeostasis is, that the impairment of EB migration and rEC renewal is not only caused by disruptions of Netrin signalling via Fra/DCC and Unc-5 but also by disruptions in actin cytoskeleton and protrusion formation, which is also confirmed in the mammalian intestine when mice mutant for Arpc3b were investigated (Krndija et al 2019). We thus think this is an intriguing finding adding to the general relevance of our data.

R3#12. line 213 and related lines, the labelling of NetAB-/-NetA- and NetAB-/-NetB- is incorrect and misleading, it should be NetA-/-NetB-/+ (only the former is showed).

We thank the reviewer for pointing out the incorrect labelling of Netrin mutant genotypes and corrected them in the revised manuscript.

R3#13. lines 228-239, the authors should clarify what types of injury, infection-induced? Drug-induced or mechanical damage?

We agree with the reviewer and added information about the type of injury within the text of the revised manuscript (lines 300-314). A description of the experimental procedure of mechanical injury can be found in Materials and Methods and was previously described in (Antonello et al 2015a)

R3#14. line 260, Gal80ts can not control *lexA* activity.

It is true, that in its original version, *lexA* cannot be repressed by Gal80^{ts}. However here we used a modified *lexA* version, which includes the transactivation domain of Gal4 and thus can be repressed by Gal80^{ts} (Yagi et al 2010). We successfully adapted this modified *lexA* version already in our previous publication introducing Rapport (Zipper et al 2025) and here in the present paper it is used for all *esg-lexA* (*esg>>*) experiments. The corresponding reference for the modified *lexA*-driver can be found in line 260 of the original manuscript (line 340 of the revised manuscript).

R3#15. lines 277-281, the authors should use markers to mark the midgut/hindgut boundary, otherwise, it is misleading to draw conclusions.

The reviewer raises an important point here and we made sure from the beginning on that we identify the MHB unequivocally in our images and specifically manipulate with *byn* (*brachyenteron*) (Singer et al 1996). Its driver line *byn-Gal4* was described as specific driver for the hindgut proliferative zone (HPZ) marking the most anterior part of the HG by the Hartenstein lab (Fig1 of Nature (Takashima et al 2008). In this publication the authors show in Fig.1n that the HPZ is subdivided and that the very first, most anterior, line is a ring of *Wg*-positive cells (anterior SCZ-spindle cell zone). By showing that this *Wg*⁺-ring overlaps with *byn>* (Fig.R7), we made sure that we are correctly identifying the MHB with *byn>H2B::RFP* as described in (Takashima et al 2008).

Hamelin

Fig.R7: *Byn>H2B::RFP* is labelling HG cells and overlaps with *aWg*

(A-A') Confocal images of midguts after 24 hours of Hamelin tracing. Hindgut cells are labelled by *byn>H2B::RFP*, midgut progenitor cells are labelled by *esg>>CD8::GFP*. Confirming an unequivocal labelling of the midgut-hindgut boundary, antibody staining targeting *Wg* overlaps with the boundary of *byn>H2B::RFP* signal.

For better visibility that MG progenitors are indeed invading the *byn-Gal4*⁺ area, we now include an inset image (*RFP* for *byn>* and *GFP* for *esg>>*) for *>NetB* in Fig.7D' that shows *esg*⁺ can clearly be found posterior to *byn>RFP* labelled HG cells. In addition, we now provide evidence that MG progenitor cells indeed invade and integrate into the HG by showing *esg>>H2B::RFP*-traced MG cells in the HG. Here, we used *Lachesin::GFP* marker labelling

pleated septate junctions in ectodermal derived epithelia like the HG that is absent in the MG (Fig.7L-L') (Lane & Skaer 1980, Llimargas et al 2004, Tepass & Hartenstein 1994). We updated the manuscript accordingly (lines 366-380).

R3#16. lines 324-6, how the authors explain the discrepancy in the CRC model and under physiological conditions? These results are contradictory to their former proposal.

We agree with the reviewer concerning this and kindly refer the reviewer to our general response in the beginning of this letter. In the following comment, we however want to shed light on the complexities of the findings presented in Fig.7 of the original manuscript, even though they are not part of the revised manuscript.

Concerning our findings, we could speculate on many different possibilities here. Unlike the physiology described in Fig.1-6 of the first submission, our CRC model analysed in Fig.7F (of the original manuscript) mimics a pathological condition specifically for gastrointestinal cancer that is already very pleiotropic even without further Fra/Unc-5 manipulations. Our CRC-model used to obtain the data shown in Fig.7F (of the original manuscript) includes expression of oncogenic >Ras^{G12V} (Fig.7B of the original manuscript). The expression of this activated Ras in turn activates several downstream signalling pathways encompassing PI3K/Akt/mTOR pathway which was already shown to stimulate cellular migration in neutrophils (Lin et al 2024, Pal et al 2023). It is expected that >Ras^{G12V} also stimulates tumour cell migration independent of Fra/DCC and Unc-5 in our Drosophila CRC model.

Interestingly our finding that Unc-5 and Frazzled RNAi have similar effects in this model, is underlined by the mutational situation in CRC patients (Bernet et al 2007, Fearon et al 1990, Kinzler & Vogelstein 1996b, Shibata et al 1996a, Shin et al 2007). In flies, our results of fra- and unc-5-RNAi both stimulate cell dispersion in a CRC model. In lines 291 – 298 of the original manuscript, we describe the observations of DCC LOH and UNC-5C LOF occurring more often in later stages of CRC and correlate with CRC spreading to multiple metastatic sites (Fig.7A of the original manuscript) which is in accordance with the patient's findings.

Together, our model reflects what happens in late stage CRC patients with very pleiotropic effects. Thus, we can only speculate that when the model is combined with >fra-RNAi and >unc-5-RNAi these migrating tumour cells increasingly lose the information (tissue identity, 'context') of where they belong or should migrate to and eventually disperse from the primary tumour in the midgut. In addition to the discussed involvement of Fra/DCC and Unc-5 in MET, there is also data from rodent models that involves both genes as 'dependence receptors' controlling apoptosis (Castets et al 2011, Grady 2013). Although, we did observe indication for apoptosis upon Fra/DCC and Unc-5 manipulations in physiology, this might very well play a part in pathology and apoptosis also facilitating metastatic behaviour. Revealing the underlying mechanisms is a current topic in the lab and an equally interesting and challenging topic of our ongoing research.

R3#17. how can w1118 be used as control? The authors should specify in what experiments, it was used as control, while what is the control in the other experiments. The authors should not just simply use the term control, they should specify it in every control panel as different genetic backgrounds affect the phenotype, thereby affecting the conclusions drawn.

w¹¹¹⁸ is the correct genetic background to be used in our experiments and we are very well aware that genetic background can alter findings. Fly lines containing UAS-transgenes were already (or have been crossed into w¹¹¹⁸ background) and it is also the background of the applied tracing systems including esg^{ReDDM}, klu^{ReDDM}, EC-Rapport, EE-Rapport and Hamelin.

In our previous Nature Communication publication (Zipper et al 2025), we were advised during the reviewing process to change 'w¹¹¹⁸' to 'Control' in our figures. We also favour this as it facilitates understanding of our data for the broad readership of Nature Communications that are not experts in Drosophila genetics.

R3#18. Fig. 1B, in addition to the model, the authors should present representative images of midgut epithelium.

Representative images of the midgut epithelium with traced klu^{ReDDM} guts presenting aEC and rEC are shown in Fig.1C-D respectively next to the model (Fig.1B).

R3#19. The authors should specify which region of midgut is examined, in text and Figs.

We agree with the reviewer that it is important to specify which region of the midgut was analysed as the midgut can be divided into five major compartments that were already shown to differ transcriptionally, functionally, and morphologically and also present distinct ISC behaviour (Buchon et al 2013, Dutta et al 2015, Marianes & Spradling 2013). In all our present experiments, we examined the R5 region of the posterior midgut of mated as it is known to show the most constant turnover under homeostasis (Antonello et al 2015a, Antonello et al 2015b, Reiff et al 2015, Zipper et al 2025, Zipper et al 2020). We now added this information to the revised manuscript main text and the figure legends.

R3#20. Fig. 2B. how ectopic expression of fra promote more and longer protrusions and more ECs are produced? Does its ectopic expression also promote ISC division? It seems to be so. How?

The reviewer points to an interesting observation of >Fra overexpression in EB that increases the number of newly generated progeny (Fig.2B,M). This issue was also raised by Reviewers 1& 2. During development a role for Fra was shown to stimulate MET (Pert et al 2015). In the revised manuscript, we now made this clearer in the text and discuss the role of Fra in MET longer now (lines 202-206 and lines 381—399 of the discussion).

Concerning direct assessment whether >fra in klu^{ReDDM} promotes ISC division, we performed experiments during the revision and stained for the mitosis marker pH3 (Fig.R8A-B, white arrows). When we analysed the number of EB (Fig.R8C) and pH3-positive cells (Fig.R8D) we found mild, but significant increases upon EB-specific expression of >fra. This strongly suggests that differentiation of EB is able to non-autonomously promote ISC proliferation as described previously (Reiff et al 2019).

Fig.R8: Fra expressed in EB non-autonomously induces ISC proliferation

(A-B) Confocal images of midguts (R5) after seven days of *klu^{ReDDM}* tracing in controls (*w¹¹¹⁸*) (A) and upon overexpression of *fra* (B). Mitotic active ISCs are labelled by antibody staining targeting pH3⁺ (white arrows). (C-D) Quantification of EB numbers (C) and mitotic active ISCs (pH3⁺, D) upon *klu^{ReDDM}* specific expression of *>fra*. (C-D) For box plots, the boxes show median, 25th and 75th percentiles, and whiskers indicate the range of values. All data points are shown by dots and means are indicated by light blue lines. 'n' are numbers of biological replicas. Asterisks denote significances from comparisons by Mann Whitney tests (**p*<0.05; ***p*<0.01).

R3#21. As EBs are accumulated upon fra knockdown, does Fra knockdown inhibit EC differentiation or promote ISC proliferation as well?

The Reviewer is correct about asking whether Fra-depletions promotes ISC mitosis, but we did not show an accumulation of EB upon fra knockdown. Perhaps, the reviewer refers to occasional EB clusters visible in the microscope image shown in Fig.2C. However, EB clusters appear also in controls with duplets and a triplet of EB being visible in Fig.2A and were also described previously (de Navascues et al 2012). To exclude that there is accumulation of EB upon knockdown of fra, we analysed EB numbers of >fra-RNAi and controls and observed no significant difference (Fig.R9E). Additionally, we investigated the number of mitotic active ISC marked by pH3 antibody upon *klu^{ReDDM}* specific knockdown of fra (Fig.R9A-D, white arrows) and detected no significant changes when compared to controls (Fig.R9F). Indeed, our data in Fig.2M shows that fra knockdown decreases numbers of renewed EC using *klu^{ReDDM}* (Fig.2M).

Together, this indicates no non-autonomous proliferation stimulus on the ISC from Fra-depleted EB.

Fig.R9: Fra depletion in EB is not affecting ISC proliferation and EB numbers

(A-D) Confocal images of midguts (R5) after seven days of *klu^{ReDDM}* tracing in controls (*w¹¹¹⁸*) (A) and upon knockdown of fra using three different RNAi lines (B-D). Mitotic active ISC are labelled by antibody staining targeting pH3⁺ (white arrows). (E-F) Quantification of EB numbers (E) and mitotic active ISC (pH3⁺, F) upon *klu^{ReDDM}* specific depletions of >fra. (E-F) For box plots, the boxes show median, 25th and 75th percentiles, and whiskers indicate the range of values. All data points are shown by dots and means are indicated by light blue lines. 'n' are numbers of biological replicas. Data have been analyzed using Kruskal Wallis tests, but do not show significant differences.

R3#22. Fig. 2I etc, the direction of protrusion showed in the image is not as claimed by the authors, in control, the newly formed ECs are not always face a protrusion. And upon fra knockdown, almost all ECs are surrounded by progenitors (incl. ISC), so does in control.

The images shown in Fig.2I-J visualize the directionality of exemplary EB protrusions pointing to either old or new EC and the images serve to underline that both directional possibilities exist. Quantification of dozens of protrusions revealed that control EB protrusions point more often towards old EC (Fig.2K), even though around 20% also point towards new EC as also shown by the red arrow in Fig.2I. If we are allowed to speculate here: we assume EB protrusions to be highly dynamic structures as described for ISC-protrusions (Hu et al 2021) and axonal growth cones. Thus, it is tempting to interpret our findings in Fig.2K as 'haptotactile checking' of EB protrusions for NetB. Indeed, our new NetB-expressing EC clones underline this idea by directing protrusions and migration towards NetB⁺-EC (Fig.6).

The reviewer is also correct with his observation that in Fig.2I-J for control and fra knockdown almost all EC are surrounded by progenitors. As we quantified in Fig.1E, the majority of EC (70%) does have direct contact to progenitor cells. In the images shown in Fig.2I-J our main focus was to show EB with protrusions that are located in between old and new EC and decided to only show small, magnified image sections to allow the reader to properly see the directionality of EB protrusions to better understand the quantification of protrusion directionality (Fig.2K).

R3#23. Fig. 2N. how you explain o/E fra also reduced life span? And LOF of fra. Esg/klu in not only express in gut, it also expresses in other tissues. So the cause of life span shortening is complex. The authors should be very cautious on this.

The reviewer raises a critical point here that we took very serious as also the fra-mutant survival logically affects extra-intestinal cells, which is why we now provide an additional survival assay. To exclude extra-intestinal effects, we repeated the survival experiments with fra-manipulations (Fig.S2Y) using the midgut specific, bipartite I-KCKT system (Buddika et al 2021) that we also commented on in R3#8. We include these new findings in the revised manuscript (lines 163-167).

R3#24. Fig. 3, the authors should use null mutants (RNAi knockdown) of key components of the Notch pathway to re-perform the experiments.

We kindly refer the reviewer to comment R3#10 here in which we discuss the different ways to induce Notch-LOF and new experiments.

R3#25. Fig 5A-C, why more ECs are accumulated in 5C compared to 5A and B?

The reviewer is right, that the original image suggests a higher density of EC. We had a closer look again at our scans of the different Netrin mutant backgrounds shown in Fig.5A-C, and could not connect the NetA^{-/-},NetB^{-/-} background to higher EC numbers.

Of note, we additionally checked our images of EC-Rapport>Net-B RNAi and did not observe higher EC numbers upon specific knockdown NetB in EC (Fig.5W,X). We thank the reviewer for this comment and substituted the image shown in Fig.5C with a better representing one.

R3#26. Fig. 6. The boundary of MG and HG should be labelled by a marker, such as Wg-lacZ etc, otherwise, when artificially draw the boundary is mis-leading. Fig. 6E, why no HA signal is seen in the MG? a gradient of HA should be seen if it is secreted from HG. Again, if NetB-Fra could promote protrusion in EBs and their migration toward the source of NetB producing cells, then why no such long protrusions toward HG are seen in Fig. 6D, E, G, and H? According to the authors' model, they should producing HG facing protrusions. However, in most of these so called migrating cells, the protrusions are barely observed, neither in the presence of NetB overexpression (Fig. 6D-H). Without these protrusions, how progenitors migrate and cross the boundary? The aim of Fig.6 is to strengthen their proposal that NetB-Fra attraction, however, this observation is in contradictory to their proposal.

Concerning the migration of midgut progenitors into the HG area, we kindly want to refer Reviewer 3 here to his comment R3#15, where we show Wg-protein together with Byn>H2B::RFP.

Concerning the HA signal in Fig.7F, so far it is not known whether human NTN5 is secreted or not (Yamagishi et al 2015), but our expression of NetB::GFP (Okada et al 2023) clearly shows that puncta cross the MHB (Fig7I). NTN5::HA might also be below detection limits of the LSM710 or be detected in a haptotactile way from the neighbouring HG cells at the MHB by MG progenitors.

Concerning the formation of protrusions and progenitor migration towards NetB-sources, we obtained more direct, detailed and convincing evidence during the revision process that now clearly shows protrusion projection with an analysis of angles and migration towards NetB-sources in Fig.6. We also provide evidence now in Hamelin, that NetB from the HG signals through Fra on MG progenitors (Fig.7G,H). For further details on analysis and experiment for Figures 6 & 7, we kindly refer the Reviewer to the above comment R3#9.

Concerning the protrusions projecting towards the hindgut, one has to keep in mind that there is always a fraction of midgut progenitors that do not respond to Netrins and remain in the midgut even after 7d. We can only speculate that these cells are ISC and thus do not possess Fra according to our findings (Fig.S1, 3). In line with this, Fra-depletion of MG progenitors does not show NetB puncta (Fig.7J), which would render them insensitive to NetB.

The fact that progenitors once crossed the MHB do not possess large protrusions can be explained as well. All invaded GFP⁺-midgut progenitors receive e.g. NetB (Fig.7D) in a very near distance from all surrounding HG cells (byn>NetB), which does not require any more protrusion formation. This hypothesis is also supported by our new findings presented in Fig. 6I, where protrusions are absent once the NetB⁺-EC is reached.

Together, based on our presented data in the new Fig.6 and new Fig.7, we present now compelling evidence for an attraction by NetB from EC through Fra-receptors on EB.

R3#27. Fig. 7G. Besides to the contradiction to their proposal, the authors should be very cautious about the cause of early death in these manipulated flies. to draw a conclusion that the shortening of life span is due to CRC, they should inhibit the activity of klu in other tissues, only letting it active in MG.

Even though, we removed the original Fig.7 now from the revised manuscript, we want to comment here: For the survival assay performed with CRC avatar flies shown in the original Fig.7G, we did not use klu^{ReDDM}, but a midgut specific split-Gal4 system (I-KCKT-Gal4). Gal4

activity is restricted to midgut progenitor cells (ISC/EB) in which a KDR recombinase expressed under control of the midgut specific CG10116 enhancer removes an intervening stop cassette in a progenitor specific, *mira*-driven *Gal4* (Buddika et al 2021). Thus, CRC is only arising in midgut progenitors and allow the conclusion that the CRC is the cause of early death of these flies.

R3#28. Fig. S1D shows that *Fra* is also expressed in EE, not as claimed by the authors in EBs. Meanwhile, as *fra* and *unc5* are both expressed in EBs and the authors propose that they function in the opposite manner, how EBs are coordinated and *Fra* is dominant over *unc5*?

This is a concern that Reviewers 1+2 also shared. In our new manuscript, we now include new data, additions to text and discussion (lines 401-431) concerning this. In the following, we sum up our lines of evidence why Fra signalling controls EB protrusions and migration and why internalization of Unc-5 allows Fra signalling.

It is generally accepted in reviews of Fra and Unc-5 in neuronal development that Unc-5 antagonizes Fra/DCC receptors by acting repulsive in heterodimers with Fra/DCC, whereas homodimers of Fra or DCC on the membrane act attractive (O'Donnell et al 2009). Mechanistically, we have evidence that either Unc-5 depletion or internalization allows Fra/DCC to signal as homodimer and direct migration of EB. Our data on the role of Unc-5 in EB migration supports previous findings from neurodevelopmental studies and point to a subordinate role below Fra/DCC:

- i) EB specific manipulations of *unc-5* show opposing effects to Fra on protrusion formation (Fig.2H,L, Fig.S2A).*
- ii) In addition, rEC renewal (Fig.2L) works normal when Unc-5 is depleted from EB suggesting Fra homodimers direct EB movement.*
- iii) Unc-5 OE in EB however reduces rEC formation (Fig.2L) and fly survival (Fig.2O) by preventing Fra signalling through its abundance.*
- iv) Internalization of Unc-5 into vesicles is described during axon guidance and indicates attractive signalling via Fra/DCC homodimers (Boyer & Gupton 2018). In midgut EB, internalisation of Unc-5 into vesicles with GFP tagged Unc-5 (*unc-5::GFP*, Fig.S1E-H') and new antibody stainings targeting Unc-5 (new Fig.S1E) is observed as well, which in turn suggests signalling through Fra/DCC homodimers.*
- v) Importantly, only homodimers of Fra/DCC were shown to connect to the F-Actin network allowing protrusion formation (Garbe et al 2007, Golenkina et al 2018).*
- vi) In vitro experiments on rat neurons showed removal of UNC5H1 from the cell surface of neurons and growth cones but leaves DCC at the cell surface. These changes of receptor levels at the cell surface could lead to a switch from UNC5H1 mediated repulsion to DCC induced attraction towards a Netrin-1 gradient (Williams et al 2003).*
- vii) As mentioned previously, membrane-bound Netrins to DCC indicate active signalling(Lai Wing Sun et al 2011), which we observe in Hamelin (Fig.7I) and discover in our new experiments that these puncta depend on the presence of Fra-receptors (Fig.7J).*
- viii) Additionally, commented in the next comment (R3#29), the Fra overexpression phenotype prevails when Unc-5 is also overexpressed.*

Together, our data suggests that the model for Fra/DCC and Unc-5 signalling regulation from neurodevelopmental studies also holds true for midgut EB. Presence of Unc-5 (by overexpression) on the membrane leads to repulsive heterodimers (Fra/Unc-5) and depletion

as well as internalization allows Fra homodimers to direct migration. We made these findings clearer in the text now and added this topic to the new discussion now (lines 401-431).

Concerning the GFP puncta, laser settings for Fig.S1B-C and S1D are identical showing that *fra::GFP* is close to no detection in Fig.S1D.

R3#29. Fig. S3B. no so called protrusions seen, but with more newly produced ECs. How come?

Indeed, the double overexpression of >fra and >unc-5 results in an expectedly complex phenotype (Fig.S3B). Underlining our conclusion that Unc-5 is subordinated to Fra, the fra-OE phenotype prevails when >unc-5 is coexpressed and an increase in newly differentiated EC observed by Reviewer 3 is prominent (Fig.S3B).

When we directly compare the protrusion lengths (13.3 μ m) of sole fra overexpression (Fig.2B,H) with the length (9.5 μ m) of combined OE of fra and unc-5 (Fig.S3B,E), it is significant that the increase in protrusion lengths is less in the combined OE of both receptors. Compared to controls (7.3 μ m) however, protrusion length and number is significantly increased (Fig.S3E,F). We also included this in the discussion of the revised manuscript now.

R3#30. Fig. S3H, where is the protrusion in cells in the regenerated region?

In the regenerated region shown in Fig.S3H, we now included a white arrow pointing to a tiny protrusion, whereas an arrowhead indicates a longer protrusion in non-regenerated areas. We updated the manuscript and figure legend accordingly.

R3#31. Fig. S3Q, this is likely an EB clone, not ISC MARCM clone.

The reviewer is right by commenting the chance of the clone shown in Fig.S3Q to be an EB clone. EB clones are naturally found also in control conditions (described in detail in (de Navascues et al 2012)) and are also found in our quantifications (Fig.S3R). In contrast to controls showing single EB clones in low numbers, fra⁴ mutant clones (Fig.S3Q) only appear as single clones (Fig.S3R). In addition, single 'EB clones' might not only be mitotically inactive ISC, but also EB stemming from an ISC-clone that actively migrate. Indeed, we previously observed intermingling of Brainbow clones in (Antonello et al 2015a), which supports our main hypothesis. Probabilities of symmetric ISC/ISC, ISC/EB and EB/EB divisions are described in detail in (de Navascues et al 2012).

Thus, the majority of EB appear in duplets with an ISC, but EB can also be found isolated without being in contact with an ISC (de Navascues et al 2012). In any case, even single EB have derived from an ISC mitosis and subsequent ISC death, to make them single, is a very rare event (Zipper et al 2022). In the light of our data, we would like to suggest that the existence of single EB can also be taken as prove that the observed single EB clones in clonal studies actually are migrating EB that do migrate towards rEC in need of replacement. In addition to our data from this manuscript, the findings of integrating migratory EB is also corroborated by previous data from our Lab that in brainbow-clones intermingle and can be found outside their clonal areas (in Fig.2 of (Antonello et al 2015a)).

*To further detail on this issue, we also quantified the fraction of singular EB, which lack contact to ISC using our *klu*^{ReDDMCas9} system combined with antibody staining targeting the adherens junction marker Arm to unequivocally identify ISC (GFP⁺RFP⁺Arm⁺) and EB (GFP⁺RFP⁺Arm⁻). In*

line with the replacement rate of rEC in homeostasis (Fig.2L) and the number of NetB⁺-EC (please also see later comment R2#9), the fraction of singular EB per total EB is low with an average of 5,1% (Fig.R3). Importantly, EB specific knockout of *fra* significantly reduces the fraction of singular EB to an average of 1.6% (Fig.R3) (but not total EB numbers). This data shows that there is already a reduction visible on single EB numbers when *fra* is depleted (Fig.R3) which logically reduces the number of new rEC (Fig.2L). We now discuss this in the revised manuscript (lines 183-184).

Together, this supports our main finding in the paper and we thank the reviewer to point to this issue.

Fig.R3: The fraction of singular EB is reduced upon knockout of *fra*

Quantification of singular EB lacking contact to ISC upon seven days of *klu*^{ReDDM*Cas9*} tracing in controls (*w*¹¹¹⁸) and upon knockout of *fra* (>*fra*^{gRNA}). EB are labelled by *klu*^{ReDDM} tracing (GFP⁺RFP⁺), whereas ISC have been identified by antibody staining targeting the adherence junction marker Arm (GFP⁺RFP⁺Arm⁺). For box plots, the boxes show median, 25th and 75th percentiles, and whiskers indicate the range of values. All data points are shown by dots and means are indicated by light blue lines. 'n' are numbers of biological replicas. Asterisks denote significances from comparison by Mann Whitney test(***p<0.001).

R3#32. Fig. S6F. Is NetB expressed in EE properly secreted? If yes, why there are no protrusions produced (not even mentioning toward NetB-producing EE)? If not, why?

Yes, initially we shared the same suspect, especially after the publication from the Jasper lab that showed that EE secrete Otk (Hu et al 2021). This is why we looked carefully after a possible release of Net-B from EE. In addition to the evidence we already presented in the manuscript (Fig.S6D-H), we provide additional evidence here in the response letter.

In Fig.7I, we show that secretion and binding of NetB can be visualized with GFP tagged UAS-NetB by the presence of NetB::GFP⁺ puncta anterior to the MHB. To test whether EE are capable of properly releasing NetB, we performed EE specific (*rab3*>) expression of UAS-NetB::GFP. When we analysed for NetB::GFP⁺ puncta around EE and the presence of GFP signal in ISC/EB membranes, we could not detect any GFP signal outside of EE (Fig.R10A-A''), although the NetB::GFP signal driven by *rab3*> is really strong. Together, with Fig.5S-U and our new Fig.6, these results further strengthen our hypothesis that EC secrete NetB whereas EE cannot properly secrete NetB even under forced expression.

>NetB::GFP

FigR10: EE do not secrete NetB related to Fig.S6

(A-A'') Confocal images showing midguts (R5) with EE specific expression of >NetB::GFP using Rab3>. Antibody staining targeting the EE marker Pros and the adherence junction marker Arm label EE and the membranes of all cell types allowing to distinguish between different cell types.

R3#33. Fig. S7B-D. in the control panel, very few protrusions are facing the injury site. Only one cell in the presented panel, while protrusions can still be formed in S7C and at the similar rate.

This is correct and follows our interpretation of this dataset that during mechanical injury conditions, protrusions on ISC are at work (Fig.S7A-D) and upon laser ablation (Hu et al., 2021). Importantly, these protrusions are formed independently of fra-depletion (Fig.S7C). Concerning trio both ISC protrusions (injury) (Hu et al., 2021) and EB protrusions (homeostasis, Fig. S4 and S5) rely on trio. We added this in lines 300–314 to the revised manuscript pointing out clearer that different mechanisms controlling protrusion formation and progenitor cell migration during homeostasis and injury are at work.

Reviewer #4 (Remarks to the Author):

We thank Reviewer 4 for reading and evaluating our manuscript.

References used throughout the response letter

- Aghajanian P, Takashima S, Paul M, Younossi-Hartenstein A, Hartenstein V. 2016. Metamorphosis of the *Drosophila* visceral musculature and its role in intestinal morphogenesis and stem cell formation. *Dev Biol* 420: 43-59
- Antonello ZA, Reiff T, Ballesta-Illan E, Dominguez M. 2015a. Robust intestinal homeostasis relies on cellular plasticity in enteroblasts mediated by miR-8-Escargot switch. *EMBO J* 34: 2025-41
- Antonello ZA, Reiff T, Dominguez M. 2015b. Mesenchymal to epithelial transition during tissue homeostasis and regeneration: Patching up the *Drosophila* midgut epithelium. *Fly (Austin)* 9: 132-7
- Bernet A, Mazelin L, Coissieux MM, Gadot N, Ackerman SL, et al. 2007. Inactivation of the UNC5C Netrin-1 receptor is associated with tumor progression in colorectal malignancies. *Gastroenterology* 133: 1840-8
- Boyer NP, Gupton SL. 2018. Revisiting Netrin-1: One Who Guides (Axons). *Frontiers in cellular neuroscience* 12: 221
- Brankatschk M, Dickson BJ. 2006. Netrins guide *Drosophila* commissural axons at short range. *Nature Neuroscience* 9: 188-94
- Buchon N, Osman D, David FP, Fang HY, Boquete JP, et al. 2013. Morphological and molecular characterization of adult midgut compartmentalization in *Drosophila*. *Cell reports* 3: 1725-38
- Buddika K, Xu J, Ariyapala IS, Sokol NS. 2021. I-KCKT allows dissection-free RNA profiling of adult *Drosophila* intestinal progenitor cells. *Development (Cambridge, England)* 148: dev196568
- Castets M, Broutier L, Molin Y, Brevet M, Chazot G, et al. 2011. DCC constrains tumour progression via its dependence receptor activity. *Nature* 482: 534-7
- Chen J, Sayadian A-C, Lowe N, Lovegrove HE, St Johnston D. 2018. An alternative mode of epithelial polarity in the *Drosophila* midgut. *PLOS Biology* 16: e3000041
- de Navascues J, Perdigoto CN, Bian Y, Schneider MH, Bardin AJ, et al. 2012. *Drosophila* midgut homeostasis involves neutral competition between symmetrically dividing intestinal stem cells. *EMBO J* 31: 2473-85
- Dorsten JN, Kolodziej PA, VanBerkum MFA. 2007. Frazzled regulation of myosin II activity in the *Drosophila* embryonic CNS. *Developmental biology* 308: 120-32
- Dutta D, Dobson AJ, Houtz PL, Glasser C, Revah J, et al. 2015. Regional Cell-Specific Transcriptome Mapping Reveals Regulatory Complexity in the Adult *Drosophila* Midgut. *Cell reports* 12: 346-58
- Fearon ER, Cho KR, Nigro JM, Kern SE, Simons JW, et al. 1990. Identification of a chromosome 18q gene that is altered in colorectal cancers. *Science* 247: 49-56
- Forsthoefel DJ, Liebl EC, Kolodziej PA, Seeger MA. 2005. The Abelson tyrosine kinase, the Trio GEF and Enabled interact with the Netrin receptor Frazzled in *Drosophila*. *Development (Cambridge, England)* 132: 1983-94
- Garbe DS, O'Donnell M, Bashaw GJ. 2007. Cytoplasmic domain requirements for Frazzled-mediated attractive axon turning at the *Drosophila* midline. *Development (Cambridge, England)* 134: 4325-34

- Golenkina S, Chaturvedi V, Saint R, Murray MJ. 2018. Frazzled can act through distinct molecular pathways in epithelial cells to regulate motility, apical constriction, and localisation of E-Cadherin. *PLoS one* 13: e0194003
- Grady WM. 2013. Context is everything for dependence receptors in colorectal cancer. *Proc Natl Acad Sci U S A* 110: 2697-8
- Hu DJ-K, Yun J, Elstrott J, Jasper H. 2021. Non-canonical Wnt signaling promotes directed migration of intestinal stem cells to sites of injury. *Nature Communications* 12: 7150
- Jin Y, Patel PH, Kohlmaier A, Pavlovic B, Zhang C, Edgar BA. 2017. Intestinal Stem Cell Pool Regulation in *Drosophila*. *Stem Cell Reports* 8: 1479-87
- Keleman K, Dickson BJ. 2001. Short- and Long-Range Repulsion by the *Drosophila* Unc5 Netrin Receptor. *Neuron* 32: 605-17
- Kinzler KW, Vogelstein B. 1996a. Lessons from Hereditary Colorectal Cancer. *Cell* 87: 159-70
- Kinzler KW, Vogelstein B. 1996b. Lessons from hereditary colorectal cancer. *Cell* 87: 159-70
- Kolodziej PA, Timpe LC, Mitchell KJ, Fried SR, Goodman CS, et al. 1996. frazzled Encodes a *Drosophila* Member of the DCC Immunoglobulin Subfamily and Is Required for CNS and Motor Axon Guidance. *Cell* 87: 197-204
- Korzelius J, Azami S, Ronnen-Oron T, Koch P, Baldauf M, et al. 2019. The WT1-like transcription factor Klumpfuss maintains lineage commitment of enterocyte progenitors in the *Drosophila* intestine. *Nat Commun* 10: 4123
- Koyama LAJ, Aranda-Diaz A, Su YH, Balachandra S, Martin JL, et al. 2020. Bellymount enables longitudinal, intravital imaging of abdominal organs and the gut microbiota in adult *Drosophila*. *PLoS biology* 18: e3000567
- Krdnja D, El Marjou F, Guirao B, Richon S, Leroy O, et al. 2019. Active cell migration is critical for steady-state epithelial turnover in the gut. *Science* 365: 705-10
- Lai Wing Sun K, Correia JP, Kennedy TE. 2011. Netrins: versatile extracellular cues with diverse functions. *Development* 138: 2153-69
- Lane NJ, Skaer HI. 1980. Intercellular Junctions in Insect Tissues In *Advances in Insect Physiology*, ed. MJ Berridge, JE Treherne, VB Wigglesworth, pp. 35-213: Academic Press
- Liang J, Balachandra S, Ngo S, O'Brien LE. 2017. Feedback regulation of steady-state epithelial turnover and organ size. *Nature* 548: 588-91
- Lin Y, Pal DS, Banerjee P, Banerjee T, Qin G, et al. 2024. Ras suppression potentiates rear actomyosin contractility-driven cell polarization and migration. *Nature cell biology* 26: 1062-76
- Llimargas M, Strigini M, Katidou M, Karagogeos D, Casanova J. 2004. Lachesin is a component of a septate junction-based mechanism that controls tube size and epithelial integrity in the *Drosophila* tracheal system. *Development (Cambridge, England)* 131: 181-90
- Marchetti M, Zhang C, Edgar BA. 2022. An improved organ explant culture method reveals stem cell lineage dynamics in the adult *Drosophila* intestine. *eLife* 11: e76010
- Marianes A, Spradling AC. 2013. Physiological and stem cell compartmentalization within the *Drosophila* midgut. *eLife* 2: e00886
- Martin JL, Sanders EN, Moreno-Roman P, Jaramillo Koyama LA, Balachandra S, et al. 2018. Long-term live imaging of the *Drosophila* adult midgut reveals real-time dynamics of division, differentiation and loss. *Elife* 7: e36248
- Mehlen P, Tauszig-Delamasure S. 2014. Dependence receptors and colorectal cancer. *Gut* 63: 1821-9
- Micchelli CA, Perrimon N. 2006. Evidence that stem cells reside in the adult *Drosophila* midgut epithelium. *Nature* 439: 475-9
- O'Donnell M, Chance RK, Bashaw GJ. 2009. Axon growth and guidance: receptor regulation and signal transduction. *Annu Rev Neurosci* 32: 383-412
- O'Donnell MP, Bashaw GJ. 2013. Distinct functional domains of the Abelson tyrosine kinase control axon guidance responses to Netrin and Slit to regulate the assembly of neural circuits. *Development (Cambridge, England)* 140: 2724-33
- Ohlstein B, Spradling A. 2006. The adult *Drosophila* posterior midgut is maintained by pluripotent stem cells. *Nature* 439: 470-4

- Ohlstein B, Spradling A. 2007. Multipotent Drosophila intestinal stem cells specify daughter cell fates by differential notch signaling. *Science* 315: 988-92
- Okada M, Takano T, Ikegawa Y, Ciesielski H, Nishida H, Yoo SK. 2023. Oncogenic stress-induced Netrin is a humoral signaling molecule that reprograms systemic metabolism in Drosophila. *EMBO J* n/a: e111383
- Pal DS, Banerjee T, Lin Y, de Trogoff F, Borleis J, et al. 2023. Actuation of single downstream nodes in growth factor network steers immune cell migration. *Developmental cell* 58: 1170-88.e7
- Patel PH, Dutta D, Edgar BA. 2015. Niche appropriation by Drosophila intestinal stem cell tumours. *Nature cell biology* 17: 1182-92
- Perdigoto CN, Schweisguth F, Bardin AJ. 2011. Distinct levels of Notch activity for commitment and terminal differentiation of stem cells in the adult fly intestine. *Development (Cambridge, England)* 138: 4585-95
- Pert M, Gan M, Saint R, Murray MJ. 2015. Netrins and Frazzled/DCC promote the migration and mesenchymal to epithelial transition of Drosophila midgut cells. *Biol Open* 4: 233-43
- Petrie RJ, Yamada KM. 2012. At the leading edge of three-dimensional cell migration. *Journal of cell science* 125: 5917-26
- Pignoni F, Zipursky SL. 1997. Induction of Drosophila eye development by Decapentaplegic. *Development (Cambridge, England)* 124: 271-78
- Reiff T, Antonello ZA, Ballesta-Illan E, Mira L, Sala S, et al. 2019. Notch and EGFR regulate apoptosis in progenitor cells to ensure gut homeostasis in Drosophila. *EMBO J* 38: e101346
- Reiff T, Jacobson J, Cognigni P, Antonello Z, Ballesta E, et al. 2015. Endocrine remodelling of the adult intestine sustains reproduction in Drosophila. *Elife* 4: e06930
- Rera M, Bahadorani S, Cho J, Koehler CL, Ulgherait M, et al. 2011. Modulation of longevity and tissue homeostasis by the Drosophila PGC-1 homolog. *Cell metabolism* 14: 623-34
- Shibata D, Reale MA, Lavin P, Silverman M, Fearon ER, et al. 1996a. The DCC Protein and Prognosis in Colorectal Cancer. *New England Journal of Medicine* 335: 1727-32
- Shibata D, Reale MA, Lavin P, Silverman M, Fearon ER, et al. 1996b. The DCC protein and prognosis in colorectal cancer. *The New England journal of medicine* 335: 1727-32
- Shin SK, Nagasaka T, Jung BH, Matsubara N, Kim WH, et al. 2007. Epigenetic and genetic alterations in Netrin-1 receptors UNC5C and DCC in human colon cancer. *Gastroenterology* 133: 1849-57
- Singer JB, Harbecke R, Kusch T, Reuter R, Lengyel JA. 1996. Drosophila brachyenteron regulates gene activity and morphogenesis in the gut. *Development* 122: 3707-18
- Takashima S, Mkrtchyan M, Younossi-Hartenstein A, Merriam JR, Hartenstein V. 2008. The behaviour of Drosophila adult hindgut stem cells is controlled by Wnt and Hh signalling. *Nature* 454: 651-5
- Tepass U, Hartenstein V. 1994. The Development of Cellular Junctions in the Drosophila Embryo. *Developmental biology* 161: 563-96
- Wadsworth WG. 2015. Understanding axon guidance: attraction, repulsion, and statistical physics. *Neural Regen Res* 10: 176-9
- Wang L, Zeng X, Ryoo HD, Jasper H. 2014. Integration of UPRER and Oxidative Stress Signaling in the Control of Intestinal Stem Cell Proliferation. *PLoS genetics* 10: e1004568
- Williams ME, Wu SC, McKenna WL, Hinck L. 2003. Surface expression of the netrin receptor UNC5H1 is regulated through a protein kinase C-interacting protein/protein kinase-dependent mechanism. *J Neurosci* 23: 11279-88
- Woods DF, Bryant PJ. 1991. The discs-large tumor suppressor gene of Drosophila encodes a guanylate kinase homolog localized at septate junctions. *Cell* 66: 451-64
- Yagi R, Mayer F, Basler K. 2010. Refined LexA transactivators and their use in combination with the Drosophila Gal4 system. *Proceedings of the National Academy of Sciences* 107: 16166-71
- Yamagishi S, Yamada K, Sawada M, Nakano S, Mori N, et al. 2015. Netrin-5 is highly expressed in neurogenic regions of the adult brain. *Frontiers in Cellular Neuroscience* Volume 9 - 2015

- Zipper L, Batchu S, Kaya NH, Antonello ZA, Reiff T. 2022. The MicroRNA miR-277 Controls Physiology and Pathology of the Adult Drosophila Midgut by Regulating the Expression of Fatty Acid beta-Oxidation-Related Genes in Intestinal Stem Cells. *Metabolites* 12
- Zipper L, Corominas-Murtra B, Reiff T. 2025. Steroid hormone-induced wingless ligands tune female intestinal size in Drosophila. *Nature Communications* 16: 436
- Zipper L, Jassmann D, Burgmer S, Gorlich B, Reiff T. 2020. Ecdysone steroid hormone remote controls intestinal stem cell fate decisions via the PPARgamma-homolog Eip75B in Drosophila. *Elife* 9: e55795

REVIEWER COMMENTS

Dear Reviewers,

First of all, we want to thank all Reviewers for your time and the valuable suggestions that helped to improve our manuscript. Following the comments of Reviewer 1 & 2, we updated the manuscript with new additional suggested analysis and updated the clarity of results, methods and specificity of our discussion where points remained to be clarified.

Reviewer #1 (Remarks to the Author):

In this revised manuscript by Zipper et al., the authors have addressed most of my prior critiques. In particular, the addition of fra LOF data in the Hamelin assay and the evidence supporting EB protrusion chemotaxis towards NetB-expressing cell clones considerably strengthened the manuscript. I have two remaining major comments that should be addressed through textual revisions.

We thank the Reviewer for this positive evaluation of our revised manuscript and detail on the requested textual revisions in the following.

First, one sticking point for me remains the description and discussion of data where effects of experimental manipulations on EB protrusion number/length/directionality do not go hand in hand with effects on EC replacement. This includes unc-5 RNAi (disrupts EB protrusion orientation towards old ECs but does not affect EC renewal) and rescue of fra LOF with fra lacking P3 (restores protrusion number and length to control levels but does not rescue EC replacement). It remains unclear to me how these results can be reconciled with a model where EC replacement depends on protrusion dynamics, and the authors need to provide a satisfying mechanistic explanation that accommodates these results in the discussion of their model.

We thank the Reviewer for this comment and agree that these issues were in need to be addressed. Based on our presented data, we extend in the following paragraphs on the connection of how protrusion directionality and dynamics finally control EC replacement. We like to split our reply to this point in the case for unc-5-RNAi and fraP3 in fra-LOF background.

Concerning the role of Unc-5, our data shows and confirms the previously described subordinate role of Unc-5 to Frazzled/DCC from neurodevelopmental studies. For convenience, we copy the according issues from the first Reviewer Comment Letter in the following in blue text:

R1#1c...In addition, we provide further evidence that support Frazzled, but not Unc-5, in the coordination of EB migration:

- i) Our experiments using klu^{ReDDM} to manipulate Fra and Unc-5 receptor levels within EB suggest that Fra mediates the attractive effect in EB migration as OE of fra induces protrusion formation (Fig.2B,H).*
- ii) KD/KO of fra disrupts protrusion formation and EB migration as shown by decreased rEC renewal (Fig.2C,F,H,L).*

- iii) *Unc-5 OE however abolishes rEC replenishment (Fig.2L) most probably by antagonistically acting on Fra-receptors (Keleman & Dickson, 2001).*
- iv) *unc-5-RNAi neither reduces rEC (Fig.2L) nor...*
- v) *...fly survival (Fig.2O) suggesting that Fra is still functioning ensuring homeostasis and midgut integrity under these conditions.*
- vi) *In addition, unc-5-RNAi does not elicit protrusions in ISC (Fig.3F,G).*
- vii) *A subordinated role of Unc-5 is further supported by its internalization (Fig.S1E-G,I'''), which is generally assumed to indicate active fra/DCC signalling (Boyer & Gupton 2018).*
- viii) *In line with this, Fra receptors remain on the membrane when Unc-5 is overexpressed and internalized (Fig.S1I-I'').*
- ix) *Double OE and KD of both receptors phenocopies the sole Fra manipulations (Fig.S3A-G).*
- x) *When Fra is depleted with >>fra-RNAi, NetB::GFP puncta on membranes of progenitors are not detected and accumulated anymore (Fig.7J) compared to wild-type progenitors (Fig.7I).*

And R1#1d: As pointed out in the previous comment (R1#1c), Unc-5 is generally accepted to antagonize Fra/DCC receptors in the literature by acting repulsive in heterodimers with Fra/DCC, whereas homodimers of Fra or DCC on the membrane act attractive (O'Donnell et al 2009). Thus, either Unc-5 depletion or internalization allows Fra/DCC to signal as homodimer and direct migration. Our data on the role of Unc-5 in EB migration supports previous findings from neurodevelopmental studies and point to a subordinate role below Fra/DCC:

- i) *EB specific manipulations of unc-5 show opposing effects to Fra on protrusion formation (Fig.2H,L, Fig.S2A).*
- ii) *In addition, rEC renewal (Fig.2L) works normal when Unc-5 is depleted from EB suggesting Fra homodimers direct EB movement.*
- iii) *Unc-5 OE in EB however reduces rEC formation (Fig.2L) and fly survival (Fig.2O) by preventing Fra signalling through its abundance.*
- iv) *As mentioned in our comment R1#1c, internalization of Unc-5 into vesicles is described during axon guidance and indicates attractive signalling via Fra/DCC (Boyer & Gupton 2018). In midgut EB, internalisation of Unc-5 into vesicles with GFP tagged Unc-5 (unc-5::GFP, Fig.S1E-H') and new antibody stainings targeting Unc-5 (new Fig.S1E) is observed as well, which in turn suggests signalling through Fra/DCC homodimers.*
- v) *Importantly, only homodimers of Fra/DCC were shown to connect to the F-Actin network allowing protrusion formation (Garbe et al 2007, Golenkina et al 2018).*
- vi) *In vitro experiments on rat neurons showed removal of UNC5H1 from the cell surface of neurons and growth cones but leaves DCC at the cell surface. These changes of receptor levels at the cell surface could lead to a switch from UNC5H1 mediated repulsion to DCC induced attraction towards a Netrin-1 gradient (Williams et al 2003).*
- vii) *As mentioned previously, membrane-bound Netrins to DCC indicate active signalling (Lai Wing Sun et al 2011), which we observe in Hamelin (Fig.7I) and discover in our new experiments that these puncta depend on the presence of Fra-receptors (Fig.7J).*

Together, we conclude from these lines of evidence that EC replenishment is mediated by Frazzled/DCC signaling, which was previously described in neurodevelopmental studies. In line with these studies, our data implies that presence of Unc-5 (by overexpression) on the membrane

probably leads to repulsive heterodimers (Fra/Unc-5) and Unc-5-depletion as well as internalization allows Fra homodimers to direct migration and MET and thus boost total EC replenishment (Fig.2M), which is discussed in the first paragraph of the discussion.

We were not entirely sure whether the Reviewer was asking about rEC or total new EC in his comment: “This includes unc-5 RNAi (disrupts EB protrusion orientation towards old ECs but does not affect EC renewal)”. Concerning total new EC, our data shows that unc-5-RNAi strongly stimulates EC renewal (Fig.2M), which we attribute to the following higher abundance of Fra homodimers. Similar to neurodevelopmental studies on axon guidance, this enables active Fra-signaling by homodimers and MET, which we think is remarkable as the expressivity is similar to forced >fra expression (Fig.2M). The same explanation holds true for the renewal of rEC, where unc-5-RNAi phenocopies fra expression as well (Fig.2L).

*For the Reviewer’s concern about protrusion directionality and how these dynamics of protrusions are connected to EC renewal this implies that whenever Unc-5 is removed (by RNAi or internalization) Fra/DCC homodimers signal and stabilize protrusion elongation (Fig.2H) and also increase protrusion number (Fig.S1A). Thus, Unc-5 LOF in the EB lineage using *klu^{ReDDM}* (Fig.2) raises the sensitivity of protrusions to detect low NetB-levels and increase their stabilization towards older EC with low NetB (Fig.2K) similar to Fra GOF. The idea of protrusion stabilization is further supported by our observations that an increase NetB-levels across all EC (Fig.5P), we observe significantly more and longer protrusions (Fig.5Q, R). Our data in Fig.3K-R also shows that protrusion formation per se is not sufficient to induce EC renewal but depends on additional fate specifying Notch-signalling and possibly presence of NetB on a worn-out EC in need of replacement (Fig.5X). When we express NetB in single EC (Fig.6) in our new experiment using clonal expression, we see a significant shift of protrusions pointing towards these NetB⁺-EC but even under these conditions some few protrusions are still (transiently) detected in our static images that point away from a NetB-source (Fig.6G). This further supports the idea of high dynamics behind these protrusions and that their existence hints to EC in need of replacement, but not necessarily means MET to EC. Independent of protrusions and important basis of our study, our quantitative assessment of rEC numbers over a large timespan using ReDDM reveals the true frequency of coordinated EB migration and renewal independent from protrusion dynamics.*

We thank the Reviewer for this constructive comment and clarify and discuss this issue now in our new revised manuscript in lines (420-431) as well as updated the methods section (lines 962-1008) regarding directionality measurements.

Concerning the rescue capacity of the FraP3 domain in Fra LOF background, we want to point out that the rescue is incomplete: It is true that it is not as strong as FraP1 and FraP2 but it neither is significantly different from controls as well as fra^{gRNA} (Fig.4I).

For the molecular basis of this observation, the existing literature on the domain structures of Fra/DCC and P3 revealed details up to crystallization studies (Xu et al 2014, Xu et al 2018). This data suggests that multimerization of Fra/DCC is a multi-step process initiated by binding of Netrins that are able to bridge two Fra/DCC ectodomains. Furthermore, for DCC it was shown that the P3 domain is the primary intracellular stabilizer of dimerization and provides a specific docking

site for Focal Adhesion Kinase (FAK) (Xu et al 2018), which is a non-receptor tyrosine kinase that upon autophosphorylation remodels the cytoskeleton for axon turning together with Src.

In the fly however, binding of Src and Fak are dispensable for embryonic CNS phenotypes and multimerization can occur even without the P3 domain (Garbe et al 2007, O'Donnell & Bashaw 2013). Prime candidates mediating the subsequent contact to the cytoskeleton are the RhoGEF Trio and the Abelson Kinase (Abl) that both bind to all Fra P-motifs and are thought to build a signaling hub (Dorsten et al 2010, Forsthoefel et al 2005). This is further supported by our data on Trio LOF that abolishes protrusion formation and rEC (Fig.S4N, S5K-O) even in the presence of Fra GOF. We thus speculate that the lack of the P3 domain weakens the function and the cytoskeletal interaction of the Fra/Trio/Abl signaling hub in a different way than loss of P1 or P2 and thus elicits only a partial rescue.

We thank the Reviewer for this comment and now discuss our observations on Fra P3 in more detail (lines 256-265) in the revised version of the manuscript. We also hope for the understanding of Reviewer 1 that we do not wish to disclose preliminary data for the function of Abl.

Second, I commend the authors for removing the cancer metastasis portion of the manuscript, but I believe that the section of the Discussion that speculates about a possible role for Net/Fra in metastasis should be dramatically shortened or removed. If it is included, this part should strictly focus on direct implications of the presented results for tumor EMT and metastasis.

We agree with the Reviewer that the last lines (519ff) of the discussion needed rework. The paragraph serves to directly point out the intriguing connections, discoveries and possibilities on the molecular level that are currently made on axon guidance signaling cues involved in EMT/MET and metastasis in some of the most frequent and aggressive cancers (CRC, pancreas, breast, glioblastoma, lung and gastric).

Epecially, metastatic growth is far less understood than primary tumor hallmarks but metastases are responsible for almost 9 out of 10 cancer-related casualties. Thus, we think that our findings investigating the role of Fra/DCC in physiological migration and that no mutations are actually necessary for progenitors to make them invade from endo- to ectoderm are of high interest to the broad readership of Nature Communications.

We thank the Reviewer for this recommendation and rephrased this part of the discussion to indicate more clearly the connections to our findings.

Minor comment:

“Notch”, used only a few times in the paper, is still abbreviated as “N” in the relevant figure. I encourage the authors to further reduce the number of abbreviations and also stay consistent with their use of abbreviations.

We apologize for having overseen this and corrected Fig.3 in the new revised manuscript. Additionally, we removed further non-frequently used abbreviations throughout the manuscript.

Reviewer #2 (Remarks to the Author):

The authors have done a great job addressing several of the major concerns raised by the reviewers. They convincingly demonstrate that Fra function is required in EBs, but not in ISCs, for the replacement of rECs. In addition, they show that progenitor cells extend directional protrusions toward individual ECs with ectopic NetB expression.

That said, several points would benefit from further clarification or additional analysis:

We thank the Reviewer for the appreciation of our newly added experiments and their importance for the revised manuscript. Concerning the remaining points, we now provide further details with new additional data on the concerns addressed below.

1. In Figure 5E, NetB appears to be upregulated in an aEC located adjacent to an EB rather than in an rEC. Would NetB expression in aECs not obscure or mask a gradient emanating from rECs? Could NetB represent a more general signal associated with worn-out aECs and rECs, rather than being rEC-specific?

Yes, we entirely agree with the Reviewer's comment that NetB is a general signal coming up in worn-out EC, independent of whether they are aEC or rEC.

Briefly, the main focus of our manuscript is that NetB and Fra/DCC signaling coordinate directed migration of EB towards NetB-expressing EC and thus spatially coordinated replacement of worn-out EC. This is a new concept during homeostatic processes in the intestine and the reason why we focus on rEC as readout because their replacement requires migration in contrast to aEC.

We also agree with the Reviewer that our data shows, that during homeostasis, NetB is upregulated in worn-out ECs and we describe it this way (e.g. summary lines 14-15). NetB⁺-EC attract and stabilize Fra guided EB protrusions (Fig.5J-U) and the following data we present in Fig.5V-X, Fig.6. and Fig.7 actually proves that this action of NetB is capable to cover the distance that EB need to migrate towards the NetB source.

We carefully checked and corrected the manuscript for clarity on this issue and added this more clearly to the discussion now (lines 508-518). Please find data and discussion on this also in the next comment.

2. Related to this, the fraction of newly generated rECs is calculated relative to the total number of new ECs, leading the authors to conclude that the rEC fraction decreases while the aEC fraction increases. However, the absolute number of newly generated aECs is not addressed. Is the number of aECs reduced upon NetB or Fra knockdown? Is NetB specifically required in rECs? The strong reduction in the total number of newly generated ECs in Figure 2M is unlikely to be explained solely by a loss of rECs. A plot showing the absolute numbers of newly generated rECs and aECs would greatly strengthen this conclusion.

In addition to the previous comment, we can also positively confirm this comment and now provide the requested plot of aEC. The graph below will be included as new dataset in Fig.S2N and directly addresses the Reviewer's comment by showing that the number of aEC is reduced in addition to rEC (Fig.2L) numbers upon depletion of fra-RNAi (Fig.2M).

We thank the Reviewer for this comment and updated the main text concerning clarity for this point. Now including Fig.S2N, we updated the results accordingly (line 155-160) and discuss this new data as Fig.S2N in the revised manuscript and extend on the role of Fra in MET in the discussion (lines 415-431 and 508-518).

A

Fig.R1/S2N: The numbers of new rEC and aEC are significantly reduced upon EB specific knockdown of fra.

Quantification of new aEC and new rEC upon seven days of *klu*^{ReDDM} tracing in controls and with expression of *>fra-RNAi*. For box plots, the boxes show median, 25th and 75th percentiles, and whiskers indicate the range of values. All data points are shown by dots and means are indicated by light blue lines. 'n' are numbers of biological replicas. Asterisks denote significances from comparison by Mann Whitney test (***)*p*<0.001).

Concerning the specific requirement of NetB in rEC, we kindly refer the Reviewer to our response to the previous comment #1.

3. In Figures 5S–T, NetB is also detected in stem and progenitor cells, which argues against a simple role as a directional guidance cue. Moreover, it is not clear from Figure 5T that NetB is upregulated specifically in rECs. I am not sure that I would include this item in the final manuscript, as the requirement for NetB downstream of E-cad to replace rECs is not corroborated by functional data.

The reviewer is right detecting the NetB signal on progenitors, and we wondered about this as well when we first detected it. However, we want to put forward the idea that the detected NetB is rather on, and not in the progenitor membrane and not endogenously derived from ISC or EB. For this we have several lines of evidence in the manuscript (line 366-378) and new functional data we now include in this letter.

- i) In the manuscript we show that NetB::GFP expressed with *byn*> from the hindgut can be detected as GFP-punctae on midgut progenitor cells (Fig.7I).
- ii) When we remove Fra-receptors from these midgut progenitors using *>>fra-RNAi*, these NetB::GFP punctae are no longer observed suggesting that Fra-receptors are required for this observations

- iii) This is in line with observations of neuronal growth cones guided by Netrin-Fra/DCC signaling in which clustering of Netrins on DCC receptors and stabilization was observed (Matsumoto & Nagashima 2010).
- iv) EB-specific knockdown of NetB in esg^{ReDDM} and klu^{ReDDM} shows no obvious effects on EB morphology, protrusions or EC renewal (Fig.R2).
- v) In addition, we did not detect NetB reporter activity in other cells than EC (Fig.S6A,B).

Fig.R2: The number of new EC is not affected upon ISC/EB or EB specific knockdown of NetB

(A-D) Confocal images of midguts (R5) after 7 days of (A-B) esg^{ReDDM} and (C-D) klu^{ReDDM} tracing in (A,C) controls and (B,D) with expression of $>NetB-RNAi$. (E-F) Quantification of new EC upon seven days of (A) esg^{ReDDM} and (B) klu^{ReDDM} tracing in controls and with expression of $>NetB-RNAi$. For box plots, the boxes show median, 25th and 75th percentiles, and whiskers indicate the range of values. All data points are shown by dots and means are indicated by light blue lines. 'n' are numbers of biological replicas. (A-B) Data have been analyzed using Kruskal Wallis tests, but do not show significant differences.

We now more thoroughly discuss the detected signal of NetB on progenitors in the revised manuscript (lines 370-378).

Concerning the connection between E-cadherin and Netrin expression, we agree with the Reviewer and now add a brief discussion on this putative intriguing connection. Previous studies have shown that the loss of E-cadherin not only releases α -catenin (Liang et al 2017) but also β -catenin from adherence junctions and leads to its translocation to the nucleus (Kuphal et al 2004, Steinhilber et al 2001). In the nucleus, β -catenin was shown to interact with the transcription factor NF- κ B and this direct interaction activates NF- κ B during intestinal inflammation (Kuphal et al 2004, Sun et al 2005). In turn, NF- κ B has been shown to activate the Netrin promoter in mouse colonic crypts (Paradisi et al 2008, Paradisi & Mehlen 2010) and thus hints to a direct molecular mechanism behind our observations. We speculate on this possible mechanism in the revised discussion (lines 459-463) and thank the Reviewer for pointing this out.

4. Based on Figure S1B, it is not entirely clear that Fra is absent from ISCs, and this conclusion may be overstated. While the functional data showing that Fra is not required in ISCs for rEC renewal are convincing, the observation that Unc-5 overexpression in ISCs reduces progeny numbers (Figure 3J) suggests that endogenous Fra activity may exist in ISCs. Although the authors propose that ectopic Unc-5 may form homodimers with stronger effects, it is equally plausible that Fra is present in ISCs at levels below the detection threshold of the endogenously tagged Fra::GFP.

We thank the Reviewer for pointing out that our imaging and functional data cannot entirely exclude Fra receptors in ISC. Although, Fra-levels in ISC might be below confocal detection levels, we do find an effect by ectopic Unc-5 expression in ISC (Fig.3J). This either suggests a role for Unc-5 homodimers in ISC or presence of Fra-Unc-5 heterodimers that stem from Fra-receptor expression below the detection limit of our confocal microscopes that then act similarly as observed in EB (Fig.2M). Homodimers of Unc-5 are well known in cancer research and involved in the control of cell survival, migration, and differentiation (Brisset et al 2021, Mehlen & Tauszig-Delamasure 2014). However, as depletions of Fra and Unc-5, do not affect protrusion length, new rEC, number of ISC and progeny (Fig.3G-J), we assume no major function of both receptors in ISC.

We now describe this accordingly and discuss it in the revised manuscript (lines 439-450).

5. Finally, although knockdown of Fra in ISCs/EBs reduces longevity, this experiment does not allow a clear assessment of the contribution of EC-derived NetB to this phenotype.

We thank the Reviewer for pointing this out. It is true that our NetA^{-/-}/NetBTM survival assay does not prove that EC-derived NetB reduces survival, but in conjunction with the survival assays performed for Fra-depletion specifically in EB, we present the finding that these constitutive mutants die earlier as additional evidence to the receptor's manipulations (Fig.2N-P, Fig.S2Z). Unfortunately, we attempted direct experimental proof, but failed several times to combine and generate living flies with EC-specific mex-Gal4, Gal80ts-repressor and double LOF NetA and NetB which is necessary as they are known to compensate each other's function (Brankatschk & Dickson 2006).

Thus, the survival assay presented in Fig.6K is the only evidence, we can present on Netrins in the context of adult fly survival. For completeness of this discussion, we think it is worth to mention that NetA^{-/-}/NetBTM flies survive 37d in average, which we think is a good hint that NetA^{-/-}/NetBTM flies do not suffer major neurodevelopmental defects. In addition, these constitutive mutants are well-described and used in the field (Brankatschk & Dickson 2006) and phenocopy our data in Fig.5. In addition, constitutive fra-mutant flies (Kolodziej et al 1996) (Fig.S3N) phenocopy NetA^{-/-}/NetBTM survival data.

We correctly introduce these mutants as constitutive now and describe the obtained data more clearly now with soft claims in the new revised manuscript (lines 324-328). We thank the Reviewer for pointing this out.

Reviewer #3 (Remarks to the Author):

The Authors addressed most of concerns raised in previous manuscript. I would recommend for publishing the revised manuscript.

We thank the Reviewer for this positive evaluation of our revised manuscript.

Reviewer #4 (Remarks to the Author):

We thank the Reviewer for evaluating our revised manuscript.

Bibliography

- Boyer NP, Gupton SL. 2018. Revisiting Netrin-1: One Who Guides (Axons). *Frontiers in cellular neuroscience* 12: 221
- Brankatschk M, Dickson BJ. 2006. Netrins guide Drosophila commissural axons at short range. *Nature Neuroscience* 9: 188-94
- Brisset M, Grandin M, Bernet A, Mehlen P, Hollande F. 2021. Dependence receptors: new targets for cancer therapy. *EMBO Mol Med* 13: e14495
- Dorsten JN, Varughese BE, Karmo S, Seeger MA, VanBerkum MFA. 2010. In the Absence of Frazzled Over-Expression of Abelson Tyrosine Kinase Disrupts Commissure Formation and Causes Axons to Leave the Embryonic CNS. *PLOS ONE* 5: e9822
- Forsthoefel DJ, Liebl EC, Kolodziej PA, Seeger MA. 2005. The Abelson tyrosine kinase, the Trio GEF and Enabled interact with the Netrin receptor Frazzled in Drosophila. *Development* 132: 1983-94
- Garbe DS, O'Donnell M, Bashaw GJ. 2007. Cytoplasmic domain requirements for Frazzled-mediated attractive axon turning at the Drosophila midline. *Development* 134: 4325-34
- Golenkina S, Chaturvedi V, Saint R, Murray MJ. 2018. Frazzled can act through distinct molecular pathways in epithelial cells to regulate motility, apical constriction, and localisation of E-Cadherin. *PLoS One* 13: e0194003
- Kolodziej PA, Timpe LC, Mitchell KJ, Fried SR, Goodman CS, et al. 1996. frazzled Encodes a Drosophila Member of the DCC Immunoglobulin Subfamily and Is Required for CNS and Motor Axon Guidance. *Cell* 87: 197-204
- Kuphal S, Poser I, Jobin C, Hellerbrand C, Bosserhoff AK. 2004. Loss of E-cadherin leads to upregulation of NFκB activity in malignant melanoma. *Oncogene* 23: 8509-19

- Lai Wing Sun K, Correia JP, Kennedy TE. 2011. Netrins: versatile extracellular cues with diverse functions. *Development* 138: 2153-69
- Liang J, Balachandra S, Ngo S, O'Brien LE. 2017. Feedback regulation of steady-state epithelial turnover and organ size. *Nature* 548: 588-91
- Matsumoto H, Nagashima M. 2010. Netrin-1 elevates the level and induces cluster formation of its receptor DCC at the surface of cortical axon shafts in an exocytosis-dependent manner. *Neuroscience Research* 67: 99-107
- Mehlen P, Tauszig-Delamasure S. 2014. Dependence receptors and colorectal cancer. *Gut* 63: 1821-9
- O'Donnell M, Chance RK, Bashaw GJ. 2009. Axon growth and guidance: receptor regulation and signal transduction. *Annu Rev Neurosci* 32: 383-412
- O'Donnell MP, Bashaw GJ. 2013. Src inhibits midline axon crossing independent of Frazzled/Deleted in Colorectal Carcinoma (DCC) receptor tyrosine phosphorylation. *J Neurosci* 33: 305-14
- Paradisi A, Maisse C, Bernet A, Coissieux MM, Maccarrone M, et al. 2008. NF- κ B Regulates Netrin-1 Expression and Affects the Conditional Tumor Suppressive Activity of the Netrin-1 Receptors. *Gastroenterology* 135: 1248-57
- Paradisi A, Mehlen P. 2010. Netrin-1, a missing link between chronic inflammation and tumor progression. *Cell Cycle* 9: 1253-62
- Steinhusen U, Weiske J, Badock V, Tauber R, Bommert K, Huber O. 2001. Cleavage and Shedding of E-cadherin after Induction of Apoptosis*. *Journal of Biological Chemistry* 276: 4972-80
- Sun J, Hobert ME, Duan Y, Rao AS, He T-C, et al. 2005. Crosstalk between NF- κ B and β -catenin pathways in bacterial-colonized intestinal epithelial cells. *American Journal of Physiology-Gastrointestinal and Liver Physiology* 289: G129-G37
- Williams ME, Wu SC, McKenna WL, Hinck L. 2003. Surface expression of the netrin receptor UNC5H1 is regulated through a protein kinase C-interacting protein/protein kinase-dependent mechanism. *J Neurosci* 23: 11279-88
- Xu K, Wu Z, Renier N, Antipenko A, Tzvetkova-Robev D, et al. 2014. Neural migration. Structures of netrin-1 bound to two receptors provide insight into its axon guidance mechanism. *Science* 344: 1275-9
- Xu S, Liu Y, Li X, Liu Y, Meijers R, et al. 2018. The binding of DCC-P3 motif and FAK-FAT domain mediates the initial step of netrin-1/DCC signaling for axon attraction. *Cell Discov* 4: 8

REVIEWER COMMENTS

Dear Reviewers,

Finally, we want to thank all Reviewers for your time and the valuable suggestions that helped to improve our manuscript to its final state.

Reviewer #1 (Remarks to the Author):

The authors have addressed my most pressing concerns, and I support publication of the manuscript in its current form.

We thank the Reviewer for the support throughout the review of our manuscript.

Reviewer #2 (Remarks to the Author):

I am satisfied with the textual changes the authors have made to address my comments. I think the manuscript is ready for publication.

We thank the Reviewer for the support throughout the review of our manuscript.